# CORectifier: Hierarchical Trajectory Rectifications Boost Reinforcement Learning for Combinatorial Optimization

## Abstract

In the rapidly advancing field of Neural Combinatorial Optimization (NCO), Reinforcement Learning (RL) stands out for its ability to naturally enforce complex constraints. However, mainstream RL-based methods suffer from reward sparsity and sample inefficiency in the vast combinatorial action space, leading to ineffective training, subpar performance, and poor scalability. To address these challenges, we propose CORectifier, a novel NCO solver learned with hierarchical supervision incorporated to regularize the arbitrary RL exploration. Technically, we design a "rectification" mechanism for training: partial policy-predicted actions in a trajectory are probabilistically replaced with high-quality segments from reference solutions. Distinct from heatmap-guided Supervised Learning (SL) or vanilla Imitation Learning (IL) methods, CORectifier prompts the model with tri-level optimal signals, operating at the batch, instance, and sub-instance levels with fragments of random lengths injected at random decision steps. This Rectified RL (RRL) paradigm helps develop optimality-aware and sample-efficient RL learners while maintaining their sequential-decision manner for constraint satisfaction, delivering a new perspective to hybridize RL and SL/IL for solving CO problems with improved flexibility and utility of limited supervisory data. Empirical results on TSP, ATSP, PCTSP, CVRP, and KP, across synthetic and real-world benchmarks, validate the superior training efficacy, generalization and applicability of CORectifier. With a simplest greedy decoder, CORectifier achieves up to 59.7% and 26.5% performance gains over RL- and SL-based baselines, respectively, and reduces the performance gap on TSP-500 (sufficiently large for RL solvers) by up to 89.8%.

## 1 Introduction

Combinatorial Optimization (CO) (Papadimitriou & Steiglitz, 1998) serves as the foundation for numerous real-world applications, including logistics (Wang & Tang, 2021), transportation (Baty et al., 2024), portfolio optimization (Wang et al., 2023), and beyond (Paschos, 2014; Singh & Rizwanullah, 2022; Guan et al., 2024), yet remains a core challenge owing to its inherent complexity, i.e., NP-hardness. In recent years, Machine Learning (ML) has been actively employed to address CO problems (COPs), giving rise to the rapidly growing ML4CO community (Bengio et al., 2021; Cappart et al., 2021). This integration yields practical advantages in terms of both solution quality and computational efficiency (Dai et al., 2016), particularly when the instances are specifically distributed, as deep neural networks are renowned for automatic discovery of effective heuristics from data.

Within the ML4CO landscape, supervised learning (SL)-based methods (Joshi et al., 2019; Hudson et al., 2022; Fu et al., 2021) represent the most straightforward paradigm. They directly generate neural predictions (e.g., probability heatmaps) that minimizes the distance from reference solutions and then guide the solution construction or searching processes. While SL-based methods exhibit satisfactory in-distribution performance, they face key limitations: beyond the need to acquire large amount of supervised data via oracle solvers (Gurobi, Concorde, LKH, etc.), their indispensable reliance on handcrafted decoders (Sun & Yang, 2023; Li et al., 2023; 2024; Ma et al., 2025a) largely hampers applicability to COPs with more complex constraints (e.g., PCTSP, CVRP). In parallel, Reinforcement Learning (RL) for CO has emerged as a dominant paradigm (Kool et al., 2018; Kwon et al., 2020; 2021; Kim et al., 2022; Berto et al., 2024), with main strengths that its sequential

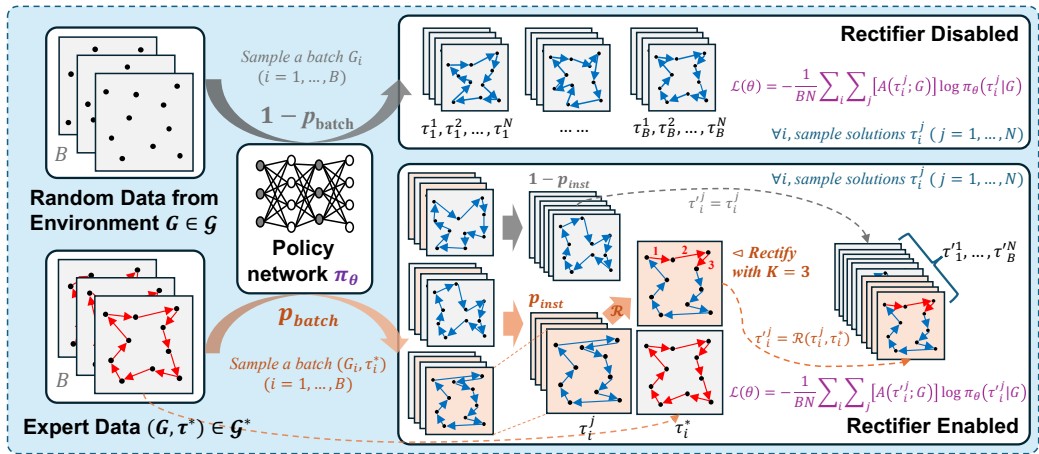

Figure 1: Rectified RL stage of CORectifier. In general, $B$-sized batches are sampled from labeled data with prob. $p_{\text{batch}}$, and for each, $N$ tours are sampled, of which a $p_{\text{inst}}$ fraction undergo rectification if enabled. $A(\cdot)$: reward-based advantage function; $K$: rectified segment length, varying at training.

decision-making manner provides greater flexibility to enforce intricate constraints during solution construction, thereby enabling better extensibility to more realistic problems or scenarios.

While promising, early RL-based approaches face fundamental bottlenecks that severely constrain their practical utility: **1) Ineffective training due to sub-optimal exploration.** Once the model converges to a plateau of decent solution quality, it becomes extremely difficult to generate improved solutions via random exploration in vast combinatorial action spaces. Thus, the probability of sampling trajectories that yield meaningful reward gains (e.g., shorter TSP tours) drops sharply, which is critical as the RL policies evolve themselves upon the advantage metric of sampled solutions over a batch-wise average baseline (Kwon et al., 2020). **2) Inefficient sampling caused by sparse rewards.** Rewards for learning COPs are typically delayed, as they are only computed after an entire trajectory is generated. This phenomenon is analogous to autoregressive language models, where prediction quality deteriorates as the generated sequence extends farther from the initial context, causing cumulative errors and instability. **3) Poor scalability and limited evaluation scope.** As widely acknowledged in prior literature (Li et al., 2025b; Ma et al., 2025a; Drakulic et al., 2025), the aforementioned limitations of RL-based CO solvers are exacerbated with increasing problem size, leaving them unable to handle graph-based COPs even with over 100 nodes. Consequently, experimental evaluations of these RL-based methods have conventionally been restricted to comparisons within the same RL family and on only quite small-scale instances (Vinyals et al., 2015; Kool et al., 2018; Kwon et al., 2021; Liao et al., 2025; Pan et al., 2025a), resulting in an incomplete assessment against other ML4CO paradigms (heatmap-guided, unsupervised, generative solving, etc.) and larger scales, e.g., TSP/ATSP $\geq$ 200.

To resolve these challenges, drawing inspiration from successful practices in broader ML domains, e.g., interleaved Imitation Learning (IL) and RL in robotics (Ball et al., 2023; Gao et al., 2025), and Teacher Forcing (TF) in sequential prediction (Williams & Zipser, 1989; Bengio et al., 2015; Lamb et al., 2016), we propose a rectified reinforcement learning paradigm for discrete optimization that incorporates hierarchical supervised signals to regularize the random exploration for combinatorial policy learning while ensuring strict compliance with task-specific constraints. During training, CORectifier probabilistically replaces segments of the model's predicted trajectory with high-quality segments derived from reference solutions. This rectification operates at a hierarchical granularity (flexibly controlled by hyper-parameters), allowing expert-guided segments of varying lengths to take effect at different solving statuses, which also aligns with the global structural properties of graph-based CO problems, avoiding the short-sightedness of local constructive methods.

Specifically, the model is first optionally trained via pure IL with few steps, followed by the rectified RL stage (depicted in Fig. 1): local rectifiers commerce their work once overfitting is observed in the IL phase. A small fraction of the sampled batches/instances fetch the rectified trajectories, whose higher-than-average rewards make them distinguishable, enabling the model to repeatedly escape local optima and be continuously guided toward exploring solution spaces with better quality. Unlike previous IL or SL attempts that used static reference trajectories or matrices, our proposed paradigm leverages reference data more efficiently at the intra-instance level, as a single training

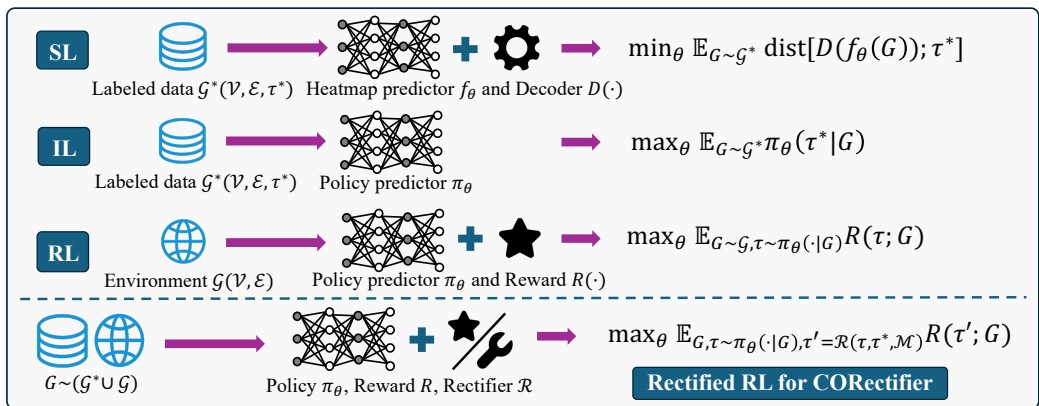

Figure 2: Mainstream NCO learning paradigms. SL minimizes the distance between model-predicted heatmaps and ground-truth matrices; IL and RL maximize the likelihood of expert trajectories or expected reward, respectively; Rectified RL aims to regularize exploration in RL via rectifiers $\mathcal{R}$.

sample (e.g., an optimal tour for a TSP graph) can generate multiple arbitrary guiding segments, as opposed to vanilla SL where each solved training instance contributes value only as an entire unit, dramatically improving the data efficiency. Furthermore, as a general-purpose framework, CORectifier can be orthogonally applied on top of existing RL-based methods to deliver consistent performance improvements, functioning as a plug-in learning mechanism. We shall also make the reusable reference datasets publicly available for future research to facilitate transparent comparison.

Extensive experiments across scales and distributions are conducted for **5 typical COPs in RL4CO literature**, i.e., Traveling Salesman Problem (TSP), Asymmetric TSP (ATSP), Prize Collecting TSP (PCTSP), Capacitated Vehicle Routing Problem (CVRP), and (0-1) Knapsack Problem (KP). It is empirically shown that CORectifier mostly outperforms existing counterparts trained by diverse paradigms, e.g., SL, RL, Unsupervised Learning (UL), or Model-Agnostic Meta-Learning (MAML).

**Major contributions of this paper can be summarized as**: **1)** We propose a new learning paradigm for solving COPs with a novel rectification mechanism to resolve RL-methods' sample efficiency and reward sparsity. **2)** To our best knowledge, CORectifier marks the first NCO attempt to explore the synergy between RL and SL/IL (as in Fig. 2), with hierarchical guiding signals operated at batch-wise, instance-wise, and intra-instance levels. **3)** We adapt our paradigm to 3 mainstream RL-based CO models and achieves enhanced training and solution quality across cases, validating the general applicability of our framework. **4)** We expand the comparison spectrum for RL-based methods to include neural solvers beyond just RL variants, showing up to **59.7%** and **26.5%** gains (w/o augmentation) over RL/SL methods, and improving RL solvers on TSP-500 by up to **89.8%**.

## 2 RELATED WORK

**Learning-free Solvers.** CPLEX (Studio, 2020) and Gurobi (Gurobi Optimization, 2023) stand as globally leading exact and general-purpose optimization solvers, while OR-Tools (Cuvelier et al., 2023) provides a suite of algorithms for diverse COPs in practice. More methods are designed for task-specific scenarios: the exact solver Concorde (Applegate et al., 2006) for 2D-TSP, the k-opt-based heuristic LKH-3 (Helsgaun, 2017) and GA-EAX (a genetic algorithm with edge crossover) (Nagata & Kobayashi, 2013) for broader TSP/ATSP; HGS (a hybrid genetic algorithm) (Vidal et al., 2012) for VRPs; and the iterated local search (ILS) metaheuristic (Lourenço et al., 2003) for the PCTSP, etc.

**Neural Heatmap-Guided Methods.** They leverage neural networks to globally *predict* the selection likelihood of each decision variable, then *decode* these probability heatmaps to generate solutions for specific COPs. Pioneering works employ vanilla graph neural networks (GNNs) as backbone, e.g., GNN4CO (Joshi et al., 2019), Att-GCN (Fu et al., 2021), and GNNGLS (Hudson et al., 2022). Recent methods adopt diffusion models and generative learning to enhance performance, e.g., DIFUSCO (Sun & Yang, 2023), T2TCO (Li et al., 2023; 2024), and DIFUCO (Sanokowski et al., 2024). Beyond the common SL paradigm, UTSP (Min et al., 2023), DIMES (Qiu et al., 2022) and Meta-EGN (Wang & Li, 2023) have explored unsupervised or meta-learning for heatmap-guided CO solving.

**Neural Sequential-Decision Methods.** These methods formulate COPs as Markov Decision Processes (MDPs), where the associated neural networks are trained to predict the optimal next-step action given the current problem state. Among these, BQ-NCO (Drakulic et al., 2023) and GOAL (Drakulic

et al., 2025) pioneered sequential-decision solvers using SL. However, most methods of this type foster a vast field of RL4CO (Berto et al., 2023; 2024). Earlier works, such as AM (Kool et al., 2018), POMO (Kwon et al., 2020), and Sym-NCO (Kim et al., 2022) were limited to the simplest small-scale 2D-TSP. Later, MatNet (Kwon et al., 2021), GLOP (Ye et al., 2024), and UniCO (Pan et al., 2025b), etc., extended the advantages of RL to the more complex ATSP task and beyond. More recent approaches, e.g., BOPO (Liao et al., 2025) and PO4CO(Pan et al., 2025a), have started to tackle the inherent drawbacks of RL by proposing strategies like Preference Optimization (PO) to boost sample efficiency, yet still remain constrained by the limited scalability and evaluated tasks.

## 3 METHODOLOGY

### 3.1 PRELIMINARY: REINFORCEMENT LEARNING FOR COMBINATORIAL OPTIMIZATION

**General[1] Problem Formulation on Graphs.** Following established convention (Joshi et al., 2019; Qiu et al., 2022; Sun & Yang, 2023; Ma et al., 2025a), we focus on graph-based CO problems where each instance is represented by a graph structure $G \in \mathcal{G} := (\mathcal{V}, \mathcal{E})$. Here, $\mathcal{G}$ denotes the set of instances for a specific COP (i.e., the training data); $\mathcal{V} = \{1, 2, ..., m\}$ denotes the set of $M$ nodes; $\mathcal{E} \subseteq \mathcal{V} \times \mathcal{V}$ denotes the set of edges, and is associated with an edge weight function $w : \mathcal{E} \to \mathbb{R}_{\geq 0}$.

The goal of a CO solver is to find a feasible solution trajectory $\tau \in \Omega$, where $\Omega$ represents the feasible region defined by problem-specific constraints. For TSP, $\tau = [v_1, v_2, ..., v_M]$ is a permutation of $\mathcal{V}$, and the solver aims to minimize the objective function $c(\tau; \mathcal{G})$, e.g., for TSP, the total travel distance of the solved Hamiltonian cycle: $c(\tau; G) = \sum_{i=1}^{M-1} w(v_i, v_{i+1}) + w(v_M, v_1)$.

**Learning a Policy.** In RL-based NCO, the solver is modeled as a stochastic policy $\pi_\theta(a_t \mid s_t; \mathcal{G})$ parameterized by $\theta$. Here, $s_t = (G \in \mathcal{G}, \tau_t = [a_1, ..., a_{t-1}])$ denotes the state at step $t$, which encodes the problem instance and the partial trajectory generated thus far. For TSP, $a_t = v_{t+1} \in \mathcal{V} \setminus \{v_1, ..., v_t\}$ denotes the action at step $t$, namely selecting the next unvisited node. As a general principle, given $s_0$ as the initial state for a specific graph instance $G$, the model is expected to ultimately characterize a probability distribution over feasible trajectories, mathematically:

$$p_\theta(\tau \mid G) \propto \pi_\theta(\tau \mid G) = \prod_{t=1}^{M'} \pi_\theta(a_t \mid s_{t-1}; G). \tag{1}$$

Practically, RL trains an agent to maximize cumulative rewards, i.e., the parameterized objective:

$$\mathcal{J}(\theta) = \mathbb{E}_{G \sim \mathcal{G}, \tau \sim \pi_\theta(\cdot | G)}[R(\tau; G)], \tag{2}$$

by interacting with an environment and estimating the REINFORCE (Williams, 1992) policy gradient:

$$\nabla_\theta \mathcal{J}(\theta) = \mathbb{E}_{\tau \sim \pi_\theta(\cdot)}\big[A(\tau; G)\nabla p_\theta(\tau \mid G)\big] = \mathbb{E}_{G \sim \mathcal{G}, \tau \sim \pi_\theta(\tau | G)}\big[A(\tau; G)p_\theta \nabla \log p_\theta(\tau \mid G)\big]$$

$$\approx \frac{1}{|\mathcal{G}|} \sum_{i=1}^{|\mathcal{G}|} \Big[\frac{1}{M'} \sum_{t=1}^{M'} \nabla_\theta \log \pi_\theta\big(a_t^{(i)} \mid s_t^{(i)}\big)\big(R(\tau; G_i) - b(G_i)\big)\Big], \tag{3}$$

where $R(\tau; G)$ is the reward function (e.g., $R(\tau; G) := -c(\tau; G)$ for TSP); $M$ is the problem size and $M'$ (often a function of $M$) is the length of a decision trajectory[2]; $\mathcal{G}$ denotes the full set of training instances; and $b(\cdot)$ denotes a baseline function (typically the average reward over a sampled batch (Kwon et al., 2020)) employed to compute the advantage term $A(\tau; G) := R(\tau; G) - b(G)$, which serves to reduce the variance of the gradient estimator. As emphasized in Sec. 1, maximizing the expected reward objective $\mathcal{J}(\theta)$ for large $M$ is challenging. The immense action space results in sparse $R(\tau; \mathcal{G})$, and more importantly, the magnitude of the advantage (i.e., $|A(\tau; \mathcal{G})|$) decreases sharply as the policy model converges to local optima, causing ineffective and unstable optimization.

### 3.2 CORECTIFIER: REINFORCEMENT LEARNING WITH HIERARCHICAL RECTIFICATIONS

**Overview.** This method aims to address the mentioned issues in RL-based NCO, as parallel complementary to recent attempts (Liao et al., 2025; Pan et al., 2025a) that target the same challenge while focusing on innovating the way the advantage signal $A(\cdot; \cdot)$ is calculated. Building on the RL-based NCO formulation in Sec. 3.1, we formally retain the core problem setting, i.e., graph

---

[1]We provide specific definitions for all the COPs covered in this work in Appendix B.

[2]For simplicity, we hereinafter use a unified term $M$, since $M' = M$ holds for the exemplary TSP case.

instances $G \in \mathcal{G}$, feasible trajectories $\tau \in \Omega$, and the objective of minimizing $\mathbb{E}_{\tau \sim \pi_\theta(\cdot|G)}[c(\tau; G)]$, but augment it with a critical absence in vanilla RL, a subset of labeled instances $\mathcal{G}^* \subset \mathcal{G}$ paired with high-quality reference trajectories $\{\tau_G^*\}_{G \in \mathcal{G}^*}$. These references are expected to satisfy a stronger quality guarantee than generic trajectories sampled from a raw RL policy after sufficient training (a conceptual proof sketch is provided in Appendix G):

$$\exists \epsilon > 0, \ \mathbb{E}_{\tau \sim \pi_\theta(\cdot|G)} \left[ c(\tau; G) - c(\tau_G^* \mid G) \right] > \epsilon, \ \forall G \in \mathcal{G}^*, \tag{4}$$

where the $\epsilon$ can be interpreted as a problem-dependent quality margin theoretically guaranteed by the (near-)optimality of labels solved by established heuristic solvers. Also, unlike vanilla IL, which uses full $\tau_G^*$ to constrain policy sampling, CORectifier *treats $\tau_G^*$ as a source of decomposable optimal signals instead of rigid templates*, enabling flexible integration with RL's exploration.

Whereas conventional RL updates policies via full trajectory gradients (Eq. 3), CORectifier introduces a rectification operator $\mathcal{R} : (\tau, \tau^*, \mathcal{M}) \mapsto \tau'$ that modifies partial segments of policy-generated trajectories $\tau$ using $\tau^*$, guided by a hierarchical mask $\mathcal{M} \in \{0, 1\}^{N \times M}$ (with $N$ as parallel trajectories per instance, $M$ as problem size). The rectified trajectory $\tau'$ retains feasibility ($\tau' \in \Omega$) and inherits local optimality from $\tau_G^*$, while preserving the policy's global exploration capacity.

This setup addresses two limitations of prior works: 1) avoiding IL's overfitting to full expert trajectories by using granular segments; 2) mitigating RL's reward sparsity by injecting local optimal signals that guide exploration toward high-quality regions. Given reward function $R(\cdot; G)$, the objective of CORectifier thus extends the vanilla RL counterpart to:

$$\max_\theta \mathbb{E}_{G \sim \mathcal{G}, \tau \sim \pi_\theta(\cdot|G), \tau' = \mathcal{R}(\tau, \tau_G^*, \mathcal{M})} \left[ R(\tau'; G) \right]. \tag{5}$$

### 3.2.1 Two-Stage Training Framework

CORectifier employs a two-stage training pipeline to progressively integrate supervised guidance into RL, negotiating between initialization stability (via imitation) and generalization (via rectified exploration). The full learning pipeline is carefully delineated in Algorithm 1 in Appendix A.

**Stage 1 (Optional): Pre-training with Imitation Learning.** To initialize the policy with reasonable prior knowledge and avoid random exploration in unpromising regions, we first pre-train using vanilla imitation learning (IL) on fully labeled instances $\mathcal{G}^*$ (with $\{\tau^*\}$). This stage aims to align the policy with expert behavior, providing a sensible starting point for subsequent RL refinement.

In this optional phase, for each batch of labeled instances $\{G_i\}_{i=1}^B \subseteq \mathcal{G}^*$ (with $B$ as batch size), we compute the IL loss by maximizing the model-predicted log-likelihood of expert trajectories $\{\tau_{G_i}^*\}$:

$$\mathcal{L}_{\text{IL}}(\theta) = -\mathbb{E}_{G \in \mathcal{G}^*} \log p_\theta(\tau_G^* \mid G) = -\frac{1}{B} \sum_{i=1}^B \left[ \log \pi_\theta(\tau_{G_i}^* \mid G_i) \right]. \tag{6}$$

Here, $\log \pi_\theta(\tau_{G_i}^* \mid G_i) = \sum_{t=1}^M \log \pi_\theta(a_{i,t}^* \mid s_{i,t})$ denotes the log-probability of the expert trajectory $\tau_{G_i}^* = [a_{i,1}^*, \ldots, a_{i,M}^*]$, with $s_{i,t}$ being the state at step $t$ (e.g., nodes visited up to $t-1$ for TSP). The policy is updated via gradient descent and runs until approximate convergence (e.g., monitored via validation loss), ensuring the policy learns basic patterns from expert trajectories without overfitting.

**Stage 2: Hierarchical Rectified Reinforcement Learning (RRL).** The core training phase integrates hierarchical rectification into RL, dynamically injecting expert segments into policy-generated trajectories, balancing three aspects: 1) *Exploration*: free policy sampling to discover novel solutions), 2) *Exploitation*: leveraging learned patterns to refine good solutions, and 3) *Expert guidance*: incorporating optimal segments from $\tau_G^*$. The mechanism is designed with tri-leveled operations.

**1) Batch Level: Probabilistically Sourced Training Data.** At each batch iteration, we probabilistically select between labeled and unlabeled instances using a batch-wise rectification probability $p_{\text{batch}} \in (0, 1)$. Formally, with $\mathcal{G}$ denoting the entire space of unlabeled problem instances, we sample

$$\{G_i\}_{i=1}^B \sim \begin{cases} \mathcal{G}^* & \text{with probability } p_{\text{batch}} \\ \mathcal{G} \setminus \mathcal{G}^* & \text{with probability } 1 - p_{\text{batch}} \end{cases}. \tag{7}$$

While $\mathcal{G}^*$ remains fixed as a training set with optimal solutions solved by oracle solvers, batches from $\mathcal{G} \setminus \mathcal{G}^*$ are randomly generated on the fly from the infinite problem instance space. This design ensures the policy generalizes to unseen instances while periodically leveraging high-quality guidance.

**2) Instance Level: Trajectory Sampling and Rectification Candidates.** For each instance $G_i$ in the batch, we sample $N = \min(M, 200)$ (the truncation is for memory efficiency) parallel trajectories $\mathcal{T}_i = \{\tau_i^j\}_{j=1}^N \sim \pi_\theta(\cdot \mid G_i)$ as in Kwon et al. (2020). Prior to the calculation of the REINFORCE baseline, a subset $\widehat{\mathcal{T}}_i \subseteq \mathcal{T}_i$ is randomly selected for rectification, where the proportion of rectified trajectories is controlled by $p_{\text{inst}}$ (instance-wise rectification probability):

$$|\widehat{\mathcal{T}}_i|/|\mathcal{T}_i| = p_{\text{inst}}. \tag{8}$$

This controls the density of expert guidance within parallel samples corresponding to each graph instance in a mini-batch. It ensures that partial trajectories stand out against the mean baseline with distinct advantages, while preventing the policy from being overwhelmed by reference signals.

**3) Intra-instance Level: Masked Rectifiers with Varied Lengths.** For trajectories in $\widehat{\mathcal{T}}_i$ (w.r.t. $G_i$), we generate a binary rectifier mask $\mathcal{M}_{G_i} \in \{0,1\}^{N \times M}$ to specify which decision steps (actions) in a trajectory shall undergo rectification. This mask is constructed as follows:

$$\mathcal{M}_{G_i} = \text{Concat}\left(\mathbf{1}^{N \times k}; \mathbf{0}^{N \times (M-k)}\right), \tag{9}$$

where $\mathbf{1}$ and $\mathbf{0}$ denote matrices filled with ones and zeros; $\text{Concat}(\cdot; \cdot)$ denotes concatenation; $k = \lfloor \gamma M \rfloor$ represents the length of the rectified segment; and $\gamma \sim \text{Uniform}(\alpha, \beta)$ (with $[\alpha, \beta]$ as hyper-parameters defining the segment length range[3]). The mask is randomly rolled along the step dimension to ensure rectification occurs at diverse positions. This design enables supervision to operate at the sub-instance level (i.e., on segments of a trajectory), allowing a single reference solution $\tau^*$ to be reused arbitrarily throughout the training process, thereby improving the data utility rate.

Subsequently, for each trajectory $\tau_i^j \in \widehat{\mathcal{T}}_i$ and action step $t \in \{2, \ldots, M\}$, the masked rectification operates iteratively following the three steps below:

• **Recommended Action Retrieval**: Given the last action $a_{i,t-1} \in \tau_i^j$, retrieve the next action $a_{i,t}^*$ from the reference trajectory, i.e., the direct successor of $a_{i,j,t-1}$ in $\tau_{G_i}^*$.

• **Constraint Check**: Verify the corresponding state of $a_{i,t}^*$, i.e., $s_{i,j,t}$, is feasible (e.g., unvisited node in TSP). Let $\text{Feas}(a_{i,t}^*, s_{i,j,t}) = 1$ if feasible, 0 otherwise.

• **Action Replacement**: If $\mathcal{M}_{G_i}[j,t] = 1$ (mask allows rectification) and $\text{Feas}(a_{i,t}^*, s_{i,j,t}) = 1$, replace the policy-selected action $a_{i,j,t} \sim \pi_\theta(\cdot \mid s_{i,j,t})$ with $a_{i,t}^*$. Formally,

$$\tau_i'^j[t] \leftarrow \begin{cases} a_{i,t}^* & \text{if } \mathcal{M}_{G_i}[j,t] = 1 \cap \text{Feas}(a_{i,t}^*, s_{i,j,t}) = 1, \\ a_{i,j,t} & \text{otherwise}. \end{cases} \tag{10}$$

The rectified actions in this process are largely ensured to be feasible ($\tau_i'^j \in \Omega_{G_i}$), as the rectification operation always replaces segments of the original trajectory with a sequence of *continuous* actions that shall logically occur in succession within the guiding reference solutions.

**RRL Loss and Policy Update.** After rectification, we compute rewards ($R_{i,j}$) and update the policy via REINFORCE with advantage ($A_{i,j}$) estimation as in Eq. 3, i.e., $A_{i,j} = R_{i,j} - \frac{1}{N}\sum_{j'=1}^N R_{i,j'}$, where the batch mean serves as a baseline. The policy gradient loss is defined as the negative expected advantage-weighted log-probability of trajectories, i.e., $\min_\theta \mathcal{L}_{\text{RRL}}(\theta)$ :

$$\mathcal{L}_{\text{RRL}}(\theta) = -\mathbb{E}_{G\sim\mathcal{G}, \tau\sim\pi_\theta(\cdot|G), \tau'=\mathcal{R}(\tau, \tau_G^*, \mathcal{M})} A(\tau'; G) \cdot \log \pi_\theta(\tau' \mid G), \tag{11}$$

$$\nabla_\theta \mathcal{L}_{\text{RRL}}(\theta) \approx -\frac{1}{BN}\sum_{i=1}^B\sum_{j=1}^N A_{i,j} \cdot \nabla_\theta \log \pi_\theta(\tau_i'^j \mid G_i), \tag{12}$$

where $\log \pi_\theta(\tau_i'^j \mid G_i) = \sum_{t=1}^M \log \pi_\theta(a_{i,j,t}' \mid s_{i,j,t})$ with $a_{i,j,t}'$ being the rectified action if applicable. The policy is updated by gradient descent $\theta \leftarrow \theta - \eta\nabla_\theta\mathcal{L}_{\text{RRL}}(\theta)$ with learning rate $\eta$.

**Dynamic Scheduling of Rectifiers.** To dynamically adjust the intensity of expert guidance druing training, we adopt a separate cosine-annealing scheduler (borrowing its successful use in adjusting learning rate, etc.) for each parameter $\mathbf{p} \in \{p_{\text{batch}}, p_{\text{inst}}, \alpha, \beta\}$ with distinct cycle lengths ($T_{\max}$):

$$\mathbf{p}(t) = \mathbf{p}_{\min} + 0.5\left(\mathbf{p}_0 - \mathbf{p}_{\min}\right)\left[1 + \cos\left(\pi t/T_{\max}\right)\right], \tag{13}$$

---

[3]This applies for the TSP family while we detail the similar operation for CVRP and KP in Appendix C.2.

Table 1: Main results on TSP. G: Greedy decoding. RL: Reinforcement Learning. SL: Supervised Learning. UL: Unsupervised Learning. As the first recent RL-based method compared with heatmap solvers and evaluated on TSP $\geq 500$, we hereinafter **bold** ours that top within the RL/Sequential family, and highlight those surpassing SL-based SOTAs on corresponding settings (w/ or w/o 2Opt).

| METHOD | TYPE | TSP-50 (1280 inst.) | | | TSP-100 (1280 inst.) | | | TSP-500 (128 inst.) | | |
|---|---|---|---|---|---|---|---|---|---|---|
| | | OBJ.↓ | DROP↓ | TIME↓ | OBJ.↓ | DROP↓ | TIME↓ | OBJ.↓ | DROP↓ | TIME↓ |
| Concorde (Applegate et al., 2006) | Exact | 5.688* | 0.000% | 0.059s | 7.756* | 0.000% | 0.238s | 16.546* | 0.000% | 18.672s |
| LKH-3 (500) (Helsgaun, 2017) | Heuristics | 5.688 | 0.001% | 0.058s | 7.756 | 0.001% | 0.176s | 16.546 | 0.002% | 1.848s |
| *Heatmap Guided Methods* | | | | | | | | | | |
| GNN4CO (Joshi et al., 2019) | SL+G | 5.718 | 0.520% | 0.006s | 7.937 | 2.334% | 0.007s | 18.148 | 9.683% | 0.047s |
| DIMES (Qiu et al., 2022) | RL+G | 6.325 | 11.233% | 0.003s | 8.757 | 12.918% | 0.005s | 18.888 | 14.155% | 0.273s |
| GNNGLS (Hudson et al., 2022) | SL+G | 6.179 | 8.651% | 0.019s | 8.966 | 15.597% | 0.021s | – | – | – |
| UTSP (Min et al., 2023) | UL+G | 7.097 | 24.808% | 0.007s | 9.377 | 20.909% | 0.009s | 19.761 | 19.435% | 0.059s |
| Fast-T2T (Li et al., 2024) | SL+G | 5.723 | 0.612% | 0.007s | 7.941 | 2.381% | 0.009s | 18.176 | 9.840% | 0.050s |
| COExpander (Ma et al., 2025a) | SL+G | 5.702 | 0.244% | 0.018s | 7.846 | 1.164% | 0.020s | 17.634 | 6.574% | 0.062s |
| GNN4CO (Joshi et al., 2019) | SL+G+2Opt | 5.691 | 0.066% | 0.007s | 7.775 | 0.243% | 0.009s | 16.769 | 1.348% | 0.063s |
| DIMES (Qiu et al., 2022) | RL+G+2Opt | 5.891 | 3.578% | 0.007s | 8.108 | 4.543% | 0.012s | 17.655 | 6.707% | 0.314s |
| GNNGLS (Hudson et al., 2022) | SL+G+2Opt | 5.707 | 0.333% | 0.020s | 7.857 | 1.295% | 0.129s | – | – | – |
| UTSP (Min et al., 2023) | UL+G+2Opt | 5.976 | 5.080% | 0.012s | 8.161 | 5.215% | 0.017s | 17.710 | 7.033% | 0.090s |
| Fast-T2T (Li et al., 2024) | SL+G+2Opt | 5.691 | 0.065% | 0.007s | 7.773 | 0.223% | 0.009s | 16.750 | 1.233% | 0.055s |
| COExpander (Ma et al., 2025a) | SL+G+2Opt | 5.689 | 0.029% | 0.018s | 7.765 | 0.124% | 0.021s | 16.684 | 0.837% | 0.070s |
| *Sequential Decision Methods* | | | | | | | | | | |
| AM (Kool et al., 2018) | RL+G | 5.747 | 1.040% | 0.008s | 7.951 | 2.520% | 0.033s | – | – | – |
| POMO (Kwon et al., 2020) | RL+G | 5.698 | 0.179% | 0.142s | 7.883 | 1.641% | 0.151s | – | – | – |
| Sym-NCO (Berto et al., 2023) | RL+G | 5.702 | 0.258% | 0.050s | 7.872 | 1.491% | 0.098s | – | – | – |
| GOAL (Drakulic et al., 2025) | SL+G | 5.742 | 0.965% | 0.393s | 7.905 | 1.920% | 0.762s | 18.006 | 8.825% | 3.964s |
| CORectifier | RL+G | **5.697** | **0.157%** | 0.027s | **7.818** | **0.795%** | 0.051s | **17.359** | **4.915%** | 0.239s |
| CORectifier ($N_A$=16) | RL+G | **5.689** | **0.017%** | 0.029s | **7.782** | **0.338%** | 0.056s | **17.247** | **4.241%** | 1.652s |
| CORectifier ($N_A$=128) | RL+G | **5.688** | **0.009%** | 0.068s | **7.775** | **0.249%** | 0.232s | **17.216** | **4.024%** | 12.138s |
| AM (Kool et al., 2018) | RL+G+2Opt | 5.733 | 0.799% | 0.079s | 7.902 | 1.887% | 0.184s | – | – | – |
| POMO (Kwon et al., 2020) | RL+G+2Opt | 5.693 | 0.103% | 0.182s | 7.854 | 1.261% | 0.225s | – | – | – |
| Sym-NCO (Berto et al., 2023) | RL+G+2Opt | 5.699 | 0.197% | 0.051s | 7.840 | 1.087% | 0.100s | – | – | – |
| CORectifier | RL+G+2Opt | 5.695 | 0.132% | 0.027s | **7.807** | **0.664%** | 0.052s | **17.140** | **3.593%** | 0.389s |
| CORectifier ($N_A$=16) | RL+G+2Opt | **5.688** | **0.011%** | 0.030s | **7.777** | **0.276%** | 0.055s | **17.068** | **3.154%** | 1.668s |
| CORectifier ($N_A$=128) | RL+G+2Opt | **5.688** | **0.007%** | 0.069s | **7.771** | **0.198%** | 0.249s | **17.039** | **2.978%** | 12.912s |

where $p_{min}$ is the lower bound for the scheduled parameter, $p_0$ is the initial value, and $t$ is the training epoch. The independent cosine-periodic updates generate diverse combinations of expert guidance modes across training, exposing the model to varying degrees of reference reliance (e.g., high $p_{batch}$ with low $\beta$, or low $p_{inst}$ with high $\alpha$) over epochs, enhancing the robustness of the learned policy.

# 4 EXPERIMENTS

## 4.1 PROTOCOL

**Hardware.** We train and test all models on a single NVIDIA RTX3090 24GB GPU with AMD Ryzen Threadripper 3970X 32-Core CPU to ensure comparative transparency and reproducibility.

**Datasets.** For **TSP**, we follow Kool et al. (2018); Joshi et al. (2019); Kwon et al. (2020); Qiu et al. (2022); Sun & Yang (2023) to sample graph instances with $M \in \{50, 100, 500\}$ nodes from the unit square $[0, 1]^2$, and obtain reference tours by exact solver Concorde (Applegate et al., 2006). For **ATSP**, adhering to Kwon et al. (2021); Ye et al. (2024); Drakulic et al. (2025); Pan et al. (2025b), we generate instances with $M \in \{50, 100, 200, 500\}$ nodes, where pairwise asymmetric distances satisfy the triangle inequality, and use LKH-3 (Helsgaun, 2017) to compute reference tours. For **PCTSP**, following Kool et al. (2018); Kim et al. (2022); Li et al. (2025a), we generate instances with $M \in \{20, 50, 100\}$ nodes, and gain reference tours by OR-Tools (Cuvelier et al., 2023) and ILS (Lourenço et al., 2003). For **CVRP**, we follow Ma et al. (2025a) to generate instances with $M \in \{50, 100, 200, 500\}$ clients, the demands of which are integers randomly sampled from $[1, 10]$, and the vehicle capacity is set as 40/50/80/100, respectively. The labels are solved by HGS (Vidal et al., 2012). For **KP**, we follow Kwon et al. (2020) to generate instances with $M \in \{50, 100, 200, 500\}$ items, the weights/values of which are uniformly sampled from $[0, 1]$, and the knapsack capacity is set as 12.5/25/25/25, respectively. The reference solutions for KP are computed by OR-Tools.

**Metrics.** 1) **OBJ.**: average objective value (e.g., tour length) on certain task. 2) **DROP**: relative drop of the objective compared to the reference solutions. 3) **TIME**: per-instance average solving time.

**Compared ML-based Methods.** Unlike previous literature, which primarily limited comparisons of RL-based approaches to similar constructive methods on small-scale instances, we expand the evaluation by incorporating both **1) classic sequential-decision methods:** AM (Kool et al., 2018), POMO (Kwon et al., 2020), Sym-NCO (Kim et al., 2022), MatNet (Kwon et al., 2021), UniCO (Pan et al., 2025b), GOAL (Drakulic et al., 2025), BQ-NCO (Drakulic et al., 2023), PO4CO (Pan et al.,

Table 2: Main results on ATSP. Note CORectifier is the first recent RL-based method transparently compared with SOTA heatmap guided SL solvers and evaluated on ATSP ≥ 200.

| METHOD | TYPE | ATSP-50 (2500 inst.) | | | ATSP-100 (2500 inst.) | | |
|---|---|---|---|---|---|---|---|
| | | OBJ.↓ | DROP↓ | TIME↓ | OBJ.↓ | DROP↓ | TIME↓ |
| LKH-3 (1000) (Helsgaun, 2017) | Heuristics | 1.554* | 0.000% | 0.097s | 1.566* | 0.000% | 0.238s |
| *Heatmap Guided Methods* | | | | | | | |
| GNN4CO (Joshi et al., 2019) | SL+G | 1.667 | 7.261% | 0.007s | 1.666 | 6.408% | 0.008s |
| Fast-T2T (Li et al., 2024) | SL+G | 1.662 | 6.948% | 0.006s | 1.665 | 6.329% | 0.008s |
| COExpander (Ma et al., 2025a) | SL+G | 1.637 | 5.304% | 0.019s | 1.646 | 5.086% | 0.023s |
| GNN4CO (Joshi et al., 2019) | SL+G+2Opt | 1.613 | 3.795% | 0.008s | 1.629 | 4.004% | 0.011s |
| Fast-T2T (Li et al., 2024) | SL+G+2Opt | 1.614 | 3.805% | 0.007s | 1.630 | 4.104% | 0.009s |
| COExpander (Ma et al., 2025a) | SL+G+2Opt | 1.601 | 3.029% | 0.019s | 1.619 | 3.361% | 0.024s |
| *Sequential Decision Methods* | | | | | | | |
| MatNet (Kwon et al., 2021) | RL+G | 1.576 | 1.373% | 0.032s | 1.620 | 3.456% | 0.053s |
| UniCO (Pan et al., 2025b) | RL+G | 1.579 | 1.601% | 0.044s | 1.625 | 3.768% | 0.060s |
| GOAL (Drakulic et al., 2025) | SL+G | 1.595 | 2.567% | 0.395s | 1.644 | 5.010% | 0.723s |
| CORectifier | RL+G | **1.565** | 0.649% | 0.036s | **1.607** | 2.633% | 0.079s |
| CORectifier (N_A=16) | RL+G | **1.556** | 0.115% | 0.047s | **1.585** | 1.192% | 0.120s |
| MatNet (Kwon et al., 2021) | RL+G+2Opt | 1.574 | 1.256% | 0.032s | 1.616 | 3.199% | 0.057s |
| GOAL (Drakulic et al., 2025) | SL+G+2Opt | 1.587 | 2.079% | 0.396s | 1.632 | 4.195% | 0.724s |
| CORectifier | RL+G+2Opt | **1.564** | 0.585% | 0.036s | **1.604** | 2.394% | 0.081s |
| CORectifier (N_A=16) | RL+G+2Opt | **1.556** | 0.113% | 0.048s | **1.584** | 1.127% | 0.121s |
| CORectifier (N_A=128) | RL+G+2Opt | **1.555** | 0.032% | 0.186s | **1.576** | 0.603% | 0.557s |

| METHOD | TYPE | ATSP-200 (100 inst.) | | | ATSP-500 (100 inst.) | | |
|---|---|---|---|---|---|---|---|
| | | OBJ.↓ | DROP↓ | TIME↓ | OBJ.↓ | DROP↓ | TIME↓ |
| LKH-3 (1000) (Helsgaun, 2017) | Heuristics | 1.565* | 0.000% | 0.724s | 1.573* | 0.000% | 4.376s |
| *Heatmap Guided Methods* | | | | | | | |
| GNN4CO (Joshi et al., 2019) | SL+G | 1.671 | 6.777% | 0.053s | 1.729 | 9.880% | 0.157s |
| Fast-T2T (Li et al., 2024) | SL+G | 1.678 | 7.280% | 0.037s | 1.675 | 6.443% | 0.127s |
| COExpander (Ma et al., 2025a) | SL+G | 1.628 | 4.078% | 0.099s | 1.646 | 4.634% | 0.403s |
| GNN4CO (Joshi et al., 2019) | SL+G+2Opt | 1.634 | 4.391% | 0.070s | 1.696 | 7.774% | 0.313s |
| Fast-T2T (Li et al., 2024) | SL+G+2Opt | 1.652 | 5.608% | 0.046s | 1.653 | 5.088% | 0.231s |
| COExpander (Ma et al., 2025a) | SL+G+2Opt | 1.616 | 3.293% | 0.111s | 1.630 | 3.592% | 0.478s |
| *Sequential Decision Methods* | | | | | | | |
| MatNet (Kwon et al., 2021) | RL+G | 3.831 | 145.083% | 0.114s | – | – | – |
| CORectifier | RL+G | **1.649** | 5.407% | 0.145s | **1.710** | 8.699% | 0.377s |
| CORectifier | RL+G+2Opt | **1.637** | 4.638% | 0.149s | **1.692** | 7.510% | 0.552s |
| CORectifier (N_A=128 / 64) | RL+G+2Opt | **1.614** | 3.123% | 2.351s | **1.670** | 6.689% | 9.351s |

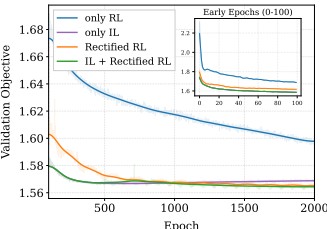

Figure 3: Train curves on ATSP-50 (greedy decoding w/o 2Opt).

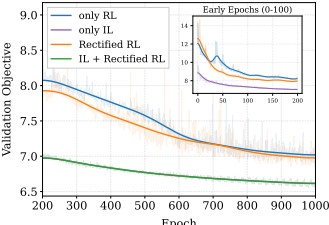

Figure 4: Train curves on PCTSP-100 (G w/o 2Opt).

Table 3: Ablation results (Drop ↓) on 50-node instances.

| TASK | TSP | ATSP | PCTSP |
|---|---|---|---|
| vanilla RL | 0.197% | 1.639% | 3.754% |
| vanilla IL | 1.818% | 2.373% | 4.020% |
| ours w/o IL | 0.287% | 0.853% | **3.407%** |
| ours w/ IL | **0.186%** | **0.649%** | 3.593% |

Table 4: Main results on PCTSP. Results are partially quoted[†] from (Li et al., 2025a). f-t: finetune.

| METHOD | TYPE | PCTSP-20 (10000 inst.) | | | PCTSP-50 (10000 inst.) | | | PCTSP-100 (10000 inst.) | | |
|---|---|---|---|---|---|---|---|---|---|---|
| | | OBJ.↓ | DROP↓ | TIME↓ | OBJ.↓ | DROP↓ | TIME↓ | OBJ.↓ | DROP↓ | TIME↓ |
| OR-Tools (60s) (Cuvelier et al., 2023) | Heuristics | 3.134* | 0.000% | 60.005s | 4.477* | 0.000% | 60.019s | 5.994 | 0.231% | 60.041s |
| Gurobi (10s) (Gurobi Optimization, 2023) | Exact | 3.134 | -0.003% | 0.330s | 4.531 | 1.183% | 6.037s | – | – | – |
| ILS (C++) (Lourenço et al., 2003) | Heuristics | 3.159 | 0.783% | 0.148s | 4.499 | 0.491% | 1.256s | 5.981* | 0.000% | 6.860s |
| *Sequential Decision Methods* | | | | | | | | | | |
| AM (Kool et al., 2018) | RL+G | 3.233 | 3.176% | 0.053s | 4.633 | 3.511% | 0.095s | 6.429 | 7.506% | 0.167s |
| MCOMAB[†] (f-t) (Wang & Yu, 2023) | RL+G | 3.34 | 6.79% | – | 4.56 | 1.83% | – | – | – | – |
| GCNCO[†] (f-t) (Li et al., 2025a) | RL+G | 3.26 | 4.25% | – | 4.54 | 1.64% | – | – | – | – |
| CORectifier | RL+G | **3.150** | 0.504% | 0.465s | **4.541** | 1.445% | 0.865s | **6.206** | **3.775%** | 1.638s |
| CORectifier (N_A=16) | RL+G | **3.147** | 0.403% | 0.782s | **4.537** | 1.348% | 1.427s | **6.201** | **3.695%** | 2.629s |

2025a), Omni-POMO (Zhou et al., 2023), LCP (Kim et al., 2021), etc.; and **2) recent powerful heatmap-guided methods:** GNN4CO (Joshi et al., 2019), DIMES (Qiu et al., 2022), GNNGLS (Hudson et al., 2022), UTSP (Min et al., 2023), Fast-T2T (Li et al., 2024), COExpander (Ma et al., 2025a).

**Model Settings.** During training, we set $p_{batch} = p_{inst} = 0.1$, $[\alpha, \beta] = [0.1, 0.2]$ for TSP, PCTSP, and CVRP, and set them as $0.5$, $[0.8, 1.0]$ for ATSP. At testing, the batch-size is set as 1 with greedy decoding for all compared methods. $N_A$ denotes data augmentation following (Kwon et al., 2021). Backbones are introduced in Appendix D, and detailed settings are provided in Appendix E.

### 4.2 MAIN RESULTS AND ANALYSIS

Experimental results are given in Table 1, 2, 4, and 5 for TSP, ATSP, PCTSP, and CVRP, respectively.

**Comparison within RL-based Methods.** CORectifier outperforms compared RL-based solvers (greedy), achieving reduced optimality drops by **47.69%**, **32.05%**, **59.67%**, **37.22%** on average with 50–100-node TSP, ATSP, PCTSP, and CVRP (a range where RL-based solvers prevail) respectively. Beyond these, we further extend the scalability of RL-based methods whereby larger CO instances could also enjoy the benefits from the constraint-friendly sequential RL learners (i.e., TSP/CVRP> 100). We also discover and discuss the substantial enhancement on TSP-500, excellent zero-shot generalization, and broader applicability on non-routing tasks (e.g., KP) in the next sub-section.

**Comparison with Broader ML-based Approaches.** On TSP and ATSP, domains dominated by SL-based heatmap-guided solvers, CORectifier delivers superior solution quality on 6 out of 7 test benchmarks (highlighted in `cyan`). When decoding greedily without any augmentations, CORectifier reduces the optimality drop by an overall **26.49%** (TSP) and **8.98%** (ATSP) across

Table 5: Main results on CVRP. Baselines are either quoted[†] or re-implemented upon recent benchmarks (Berto et al., 2023; Ma et al., 2025b). LS: classic Local Search, proposed in Ma et al. (2025a).

| METHOD | TYPE | CVRP-50 (10000 inst.) | | | CVRP-100 (10000 inst.) | | |
|---|---|---|---|---|---|---|---|
| | | OBJ.↓ | DROP↓ | TIME↓ | OBJ.↓ | DROP↓ | TIME↓ |
| HGS (Vidal et al., 2012) | Heuristics | 10.366* | 0.000% | 1.005s | 15.563* | 0.000% | 20.027s |
| *Heatmap Guided Methods* | | | | | | | |
| Fast-T2T (Li et al., 2024) | SL+G | 12.640 | 21.835% | 0.009s | 19.202 | 23.333% | 0.010s |
| COExpander (Ma et al., 2025a) | SL+G | 11.979 | 15.407% | 0.033s | 17.497 | 12.343% | 0.047s |
| Fast-T2T (Li et al., 2024) | SL+G+LS | 10.871 | 4.836% | 0.013s | 16.294 | 4.698% | 0.018s |
| COExpander (Ma et al., 2025a) | SL+G+LS | 10.773 | 3.903% | 0.037s | 16.224 | 4.253% | 0.055s |
| *Sequential Decision Methods* | | | | | | | |
| AM[†] (Kool et al., 2018) | RL+G | 10.98 | 5.86% | – | 16.80 | 7.34% | – |
| POMO[†] (Kwon et al., 2020) | RL+G | 10.74 | 3.52% | – | 16.15 | 3.00% | – |
| Sym-NCO (Kim et al., 2022) | RL+G | 10.769 | 3.891% | 0.087s | 16.220 | 4.241% | 0.166s |
| GOAL (Drakulic et al., 2025) | SL+G | 10.906 | 5.193% | 0.504s | 16.342 | 5.005% | 0.962s |
| CORectifier | RL+G | **10.540** | **1.668%** | 0.042s | **15.939** | **2.425%** | 0.079s |
| CORectifier (N_A=16) | RL+G | **10.447** | **0.768%** | 0.049s | **15.799** | **1.521%** | 0.108s |
| Sym-NCO (Kim et al., 2022) | RL+G+LS | 10.505 | 1.910% | 0.168s | 15.933 | 2.379% | 0.173s |
| GOAL (Drakulic et al., 2025) | SL+G+LS | 10.628 | 2.519% | 0.507s | 15.959 | 2.548% | 0.969s |
| CORectifier | RL+G+LS | **10.469** | **0.984%** | 0.045s | **15.796** | **1.496%** | 0.085s |
| CORectifier (N_A=16) | RL+G+LS | **10.412** | **0.437%** | 0.052s | **15.706** | **0.919%** | 0.113s |

| METHOD | TYPE | CVRP-200 (100 inst.) | | | CVRP-500 (100 inst.) | | |
|---|---|---|---|---|---|---|---|
| | | OBJ.↓ | DROP↓ | TIME↓ | OBJ.↓ | DROP↓ | TIME↓ |
| HGS (Vidal et al., 2012) | Heuristics | 19.630* | 0.000% | 60.024s | 37.154* | 0.000% | 360.376s |
| *Heatmap Guided Methods* | | | | | | | |
| Fast-T2T (Li et al., 2024) | SL+G | 25.064 | 27.616% | 0.059s | 47.749 | 28.509% | 0.091s |
| COExpander (Ma et al., 2025a) | SL+G | 22.402 | 13.977% | 0.145s | 43.901 | 18.199% | 0.554s |
| Fast-T2T (Li et al., 2024) | SL+G+LS | 20.662 | 5.290% | 0.063s | 39.195 | 5.530% | 0.215s |
| COExpander (Ma et al., 2025a) | SL+G+LS | 20.587 | 4.893% | 0.153s | 39.121 | 5.337% | 0.605s |
| *Sequential Decision Methods* | | | | | | | |
| Sym-NCO (Kim et al., 2022) | RL+G | 20.662 | 5.274% | 0.320s | 40.382 | 8.723% | 0.769s |
| CORectifier | RL+G | **20.270** | **3.260%** | 0.159s | **39.129** | **5.343%** | 0.368s |
| CORectifier (N_A=16) | RL+G | **20.129** | **2.541%** | 0.219s | **38.874** | **4.650%** | 1.485s |
| Sym-NCO (Kim et al., 2022) | RL+G+LS | 20.193 | 2.880% | 0.341s | 38.700 | 4.173% | 0.883s |
| CORectifier | RL+G+LS | **20.032** | **2.052%** | 0.171s | **38.329** | **3.185%** | 0.461s |
| CORectifier (N_A=16) | RL+G+LS | **19.952** | **1.638%** | 0.237s | **38.176** | **2.758%** | 1.567s |
| CORectifier (N_A=128) | RL+G+LS | **19.932** | **1.534%** | 0.990s | **38.150** | **2.697%** | 10.85s |

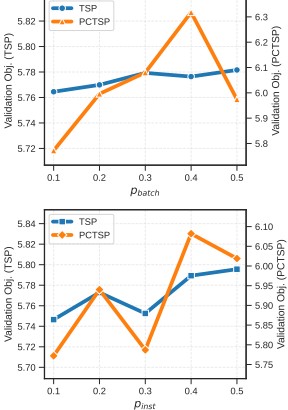

Figure 5: Effect of rectification probabilities $p_{\text{batch}}$ and $p_{\text{inst}}$ on TSP and PCTSP.

Table 6: Effect (OBJ.) of the rectified length ratio range $(\alpha, \beta)$ on TSP and PCTSP.

| $[\alpha, \beta]$ | TSP-50 | PCTSP-50 |
|---|---|---|
| [0.1, 0.2] | 5.770 | 5.722 |
| [0.2, 0.3] | 5.778 | 5.671 |
| [0.3, 0.4] | 5.799 | 5.732 |
| [0.4, 0.5] | 5.822 | 5.548 |
| [0.5, 1.0] | 5.782 | 5.688 |

50–500-node instances, respectively. Admittedly, heatmap-based solvers are still powerful for large-scale TSP/ATSP, however, they can hardly model CO tasks with more complex constraints (e.g., PCTSP, CVRP, and KP), which highlights the necessity of progressively advancing RL-driven solvers.

## 4.3 EXTENDED EVALUATIONS

**Ablation Study.** We train the models for 2000 epochs across different learning paradigms (with greedy decoding) and present the ablation results in Table 3. Notably, our rectified RL paradigm consistently outperforms its vanilla counterparts. The superior result on PCTSP without IL pre-training also informs our decision on whether to enable the optional stage for specific tasks. Fig. 3/4 further validate the efficacy of our design by ablating on 3 baselines: vanilla RL, IL, and CORectifier (w/ and w/o IL pre-training), confirming CORectifier's synergistic performance gain over conventional RL and IL.

Table 7: Generalization results on TSPLIB[50-200] (Li et al., 2023).

| METHOD | DROP↓ |
|---|---|
| GNN4CO (Joshi et al., 2019) | 40.04% |
| AM (Vaswani et al., 2017) | 16.77% |
| POMO (Kwon et al., 2020) | 3.20% |
| ICAM (Zhou et al., 2024) | 2.38% |
| Learn2OPT (da Costa et al., 2020) | 1.73% |
| GNNGLS (Hudson et al., 2022) | 1.53% |
| CORectifier (ours) | **1.05%** |

**Generalization Study.** By convention, we evaluate our models (trained on uniform TSP/CVRP-100) on real-world TSPLIB with 50-200 nodes (Table 7) and CVRPLIB-Set-X with 100-1000 clients (Table 8). Note CORectifier outperforms all compared baselines, showing remarkably reduced optimality gap over popular RL solvers (**ours: 1.05%** v.s. AM: 16.77%/POMO: 3.20% for TSP; **ours: 5.47%** v.s. Sym-NCO: 23.32%/POMO: 10.37% for CVRP). We leave the per-instance results and extended analyses on TSPLIB and CVRPLIB in Appendix F.1 and F.2, respectively.

Table 8: Generalization results on CVRPLIB (Pan et al., 2025a).

| METHOD | DROP↓ |
|---|---|
| Sym-NCO (Kim et al., 2022) | 23.32% |
| POMO (Kwon et al., 2020) | 10.37% |
| LEHD (Luo et al., 2023) | 12.25% |
| BQ-NCO (Drakulic et al., 2023) | 10.89% |
| Omni-POMO (Zhou et al., 2023) | 6.52% |
| ELG (RF) (Gao et al., 2024) | 6.03% |
| ELG (PO) (Pan et al., 2025a) | 5.94% |
| CORectifier (ours) | **5.47%** |

Jointly, they exhibit robust generalizability over unseen instances with varying scales and distributions.

**Hyper-parameter Studies.** We conduct ablative experiments to analyze the impact of three core hyper-parameters that govern our tri-level hierarchical rectification mechanism: **1) Batch-wise Rectification Probability** ($p_{\text{batch}}$), **2) Instance-wise Rectification Probability** ($p_{\text{inst}}$), and **3) Length Range of Rectified Segments** ($[\alpha, \beta]$). As illustrated in Fig. 5, the model's performance remains relatively stable as $p_{\text{batch}}$ and $p_{\text{inst}}$ vary across different values, with a slight degradation is observed when these probabilities increase. This empirical trend validates our choice of setting both $p_{\text{batch}}$ and $p_{\text{inst}}$ to 0.1 for POMO and AM models. For the rectified segment length range $[\alpha, \beta]$, Table 6 shows a

Figure 6: L: Advantage separation. M: Advantage scale distribution. R: Trajectory entropy. TSP-100.

Table 9: Main results on KP. We generate test data following Kwon et al. (2020); Drakulic et al. (2025) and quote the relative drop results from them for the best of comparability.

| METHOD | TYPE | KP-50 (1280 inst.) | | KP-100 (1280 inst.) | | KP-200 (1280 inst.) | | KP-500 (1280 inst.) | |
|---|---|---|---|---|---|---|---|---|---|
| | | OBJ.↑ | DROP↓ | OBJ.↑ | DROP↓ | OBJ.↑ | DROP↓ | OBJ.↑ | DROP↓ |
| OR-Tools (Cuvelier et al., 2023) | Heuristics | 20.021* | 0.000% | 40.302* | 0.000% | 57.402* | 0.000% | 91.128* | 0.000% |
| *Sequential Decision Methods* | | | | | | | | | |
| AM (Kool et al., 2018) | RL+G | – | 0.173% | – | 0.211% | – | 0.325% | – | – |
| POMO (Kwon et al., 2020) | RL+G | – | 0.130% | – | 0.19% | – | 0.50% | – | 6.41% |
| BQ-NCO (Drakulic et al., 2023) | SL+G | – | – | – | 0.10% | – | 0.14% | – | 0.74% |
| GOAL (Drakulic et al., 2025) | SL+G | – | – | – | 0.12% | – | 1.63% | – | 2.40% |
| CORectifier | RL+G | 20.018 | 0.013% | 40.298 | 0.012% | 57.397 | 0.009% | 91.121 | 0.007% |

similar stability trend. Notably, we avoid setting $p_{batch}$, $p_{inst}$, $\alpha$, or $\beta$ to values smaller than 0.1 in case the model would degenerate into a vanilla RL paradigm. We defer results on MatNet in Appendix F.3.

**Scalability Study.** Table 10 stacks results of commonly compared RL-based solvers on TSP-500. Observable is that CORectifier with greedy decoding achieves substantial improvements, reducing the performance drop by up to **89.8%** compared to diverse RL baselines. While we admit that CORectifier still slightly lags behind SOTA SL methods (e.g., Ma et al. (2025a); Li et al. (2024)) at this scale, our work marks an initial step towards alleviating the "scalability curse" as a lingering bottleneck on graph-based problems with more than 100 nodes in RL4CO practices.

Table 10: Prevalent **RL-based** results on TSP-500, partially quoted from (Li et al., 2023; Ma et al., 2025a; Ye et al., 2024).

| METHOD | DROP↓ |
|---|---|
| AM (Vaswani et al., 2017) | 20.99% |
| POMO+EAS-Emb (Hottung et al., 2021) | 16.25% |
| POMO+EAS-Tab (Hottung et al., 2021) | 48.22% |
| Sym-NCO (Kim et al., 2022) | 27.56% |
| LCP (Kim et al., 2021) | 25.80% |
| EAN+2Opt (Deudon et al., 2018) | 43.57% |
| CORectifier (Greedy) | **4.92%** |

**Diversity Study.** Fig. 6 shows CORectifier method effectively mitigates core limitations of RL4CO with 1) a clearer separation of trajectory advantages for informative sampling, 2) a smoother and more scattered distribution for advantage scales, and 3) substantially higher trajectory entropy, together confirming that our method encourages more diverse exploration. See full plots in Appendix F.6.

**Stability Study.** Following Ma et al. (2025a); Li et al. (2025b), we report the standard deviation of drops (partially in Table 18; full results in Appendix F.4). **Remark 1.** The observed variances stem from the inherent differences in instance difficulty instead of inferential randomness. Rather, the model output remains consistent across repeated runs (free of execution-wise randomness). **Remark 2.** Guaranteed by its design principle, as the quality of expert data decreases, CORectifier is (in the worst case) degraded to and bounded by vanilla RL's performance.

Table 11: Standard deviations of the DROPs.

| TASK | DROP±STD. |
|---|---|
| TSP-50 | 0.132±0.281% |
| TSP-100 | 0.664±0.604% |
| ATSP-50 | 0.585±0.674% |
| ATSP-100 | 2.394±1.024% |
| PCTSP-50 | 1.445±1.328% |
| PCTSP-100 | 3.775±1.964% |

**Adaptability Study. 1) Backbones.** Following Kim et al. (2022), we showcase the adaptability of our learning paradigm to some best-known RL4CO methods: POMO, MatNet, and AM by employing them as the backbone (detailed in Appendix D) in main experiments for TSP/CVRP, ATSP, and PCTSP, respectively. **2) More Tasks.** We apply CORectifier on the Knapsack Problem (KP) to showcase its applicability beyond the routing problems. Results in Table 9 demonstrate overwhelming advantages over previous SOTA. The solving times and further discussions are provided in Appendix F.5.

## 5 CONCLUSION AND OUTLOOK

In this paper, we propose CORectifier, a new NCO solver trained via a novel Rectified RL (RRL) paradigm. By leveraging hierarchically controlled expert guidance during training, RRL enables sample-efficient and optimality-aware learning, immensely advancing scalability and performance with RL4CO. Limitations and broader impacts are discussed in Appendices H and I. Future work will extend assessment to more CO tasks (e.g., orienteering, scheduling, portfolio problems, etc.) and explore the possibility of incorporating more advanced backbones for orthogonal performance gains.

ETHICS STATEMENT

The methods proposed in this paper aim to improve the field of neural combinatorial optimization (NCO). To our best knowledge, no ethical issues or harmful insights of this work need to be otherwise stated. Broader impact of this work is further discussed in Appendix I, and the use of Large Language Models (LLMs) are stated in Appendix J.

REPRODUCIBILITY STATEMENT

The hardware and the preparation of the used data are described in Sec. 4.1. The detailed parameterization and implementation of the models for training and testing are provided in Appendix E. Source code and datasets shall be fully open-sourced upon publication.

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

APPENDIX

CONTENTS

## A    ALGORITHM 1: COMPLETE PIPELINE OF CORECTIFIER

---

**Algorithm 1** CORectifier for Neural Combinatorial Optimization

---

**Input:** Random training data $\mathcal{G}$ of certain distribution; a subset of which, i.e., $\mathcal{G}^* \subset \mathcal{G}$, labeled with $\{\tau_i^*\}_{G_i \in \mathcal{G}^*}$ as reference trajectories; problem-specific objective function $c(\cdot|\cdot)$ and constrained feasible region $\Omega$; batch-wise rectification probability $p_{\text{batch}}$; instance-wise rectification probability $p_{\text{inst}}$ (within a mini-batch); length range for rectified segments $[\alpha, \beta]$ (within an instance); problem size $M$, batch-size $B$, maximum training epochs $N_{\text{ep}}$, number of data samples per epoch $N_s$; learning rate $\eta$; and schedulers for $p_{\text{batch}}, p_{\text{inst}}, \alpha, \beta$.

$-\ -\ -\ -\ -\ -\ -\ -\ -\ -\ -\ -\ -\ -\ -\ -\ -\ -\ -\ -\ -\ -\ -\ -\ -\ -\ -\ -\ -\ -\ -\ -$

Initialize the policy network $\pi_\theta$ with random parameters $\theta$

**while** not done **do** ▷ **Stage 1 (Optional)**: *Pretrain the model with vanilla IL*

    Sample a batch of instances $\{G_i\}_{i=1}^B \sim \mathcal{G}^*$ with expert trajectories $\{\tau_i^*\}$

    Compute imitation loss $\mathcal{L}_{\text{IL}}(\theta) \leftarrow \frac{1}{B} \sum_{i=1}^B \left[ -\mathbb{E}_{\tau \sim \pi_\theta(\cdot|G_i)} \log \pi_\theta(\tau \mid G_i) \right]$

    $\theta \leftarrow \theta - \eta \nabla_\theta \mathcal{L}_{\text{IL}}(\theta)$ ▷ *Mimick expert trajectories until a rough convergence*

**end while**

$-\ -\ -\ -\ -\ -\ -\ -\ -\ -\ -\ -\ -\ -\ -\ -\ -\ -\ -\ -\ -\ -\ -\ -\ -\ -\ -\ -\ -\ -\ -\ -$

**for** epoch $\leftarrow 1$ to $N_{\text{ep}}$ **do** ▷ **Stage 2 (Core)**: *Learning with hierarchical RRL*

    **for** batch index $\leftarrow 1$ to $\lfloor N_s/B \rfloor$ **do**

        **with probability** $p_{\text{batch}}$: Sample batch $\{G_i\}_{i=1}^B \sim \mathcal{G}^*$ with expert trajectories $\{\tau_i^*\}$

        **with probability** $1 - p_{\text{batch}}$: Generate (on the fly) batch $\{G_i\}_{i=1}^B \sim \mathcal{G} \setminus \mathcal{G}^*$ without labels

        Sample $N = \min(M, 200)$ solutions $\mathcal{T}_i \leftarrow \{\tau_i^j\}_{j=1}^N \sim \pi_\theta(\tau \mid G_i)$ for each instance $G_i$

        **if** current batch enables expert guidance **then**

            Select a proportion of trajectories $\widehat{\mathcal{T}}_i \subseteq \mathcal{T}_i$ such that $|\widehat{\mathcal{T}}_i| / |\mathcal{T}_i| = p_{\text{inst}}$ for rectification

            Determine rectifier size $k \leftarrow \lfloor \gamma M \rfloor, \gamma \in [0,1] \sim \text{Uniform}(\alpha, \beta)$

            Generate rectifier mask $\mathcal{M}_{G_i} \in \{0,1\}^{N \times M} \leftarrow \text{Concat}([\mathbf{1}^{N \times k}; \mathbf{0}^{N \times (M-k)}])$ w.r.t. $N$ solutions

            **for** $j \in \{1, \cdots N\} \cap \{n \mid \tau_i^n \in \widehat{\mathcal{T}}_i\}$ **do** ▷ *Start rectification in parallel*

                **for** step $t \leftarrow 2$ to $M$ (assuming $|\tau| = M$ e.g. for TSP) **do**

                    Retrieve optimal next action $a_t^*$ from $\tau_i^*$, given last action taken $a_{t-1} \in \tau_i^j$

                    **if** $\mathcal{M}_{G_i}^{j,t}$ is True and $a_t^*$ does not violate problem constraints **then**

                        $\tau_i^j \leftarrow {\tau'}_i^j$ (by replacing $a_t \in \tau_i^j$ with $a_t^*$) ▷ *Always ensure that $\tau_i^j \in \Omega$*

                  **end if**

                **end for**

            **end for**

        **end if**

        Reward $R_{i,j} \leftarrow -c(\tau_i^j; G_i)$; advantage $A_{i,j} \leftarrow R_{i,j} - \frac{1}{N} \sum_{j'=1}^N R_{i,j'}$;

        Loss $\mathcal{L}_{\text{RL}}(\theta) \leftarrow -A \cdot \mathbb{E}_{\tau \sim \pi_\theta} \log \pi_\theta(\tau \mid G)$

        Compute RL policy gradient $\nabla_\theta \mathcal{L}_{\text{RL}}(\theta) \leftarrow -\frac{1}{BN} \sum_{i=1}^B \sum_{j=1}^N A_{i,j} \nabla_\theta \log \pi_\theta(\tau_i^j \mid G_i)$

        $\theta \leftarrow \theta - \eta \nabla_\theta \mathcal{L}_{\text{RL}}(\theta)$

    **end for**

    $p_{\text{batch}}, p_{\text{inst}}, \alpha, \beta \leftarrow \text{Schedulers}(p_{\text{batch}}, p_{\text{inst}}, \alpha, \beta).\text{step}()$ ▷ *Dynamically scheduling the rectifiers*

**end for**

---

## B    FORMAL PROBLEM DEFINITIONS

**Traveling Salesman Problem (TSP).** Given $G$ and a cost matrix $\mathbf{C} \in \mathbb{R}^{N \times N}$, where $\mathbf{C}_{ij}$ denotes the cost of edge $(i,j)$, the objective is to find a tour $\tau = (i_1, \cdots, i_N)$ that minimizes the total cost: $\sum_{k=1}^{N-1} \mathbf{C}_{i_k i_{k+1}} + \mathbf{C}_{i_N i_1}$.

**Asymmetric Traveling Salesman Problem (ATSP).** ATSP is a variant of TSP where the cost matrix $\mathbf{C}$ is not necessarily symmetric, i.e., $\mathbf{C}_{ij} = \mathbf{C}_{ji}$ does not always hold for all $i, j \in \mathcal{V}$. In this paper, we follow Kwon et al. (2021); Pan et al. (2025b); Ma et al. (2025a) to study the metric ATSP, where the triangle inequality holds: $\mathbf{C}_{ij} + \mathbf{C}_{jk} \geq \mathbf{C}_{ik}$ for any distinct nodes $i, j$, and $k$.

**Prize-Collecting Traveling Salesman Problem (PCTSP).** As defined in Balas (1989), PCTSP extends the classic TSP by associating each node $v \in \mathcal{V}$ with two values: a prize $p_v \in \mathbb{R}_{\geq 0}$ and a penalty $q_v \in \mathbb{R}_{\geq 0}$. Given a cost matrix $\mathbf{C} \in \mathbb{R}^{N \times N}$ (where $\mathbf{C}_{ij}$ denotes the travel cost between node $i$ and node $j$), the objective is to find a tour $\tau = (i_1, i_2, \cdots, i_k)$ (visiting a subset of nodes,

with $k \leq N$) that satisfies a minimum total prize constraint $\sum_{i \in \tau} p_i \geq P_{\min}$ (where $P_{\min} \in \mathbb{R}_{\geq 0}$ is a predefined threshold), while minimizing the total cost: $\sum_{m=1}^{k-1} \mathbf{C}_{i_m i_{m+1}} + \mathbf{C}_{i_k i_1} + \sum_{v \notin \tau} q_v$.

**Capacitated Vehicle Routing Problem (CVRP).** Given $G$, a depot node $v_0 \in \mathcal{V}$, a cost matrix $\mathbf{C} \in \mathbb{R}^{N \times N}$, a demand vector $\mathbf{d} \in \mathbb{R}_+^N$, and a vehicle capacity $Q > 0$, the goal is to plan a set of routes $\mathcal{R}$, each route $r \in \mathcal{R}$ starting and ending at the depot $v_0$, such that each customer node is visited exactly once and the total demand on each route does not exceed $Q$, i.e., $\sum_{i \in r} \mathbf{d}_i \leq Q$. The objective is to minimize the total cost of all routes: $\min_{\mathcal{R}} \sum_{r \in \mathcal{R}} \sum_{(i,j) \in r} \mathbf{C}_{ij}$.

**Knapsack Problem (KP).** Given a set of items $\mathcal{I} = \{1, 2, \cdots, M\}$, where each item $i \in \mathcal{I}$ is associated with two attributes: a weight $w_i \in \mathbb{R}_+$ and a value $v_i \in \mathbb{R}_+$, and a knapsack with a maximum weight capacity $W \in \mathbb{R}_+$, the objective is to select a subset $\mathcal{S} \subseteq \mathcal{I}$ of items such that the total weight of the selected items does not exceed the knapsack capacity, i.e., $\sum_{i \in \mathcal{S}} w_i \leq W$, while maximizing the total value of the selected items: $\max_{\mathcal{S} \subseteq \mathcal{I}} \sum_{i \in \mathcal{S}} v_i$. In this work, we focus exclusively on the **0-1 Knapsack Problem** that aligns with the settings in (Drakulic et al., 2025; 2023; Kwon et al., 2020), where each item can be optionally included in the subset $\mathcal{S}$ only once.

## C SUPPLEMENTARY CLARIFICATION OF THE RECTIFICATION OPERATOR $\mathcal{R}$

### C.1 IMPLEMENTATION DETAILS

**Definition and practical implementation.** For mathematical clarity, the rectification operator $\mathcal{R}(\tau, \tau_G^*, \mathcal{M})$ is defined in the main text from a *global* perspective, i.e., as if the entire policy-sampled trajectory $\tau$ and expert trajectory $\tau_G^*$ were both available beforehand, and a masked replacement is applied afterward. This description is intended to make the concept of "replacing segments of a trajectory" transparent at a macroscopic level.

In practice, however, the rectification procedure is implemented *locally and autoregressively at each step*, rather than by first sampling a full trajectory and then post-hoc modifying it. More precisely, a rectified trajectory $\tau' = \mathcal{R}(\tau, \tau_G^*, \mathcal{M})$ is constructed on the fly as follows. At decision step $t$, given the current partial trajectory $\tau'_{1:t-1}$, the policy network produces a distribution over feasible next actions:

$$a_t \sim \pi_\theta(\cdot \mid s_t), \quad \text{where } s_t = s(\tau'_{1:t-1}). \tag{14}$$

If the rectification mask satisfies $\mathcal{M}[j, t] = 1$, and the expert successor $a_t^*$ in $\tau_G^*$ is feasible under the current state (i.e., not violating task constraints), then the expert action is taken as in Eq. 10:

$$a_t \leftarrow a_t^*. \tag{15}$$

Otherwise, the model prediction $a_t$ sampled from $\pi_\theta$ is adopted. The chosen action is appended to the partial trajectory and the environment transitions to the next state. The above procedure repeats until the trajectory is completed, resulting in a fully valid and feasible auto-regressive rollout.

**Why does the suffix $\tau_{t+k+1:M}$ remain feasible?** The implementation naturally resolves this question as rectification is performed *before* the next state is computed, the environment always observes the rectified prefix $\tau'_{1:t+k}$. Consequently, the subsequent actions $\tau'_{t+k+1:M}$ are generated:

$$a_{t+k+1} \sim \pi_\theta(\cdot \mid s_{t+k+1}), \tag{16}$$

where $s_{t+k+1}$ is the correct state resulting from the rectified actions. Thus, the suffix is *not* inherited from a trajectory generated under a different state; instead, it is recomputed autoregressively under consistent feasibility masks (e.g., visited-node masks in TSP, capacity-feasibility masks in CVRP, etc.). This guarantees that the entire rectified trajectory remains feasible.

**An example code for TSP.** Below we show a simplified PyTorch-style implementation for the TSP case, consistent with the notation in the paper, where $B$ is the batch size, $N$ is the number of parallel samples per instance, and $M$ is the problem size (number of nodes):

```
1  class TSPTrainer:
2      """
3      Skipping other properties and methods ...
4      """
```

```
5       def _train_one_batch(self):
6           # data preparations ... skipped
7
8           # randomly decide the number of rectified actions
9           rectify_level = int(M * random.uniform(alpha, beta))
10          # construct global mask as in Eq. (9): shape (B * N, M)
11          global_action_mask = torch.zeros(B * N, M).bool()
12          global_action_mask[:, :rectify_level] = True
13          global_action_mask.roll(random.randint(0, M - 1), dims=1)
14          # construct instance mask with prob. p_inst: shape (B * N,)
15          instance_mask = torch.rand(B * N) < p_inst if enable_rectify
        else None
16
17          while not done:
18              # extract intra-instance level mask at current step
19              action_mask = global_action_mask[:,
        self.env.selected_count].reshape(-1)
20              # forward pass (rectification happens inside the model)
21              action, prob = self.model.forward(state, instance_mask,
        action_mask)
22              # environment transition and reward computation
23              state, reward, done = self.env.step(action)
24              # record probabilities for policy gradient
25              prob_list = torch.cat((prob_list, prob[:, :, None]),
        dim=2)
26
27          # backpropagation ... skipped
```

Inside the model, the rectification behavior is implemented as follows:

```
1   class TSPModel(nn.Module):
2       """
3       Skipping other properties and methods ...
4       """
5       def forward(self, state, instance_mask=None, action_mask=None):
6           # load reference expert trajectories
7           ref_tours = state.ref_tours
8           if state.current_node is None:
9               # first step: same as POMO
10              selected = torch.arange(N)[None, :].expand(B, N)
11              prob = torch.ones(size=(B, N))
12          else:
13              # decode probabilities over feasible next nodes: (B, N, M)
14              node_embed = _get_encoding(state.current_node)
15              probs = self.decoder(
16                  node_embed, ninf_mask=state.ninf_mask
17              )
18              if self.training:
19                  # sample initial next action from current policy
20                  init_action = probs.reshape(B * N, -1).multinomial(1)
21                  init_action = init_action.squeeze(1).reshape(B, N)
22                  if instance_mask is not None:
23                      # prepare tensors for rectification
24                      init_action = init_action.reshape(B * N)
25                      cur_node = state.current_node.reshape(B * N)
26                      unvisited = (state.ninf_mask==0).reshape(B*N, -1)
27                      ref_tours = ref_tours.expand(B,N,M).reshape(B*N,M)
28                      # rectified action at current step
29                      rectified_action = _rectify_action(
30                          cur_node, init_action,
31                          instance_mask, action_mask,
32                          ref_tours, unvisited
33                      ).reshape(B, N)
34                      prob = _get_action_prob(probs, rectified_action)
35                  else:
```

```
36                        rectified_action = init_action
37          return rectified_action, prob
38
39  def _rectify_action(
40      cur_node,         # shape: (B * N,)
41      init_action,      # shape: (B * N,)
42      instance_mask,    # shape: (B * N,)
43      action_mask,      # shape: (B * N,)
44      ref_tour,         # shape: (B * N, M)
45      unvisited         # shape: (B * N, M)
46  ):
47      # position of current node in the reference tour
48      cur_node_idx = _get_idx_by_node(ref_tour, cur_node)
49      # next position and next node in the reference tour
50      ref_next_idx = (cur_node_idx + 1) % M
51      ref_next_node = _get_node_by_idx(ref_tour, ref_next_idx)
52      # feasibility check: reference next node must be unvisited
53      unvisited_mask = unvisited[torch.arange(B * N), ref_next_node]
54      # candidate rectified action (respecting feasibility)
55      rectified_action = torch.where(unvisited_mask, ref_next_node,
            init_action)
56      # apply instance- and action-level rectification masks
57      rectified_action = torch.where(action_mask & instance_mask,
                                       rectified_action, init_action)
59      return rectified_action
```

This code clarifies that feasibility is preserved automatically, since every action (rectified or not) is validated against the current environment state and all subsequent decisions are sampled conditioned on this updated state.

## C.2    ADAPTED RECTIFICATION OPERATION FOR CVRP AND KP

In Sec. 3.2.1, we have delineated the process of our proposed tri-level rectification mechanism, with TSP (and its variants) used for illustration. Here, we elaborate on how this scheme is adapted to new tasks, namely CVRP and KP. Notably, the first two levels of rectifiers, i.e., batch-level and instance-level, shall remain identical to those employed in training TSP models. However, slight differences emerge in the third step (intra-instance-level rectification) when applied to CVRP and KP.

Specifically, for CVRP, we take into account a key characteristic of its optimal solutions: the overall tour can be decomposed into multiple sub-tours, and only specific client nodes serve as the first-hop nodes from the depot in the optimal solution. Accordingly, we restrict the intra-instance-level rectification for CVRP to the selection of optimal sub-trajectories, i.e., the segment of a route from a vehicle's departure from the depot (usually indexed as 0) to its return. This implies that the decision whether to integrate an optimal sub-tour for the subsequent actions (with probability $p_{\text{sub}}$) is only made when the agent is positioned at the depot and remains until the next time it returns. A mathematical illustration is as follows,

$$\mathcal{M}_{G_i} = \begin{pmatrix} 1\cdots1, 0\cdots0, \cdots, 1\cdots1 \\ 1\cdots1, 0\cdots0, \cdots, 0\cdots0 \\ \cdots \\ 0\cdots0, 1\cdots1, \cdots, 1\cdots1 \end{pmatrix}_{N \times M}, \qquad (17)$$

where consecutive ones denote the sub-trajectories selected for rectification.

For KP, by contrast, item selection is unordered. Since the intra-instance rectifier was originally designed to better capture sequential relationships within local regions for routing problems (which is a feature irrelevant to KP), we disable the intra-instance rectification operation when training KP models. Instead, when a sampled trajectory is masked as `True` for instance-level rectification, the entire trajectory is replaced with a randomly permuted trajectory that includes all items from the optimal reference solution.

# D ADAPTED MODEL ARCHITECTURES

As described in Sec. 4.3, we employ AM Kool et al. (2018), POMO Kwon et al. (2020), and MatNet Kwon et al. (2021) as the neural backbones for instantiating CORectifier paradigm to learn PCTSP, TSP, and ATSP, respectively, which simultaneously validates the solving performance as well as the wide applicability of our approach by adapting it to popular existing RL-based learning frameworks. Beyond the TSP family, we further develop models upon POMO to conduct the rectified RL learning on more complex tasks, i.e., CVRP and KP. In this section, we provide a concise introduction to the layer-level neural architecture for the corresponding backbone models for your convenient reference.

## D.1 ATTENTION MODEL (AM)

**Encoder.** The encoder converts node features into embedding vectors containing global information. It adopts a multi-layer attention structure similar to the classical Transformer Vaswani et al. (2017), with no positional encoding to ensure input order invariance.

Specifically, the initial embedding $h_i^{(0)}$ of node $i$ is obtained via linear projection:

$$h_i^{(0)} = W^x x_i + b^x, \tag{18}$$

where $W^x \in \mathbb{R}^{d_h \times d_x}$ and $b^x \in \mathbb{R}^{d_h}$ are learnable parameters, and $d_h = 128$ (embedding dimension).

The encoder consists of $N = 3$ attention layers. Each layer comprises a Multi-Head Attention (MHA) sublayer and a node-wise Feed-Forward (FF) sublayer, with residual connections and Batch Normalization (BN) added to each sublayer. The update formula for the $\ell$-th layer is:

$$\hat{h}_i = \text{BN}^\ell \left( h_i^{(\ell-1)} + \text{MHA}_i^\ell \left( h_1^{(\ell-1)}, ..., h_n^{(\ell-1)} \right) \right), \tag{19}$$

$$h_i^{(\ell)} = \text{BN}^\ell \left( \hat{h}_i + \text{FF}^\ell(\hat{h}_i) \right). \tag{20}$$

The MHA uses $M = 8$ attention heads, each with dimension $d_k = d_h/M = 16$. Compatibility between nodes is calculated via dot-product, and information is aggregated by weighted summation. The FF block is a single hidden layer (dimension 512) with ReLU activation, formulated as $\text{FF}(\hat{h}_i) = W^{ff,1} \cdot \text{ReLU}(W^{ff,0}\hat{h}_i + b^{ff,0}) + b^{ff,1}$. The encoder ultimately outputs two types of embeddings: 1) node embeddings: $h_i^{(N)}$ (node-level embeddings from the $N$-th layer), and 2) graph embedding: $\overline{h}^{(N)}$ (mean of all node embeddings, containing global information):

$$\overline{h}^{(N)} = \frac{1}{n} \sum_{i=1}^{n} h_i^{(N)}. \tag{21}$$

**Decoder.** The decoder generates the visitation sequence step-by-step. It introduces a "context node" $(c)$ to efficiently aggregate encoder information and partial sequence states, with two core attention computations (glimpse layer and output probability layer). The embedding $h_{(c)}^{(N)}$ of the context node integrates the graph embedding, and information about the "first node" and "last node" of the partial sequence (to adapt to tasks like TSP that require returning to the starting point). Its formulation is:

$$h_{(c)}^{(N)} = \begin{cases} [\overline{h}^{(N)}, h_{\pi_{t-1}}^{(N)}, h_{\pi_1}^{(N)}] & t > 1 \\ [\overline{h}^{(N)}, v^l, v^f] & t = 1 \end{cases}. \tag{22}$$

For $t = 1$ (initial step), learnable placeholders $v^l, v^f \in \mathbb{R}^{d_h}$ are used; $\pi_{t-1}$ is the node visited in the previous step; and $[\cdot, \cdot, \cdot]$ denotes horizontal concatenation (resulting in a dimension of $3d_h$).

The context node embedding $h_{(c)}^{(N+1)}$ is updated via MHA, with attention only computed between the context node and all other nodes (to improve efficiency):

First, compute queries (from the context node), keys, and values (from encoder node embeddings):

$$q_{(c)} = W^Q h_{(c)}, \quad k_i = W^K h_i^{(N)}, \quad v_i = W^V h_i^{(N)}, \tag{23}$$

where $W^Q, W^K \in \mathbb{R}^{Md_k \times 3d_h}$ and $W^V \in \mathbb{R}^{Md_v \times d_h}$ are parameters.

Second, compatibility calculation and masking (to block already visited nodes):

$$u_{(c)j} = \begin{cases} \frac{q_{(c)}^T k_j}{\sqrt{d_k}} & j \notin \{\pi_1, ..., \pi_{t-1}\} \\ -\infty & \text{otherwise} \end{cases}. \tag{24}$$

Third, weighted aggregation to obtain $h_{(c)}^{(N+1)}$ (following the multi-head fusion logic of MHA). A single-head attention layer converts $h_{(c)}^{(N+1)}$ into the probability of the next node to visit. First, a compatibility clipping is used to avoid numerical overflow:

$$u_{(c)j} = \begin{cases} C \cdot \tanh\left(\frac{q_{(c)}^T k_j}{\sqrt{d_h}}\right) & j \notin \{\pi_1, ..., \pi_{t-1}\} \\ -\infty & \text{otherwise} \end{cases}, \tag{25}$$

where $C = 10$ is the clipping threshold. Finally, softmax normalization is computed to get probabilities:

$$p_\theta(\pi_t = i | s, \pi_{1:t-1}) = \frac{e^{u_{(c)i}}}{\sum_j e^{u_{(c)j}}}. \tag{26}$$

### D.2 POLICY OPTIMIZATION WITH MULTIPLE OPTIMA (POMO)

POMO (Kwon et al., 2020) adopts the core encoder architecture from AM (Kool et al., 2018), while extending AM's training paradigm by introducing symmetric parallel trajectories to improve baseline estimation and solution space exploration.

Specifically, POMO abandons AM's trainable start token and instead generates $N$ parallel trajectories (one per node as the initial starting point, e.g., 100 trajectories for TSP-100). This enables symmetric exploration of the solution space by covering all possible initial nodes for each instance. For trajectory $i$ (starting at node $a_1^i$), the initial decoder state is directly set to the encoder's embedding of $a_1^i$:

$$h_{\pi_1^i} = h_{a_1^i}, \tag{27}$$

eliminating the need for AM's trainable start token parameters $v^l, v^f$.

For $t > 1$, the decoder's context embedding for trajectory $i$ follows AM's design, fusing three components: the global graph embedding $\bar{h}^{(N)}$, the embedding of the last visited node $h_{\pi_{t-1}^i}$, and the embedding of the trajectory's initial node $h_{\pi_1^i}$:

$$h_{(c)}^i = \left[\bar{h}^{(N)}, h_{\pi_{t-1}^i}, h_{\pi_1^i}\right]. \tag{28}$$

To reduce computational overhead, $N$ context embeddings are stacked into a single matrix, and AM's attention mechanism is leveraged to compute all trajectories in parallel, avoiding the inefficiency of sequential trajectory generation.

### D.3 MATRIX ENCODING NETWORK (MATNET)

The highlight of MatNet Kwon et al. (2021) lies in the fact that it pioneers the applicability of RL-based models to learn the more complex asymmetric TSP (ATSP) without 2D-coordinates as input node features, which designs a mixed-score attention module to incorporate information from the distance matrix. As for training, MatNet follows POMO to utilize an average reward from $N$ different samples as the REINFORCE baseline.

**Encoder.** The encoder models node relationships using two parallel embedding matrices ($\mathbf{A}^l, \mathbf{B}^l \in \mathbb{R}^{d \times N}$ for the $l$-th layer, where $d$ denotes embedding dimension and $N$ denotes node count) and a mixed-score attention mechanism that fuses cost matrix information:

Embedding Initialization: $\mathbf{A}^0$ is initialized as a zero embedding matrix, while $\mathbf{B}^0$ is initialized as a one-hot embedding matrix. To ensure $\mathbf{B}^0$ matches dimension $d$, MatNet enforces $d \geq N$ and pads a

square one-hot matrix $\mathbf{OE} \in \mathbb{R}^{N \times N}$ with zeros, i.e., $\mathbf{B}^0 = [\mathbf{OE}, \mathbf{0}]^\top$ (where $\mathbf{0} \in \mathbb{R}^{N \times (d-N)}$). For MatNetPOE, $\mathbf{B}^0$ uses a position-aware one-hot embedding ($\mathbf{POE}$) to avoid padding.

Mixed-Score Attention: Taking $\mathbf{A}$-embedding update as an example, query, key, and value matrices are computed as:

$$\mathbf{Q}_a^l = \mathbf{W}_a^Q \mathbf{A}^l, \quad \mathbf{K}_a^l = \mathbf{W}_a^K \mathbf{B}^l, \quad \mathbf{V}_a^l = \mathbf{W}_a^V \mathbf{B}^l, \tag{29}$$

where $\mathbf{W}_a^Q, \mathbf{W}_a^K, \mathbf{W}_a^V \in \mathbb{R}^{d_{qkv} \times d}$ are learnable attention parameters. A $2 \rightarrow 1$-dimensional MLP (MLP$_1$) concatenates and mixes the cost matrix $\mathbf{C}$ with scaled dot-product attention scores ($\mathbf{Q}_a^{l^\top} \mathbf{K}_a^l / \sqrt{d_{qkv}}$), generating the mixed attention score matrix:

$$\mathbf{MixedScoreAtt}_a^l = \mathrm{softmax}\left(\mathrm{MLP}_1\left(\left[\mathbf{C}; \frac{\mathbf{Q}_a^{l^\top} \mathbf{K}_a^l}{\sqrt{d_{qkv}}}\right]\right)\right) \in \mathbb{R}^{N \times N}. \tag{30}$$

A $d$-dimensional MLP (MLP$_2$, input/output dimension $d$) then updates $\mathbf{A}^{l+1}$:

$$\mathbf{A}^{l+1} = \mathrm{MLP}_2\left(\mathbf{MixedScoreAtt}_a^l \mathbf{V}_a^{l^\top}\right)^\top. \tag{31}$$

Symmetric Update for $\mathbf{B}$: $\mathbf{B}^{l+1}$ is computed symmetrically by swapping $\mathbf{A}^l$ and $\mathbf{B}^l$ and using separate parameters ($\mathbf{W}_b^Q, \mathbf{W}_b^K, \mathbf{W}_b^V$). The encoder extends this mechanism to multi-head attention (following the original Transformer (Vaswani et al., 2017)) for richer node representations.

**Decoder.** The decoder performs "rollout" to generate a tour $\tau = \{i_1, i_2, \cdots, i_N\}$ using the encoder's final $\mathbf{A}, \mathbf{B}$ embeddings:

Key/Value Initialization: Decoder key and value matrices are precomputed as $\mathbf{K}_{dec} = \mathbf{W}_{dec}^K \mathbf{B}$ and $\mathbf{V}_{dec} = \mathbf{W}_{dec}^V \mathbf{B}$, where $\mathbf{W}_{dec}^K, \mathbf{W}_{dec}^V \in \mathbb{R}^{d_{qkv} \times d}$ are trainable parameters.

Query Update: The query vector accumulates information of previously selected nodes. For the first node $i_1$, $\mathbf{q}_{dec}^1 = \mathbf{W}_{dec}^{Q^1} \mathbf{a}_{i_1}$; for subsequent nodes, it is updated as:

$$\mathbf{q}_{dec}^{n+1} = \mathbf{W}_{dec}^{Q^0} \mathbf{a}_{i_n} + \mathbf{q}_{dec}^1, \tag{32}$$

where $\mathbf{W}_{dec}^{Q^1}, \mathbf{W}_{dec}^{Q^0} \in \mathbb{R}^{d_{qkv} \times d}$ are learnable parameters.

Next Node Selection: A masking vector $\mathbf{Inf}_{i_{n'} \leq n}$ (elements corresponding to visited nodes $i_{n'}$ set to $-\infty$, others 0) is used to avoid revisiting nodes. The output embedding of the $n$-th rollout iteration is:

$$\mathbf{o}_{dec}^{n+1} = \mathrm{Linear}\left(\mathrm{softmax}\left(\mathbf{Inf}_{i_{n'} \leq n} + \frac{\mathbf{q}_{dec}^{n^\top} \mathbf{K}_{dec}}{\sqrt{d_{qkv}}}\right) \mathbf{V}_{dec}\right), \tag{33}$$

where $\mathrm{Linear}(\cdot)$ denotes a linear layer (input dimension $d_{qkv}$, output dimension $d$). The probability of selecting the next node $i_{n+1}$ is:

$$\mathbf{p}_\theta(i_{n+1}|i_{n'} \leq n) = \mathrm{softmax}\left(\mathbf{Inf}_{i_{n'} \leq n} + \tanh\left(\frac{\mathbf{B}^\top \mathbf{o}_{dec}^{n+1}}{\sqrt{d}}\right)\right). \tag{34}$$

Multi-head attention enhances this process in practice; a complete tour is obtained after $N$ rollout iterations.

# E    DETAILED EXPERIMENTAL SETTINGS

In this section, we provide the hyper-parameters and detailed training and solving settings of our method in Table 12 and Table 13. The hyper-parameters w.r.t. the backbones are in consistency with established implementations from Vaswani et al. (2017); Kwon et al. (2020; 2021); Pan et al. (2025b). Additionally, we use Adam Kingma & Ba (2015) optimizer with a unified weight decay of 1e-6 for all tasks. We set the period ($T_{\max}$) for the independent cosine schedulers for the rectifier-related probabilities (introduced in the last paragraph of Sec. 3.2.1) as 10, 20, 40, 40, 40 for $p_{\text{batch}}$, $p_{\text{inst}}$, $\alpha$, $\beta$, $p_{\text{sub}}$ where applicable, respectively. When training POMO Kwon et al. (2020) for TSP and CVRP, we adopt a parallel sampling size (pomo_size) as $\min(200, M)$ with $M$ being the number of nodes of the graph instances. Note that we have empirically justified the selection of some core parameters in Sec. 4.3 and Appendix F.3.

Table 12: Hyper-parameters and training settings of CORectifier on TSP/ATSP/PCTSP.

| | TSP | | | ATSP | | | | PCTSP | | |
|---|---|---|---|---|---|---|---|---|---|---|
| | 50 | 100 | 500 | 50 | 100 | 200 | 500 | 20 | 50 | 100 |
| Number of pre-training epochs (IL) | 200 | 100 | 100 | 200 | 100 | 100 | 100 | 200 | 200 | 200 |
| Number of training epochs (RRL) | 2000 | 10000 | 1000 | 8000 | 3000 | 3000 | 2000 | 10000 | 10000 | 10000 |
| Samples per epoch | 10000 | 10000 | 10000 | 10000 | 10000 | 10000 | 10000 | 6400 | 6400 | 6400 |
| Learning rate $\eta$ | 2e-4 | 1e-4 | 1e-4 | 4e-4 | 2e-4 | 2e-4 | 5e-5 | 1e-4 | 1e-4 | 1e-4 |
| Training batch-size $B$ | 200 | 64 | 2 | 200 | 64 | 8 | 2 | 64 | 64 | 64 |
| Max Batch-wise rectify probability $p_{batch}$ | 0.1 | 0.1 | 0.1 | 0.5 | 0.5 | 0.5 | 0.5 | 0.1 | 0.1 | 0.1 |
| Max Instance-wise rectify probability $p_{inst}$ | 0.1 | 0.1 | 0.1 | 0.5 | 0.5 | 0.5 | 0.5 | 0.1 | 0.1 | 0.1 |
| Max Intra-instance rectify length ratio $\alpha$ | 0.1 | 0.1 | 0.1 | 0.8 | 0.8 | 0.8 | 0.8 | 0.1 | 0.1 | 0.1 |
| Max Intra-instance rectify length ratio $\beta$ | 0.2 | 0.2 | 0.2 | 1.0 | 1.0 | 1.0 | 1.0 | 0.2 | 0.2 | 0.2 |

Table 13: Hyper-parameters and training settings of CORectifier on CVRP and KP.

| | CVRP | | | | KP | | | |
|---|---|---|---|---|---|---|---|---|
| | 50 | 100 | 200 | 500 | 50 | 100 | 200 | 500 |
| Number of pre-training epochs (pretrain) | 1000 | 1000 | 500 | 100 | 1000 | 1000 | 500 | 100 |
| Number of training epochs (RRL) | 12000 | 12000 | 8000 | 1000 | 5000 | 5000 | 1000 | 1000 |
| Samples per epoch | 10000 | 10000 | 10000 | 10000 | 10000 | 10000 | 10000 | 10000 |
| Learning rate $\eta$ | 2e-4 | 2e-4 | 1e-4 | 1e-4 | 1e-4 | 1e-4 | 5e-5 | 5e-5 |
| Training batch-size $B$ | 200 | 128 | 16 | 4 | 200 | 200 | 64 | 16 |
| Max Batch-wise probability $p_{batch}$ | 0.1 | 0.1 | 0.1 | 0.1 | 0.1 | 0.1 | 0.1 | 0.1 |
| Max Instance-wise probability $p_{inst}$ | 0.1 | 0.1 | 0.1 | 0.1 | 0.1 | 0.1 | 0.1 | 0.1 |
| Max Intra-instance sub-tour probability $p_{sub}$ | 0.1 | 0.1 | 0.1 | 0.1 | N/A | N/A | N/A | N/A |

Table 14: Per-instance optimality drop for methods trained on uniform TSP-100 and evaluated on TSPLIB instances with 50-200 nodes (29 instances). The comparison partially includes baselines in Table 3 in Hudson et al. (2022). Our method is evaluated with greedy decoding with $N_A = 128$ and 2OPT searching enabled. We normalize the 2D coordinates for consistent comparison following Ma et al. (2025a).

| INSTANCES | AM (Kool et al., 2018) | POMO (Kwon et al., 2020) | GCN (Joshi et al., 2019) | Learn2OPT (da Costa et al., 2020) | GNNGLS (Hudson et al., 2022) | Ours |
|---|---|---|---|---|---|---|
| eil51 | 16.767% | 0.719% | 40.025% | 1.725% | 1.529% | 0.057% |
| berlin52 | 4.169% | 0.004% | 33.225% | 0.449% | 0.142% | 0.004% |
| st70 | 1.737% | 0.013% | 24.785% | 0.040% | 0.764% | 0.000% |
| eil76 | 1.992% | 0.004% | 27.411% | 0.096% | 0.163% | 0.613% |
| pr76 | 0.816% | 0.000% | 27.793% | 1.228% | 0.039% | 0.000% |
| rat99 | 2.645% | 6.469% | 17.633% | 0.123% | 0.550% | 1.581% |
| kroA100 | 4.017% | 3.247% | 28.828% | 18.313% | 0.728% | 0.412% |
| kroB100 | 5.142% | 2.817% | 34.686% | 1.119% | 0.147% | 0.912% |
| kroC100 | 0.972% | 1.377% | 35.506% | 0.349% | 1.571% | 1.417% |
| kroD100 | 2.717% | 1.995% | 38.018% | 0.866% | 0.572% | 1.361% |
| kroE100 | 1.470% | 3.088% | 26.589% | 1.832% | 1.216% | 0.811% |
| rd100 | 3.407% | 0.000% | 50.432% | 1.725% | 0.003% | 0.000% |
| eil101 | 2.994% | 0.060% | 26.701% | 0.387% | 1.529% | 1.069% |
| lin105 | 1.739% | 1.393% | 34.902% | 1.867% | 0.606% | 0.756% |
| pr107 | 3.933% | 1.344% | 80.564% | 0.898% | 0.439% | 0.504% |
| pr124 | 3.677% | 1.052% | 70.146% | 10.322% | 0.755% | 0.782% |
| bier127 | 5.908% | 6.009% | 45.561% | 3.044% | 1.948% | 1.387% |
| ch130 | 3.182% | 0.663% | 39.090% | 0.709% | 3.519% | 0.854% |
| pr136 | 5.064% | 1.040% | 58.673% | 0.000% | 3.387% | 1.252% |
| pr144 | 7.641% | 1.129% | 55.837% | 1.526% | 3.581% | 0.504% |
| ch150 | 4.584% | 0.546% | 49.743% | 0.312% | 2.113% | 0.846% |
| kroA150 | 3.784% | 3.034% | 45.411% | 0.724% | 2.984% | 1.118% |
| kroB150 | 2.437% | 2.968% | 56.745% | 0.886% | 3.258% | 1.469% |
| pr152 | 7.494% | 2.534% | 33.925% | 0.029% | 3.119% | 0.295% |
| u159 | 7.551% | 1.245% | 38.338% | 0.054% | 1.020% | 1.081% |
| rat195 | 6.893% | 9.623% | 24.968% | 0.743% | 1.666% | 4.923% |
| d198 | 373.020% | 32.555% | 62.351% | 0.522% | 4.772% | 2.461% |
| kroA200 | 7.106% | 3.563% | 40.885% | 1.441% | 2.029% | 1.806% |
| kroB200 | 8.541% | 4.372% | 43.643% | 2.064% | 2.589% | 2.096% |
| Mean | 16.767% | 3.202% | 40.025% | 1.725% | 1.529% | **1.049%** |

# F  SUPPLEMENTARY RESULTS

## F.1  EXTENDED RESULTS ON TSPLIB

Following Li et al. (2023); Ma et al. (2025a); Hudson et al. (2022); Kim et al. (2022), we present the per-instance solving result (optimality drop) on the real-world TSPLIB dataset (zero-shot generalization) in Table 14, comparing CORectifier with the most commonly quoted results as well as former RL-based approaches. The CORectifier model evaluated is trained on uniform TSP instances with 100 nodes.

Table 15: Per-instance comparison with SOTA generalization results from the latest ML4CO Benchmark (Ma et al., 2025b) on 70 CVRPLIB-Set-X instances with 100–512 client nodes.

| INSTANCE | OPTIMAL | ML4CO-Bench (Best) | | CORectifier | |
|---|---|---|---|---|---|
| | | OBJ.↓ | DROP↓ | OBJ.↓ | DROP↓ |
| X-n101-k25 | 27.905 | 30.563 | 9.524% | 29.465 | 5.588% |
| X-n106-k14 | 26.655 | 27.105 | 1.668% | 27.074 | 1.570% |
| X-n110-k13 | 14.979 | 15.097 | 0.784% | 14.982 | 0.014% |
| X-n115-k10 | 12.933 | 13.574 | 4.957% | 13.536 | 4.663% |
| X-n120-k6 | 13.437 | 13.611 | 1.296% | 13.857 | 2.089% |
| X-n125-k30 | 56.164 | 59.041 | 5.122% | 60.901 | 5.254% |
| X-n129-k18 | 29.005 | 29.356 | 1.211% | 29.454 | 0.734% |
| X-n134-k13 | 11.351 | 11.633 | 2.479% | 11.547 | 1.729% |
| X-n139-k10 | 13.678 | 13.804 | 0.920% | 13.835 | 1.151% |
| X-n143-k7 | 15.776 | 16.533 | 4.798% | 15.842 | 0.421% |
| X-n148-k46 | 43.671 | 45.190 | 3.478% | 44.615 | 2.161% |
| X-n153-k22 | 21.572 | 28.809 | 33.547% | 23.920 | 10.882% |
| X-n157-k13 | 17.572 | 17.708 | 0.775% | 17.702 | 0.740% |
| X-n162-k11 | 14.412 | 14.705 | 2.029% | 14.583 | 1.183% |
| X-n167-k10 | 20.766 | 21.244 | 2.303% | 21.032 | 1.177% |
| X-n172-k51 | 46.974 | 59.677 | 27.041% | 49.557 | 5.497% |
| X-n176-k26 | 47.816 | 51.999 | 8.748% | 53.496 | 11.767% |
| X-n181-k23 | 26.809 | 27.067 | 0.963% | 27.179 | 1.381% |
| X-n186-k15 | 24.251 | 24.720 | 1.933% | 24.886 | 2.618% |
| X-n190-k8 | 17.037 | 17.418 | 2.238% | 17.496 | 2.386% |
| X-n195-k51 | 44.274 | 45.772 | 3.385% | 47.080 | 6.339% |
| X-n200-k36 | 60.084 | 61.834 | 2.914% | 63.205 | 5.196% |
| X-n204-k19 | 19.588 | 19.967 | 1.936% | 20.029 | 2.250% |
| X-n209-k16 | 30.691 | 31.107 | 1.355% | 31.484 | 2.582% |
| X-n214-k11 | 10.879 | 11.278 | 3.665% | 11.159 | 2.572% |
| X-n219-k73 | 117.601 | 124.696 | 6.033% | 124.616 | 5.435% |
| X-n223-k34 | 40.478 | 41.308 | 2.051% | 41.429 | 2.347% |
| X-n228-k23 | 25.747 | 26.867 | 4.350% | 27.699 | 7.581% |
| X-n233-k16 | 19.297 | 19.731 | 2.249% | 20.189 | 4.623% |
| X-n237-k14 | 27.078 | 27.397 | 1.178% | 27.742 | 1.940% |
| X-n242-k48 | 82.757 | 84.427 | 2.018% | 85.339 | 2.501% |
| X-n247-k50 | 37.966 | 55.808 | 46.994% | 41.499 | 9.305% |
| X-n251-k28 | 38.885 | 39.798 | 2.348% | 39.734 | 2.185% |
| X-n256-k16 | 18.942 | 19.379 | 2.306% | 19.483 | 2.854% |
| X-n261-k13 | 26.591 | 27.586 | 3.741% | 27.323 | 2.030% |
| X-n266-k58 | 75.489 | 78.007 | 3.336% | 81.174 | 7.530% |
| X-n270-k35 | 35.432 | 36.133 | 1.978% | 36.421 | 2.789% |
| X-n275-k28 | 22.924 | 23.488 | 2.464% | 23.433 | 2.220% |
| X-n280-k17 | 33.503 | 34.597 | 3.265% | 34.462 | 2.758% |
| X-n284-k15 | 20.421 | 21.252 | 4.071% | 21.026 | 2.963% |
| X-n289-k60 | 95.35 | 103.478 | 8.523% | 101.154 | 6.086% |
| X-n294-k50 | 47.315 | 48.783 | 3.102% | 51.678 | 9.221% |
| X-n298-k31 | 34.247 | 35.464 | 3.553% | 35.614 | 3.993% |
| X-n303-k21 | 21.961 | 22.659 | 3.179% | 22.491 | 2.467% |
| X-n308-k13 | 25.980 | 26.723 | 2.859% | 28.104 | 8.173% |
| X-n313-k71 | 94.437 | 103.575 | 9.676% | 100.691 | 6.633% |
| X-n317-k53 | 78.456 | 86.456 | 10.198% | 88.001 | 12.055% |
| X-n322-k28 | 29.846 | 30.590 | 2.492% | 30.886 | 3.483% |
| X-n327-k20 | 27.681 | 28.546 | 3.127% | 28.762 | 3.904% |
| X-n331-k15 | 31.155 | 31.993 | 2.690% | 31.783 | 2.021% |
| X-n336-k84 | 139.283 | 157.244 | 12.896% | 148.214 | 6.113% |
| X-n344-k43 | 42.194 | 43.225 | 2.442% | 43.560 | 3.243% |
| X-n351-k40 | 25.948 | 27.200 | 4.822% | 28.216 | 8.919% |
| X-n359-k29 | 51.512 | 52.817 | 2.535% | 53.122 | 3.023% |
| X-n367-k17 | 24.741 | 25.711 | 3.920% | 25.324 | 2.354% |
| X-n376-k94 | 147.881 | 148.786 | 0.612% | 149.029 | 0.777% |
| X-n384-k52 | 66.086 | 67.689 | 2.426% | 70.786 | 7.123% |
| X-n393-k38 | 38.386 | 39.702 | 3.427% | 39.847 | 3.805% |
| X-n401-k29 | 66.204 | 67.696 | 2.253% | 67.577 | 1.817% |
| X-n411-k19 | 19.741 | 20.966 | 6.210% | 20.474 | 3.766% |
| X-n420-k130 | 108.137 | 134.019 | 23.935% | 116.761 | 7.975% |
| X-n429-k61 | 65.777 | 67.992 | 3.368% | 67.791 | 3.140% |
| X-n439-k37 | 36.484 | 38.941 | 6.735% | 39.588 | 8.509% |
| X-n449-k29 | 55.282 | 56.635 | 2.447% | 59.518 | 7.620% |
| X-n459-k26 | 25.279 | 26.603 | 5.236% | 26.423 | 4.534% |
| X-n469-k138 | 222.364 | 254.665 | 14.526% | 253.272 | 13.944% |
| X-n480-k70 | 89.842 | 92.585 | 3.053% | 92.446 | 2.907% |
| X-n491-k59 | 66.591 | 68.596 | 3.010% | 69.526 | 4.452% |
| X-n502-k39 | 69.262 | 74.836 | 8.049% | 76.976 | 11.150% |
| X-n513-k21 | 24.273 | 25.259 | 4.061% | 25.236 | 3.964% |
| *mean* | **45.183** | **48.263** | **5.469%** | **47.748** | **4.440%** |

Table 16: Detailed comparative results on CVRPLIB-Set-X benchmarks (1000 instances in total, 22 within (100, 200] clients and 78 within (203, 1000] clients). Data compared are quoted from one of the latest works, i.e., PO (Pan et al., 2025a), in the RL4CO field.

| SOLVER | TYPE | CVRPLIB | | | |
| --- | --- | --- | --- | --- | --- |
| | | (0, 200] DROP | (200, 1000] DROP | Total DROP | TIME |
| LKH3 (Helsgaun, 2017) | Heuristic | 0.36% | 1.18% | 1.00% | 16m |
| HGS (Vidal et al., 2012) | Heuristic | 0.01% | 0.13% | 0.11% | 16m |
| NeuroLKH (Xin et al., 2021) | Heuristic+SL | 0.47% | 1.16% | 0.88% | 16m |
| LEHD (Luo et al., 2023) | SL | 11.11% | 12.73% | 12.25% | 1.67s |
| BQ-NCO (Drakulic et al., 2023) | | 10.60% | 10.97% | 10.89% | 3.36s |
| POMO (Kwon et al., 2020) | | 5.26% | 11.82% | 10.37% | 0.80s |
| Sym-NCO (Kim et al., 2022) | | 9.99% | 27.09% | 23.32% | 0.87s |
| Omni-POMO (Zhou et al., 2023) | RL | 5.04% | 6.95% | 6.52% | 0.75s |
| ELG (RF) (Gao et al., 2024) | | 4.51% | 6.46% | 6.03% | 1.90s |
| ELG (PO) (Pan et al., 2025a) | | 4.39% | 6.37% | 5.94% | 1.90s |
| CORectifier | IL+RRL | **3.49%** | **5.98%** | **5.47%** | 2.11s |

## F.2 EXTENDED RESULTS ON CVRPLIB

Table 15 gives the per-instance solving quality on CVRPLIB (Set-X) with 100-512 client nodes. Note that the "ML4CO-Best" results are selected from the best performance of Sym-NCO models trained on uniform CVRP-{100, 200, 500} data, as reported by the authors of Ma et al. (2025b) with performance enhancements over the original implementation (Kim et al., 2022), while our model is merely trained on CVRP-100 data without such best-picking trick among candidate models. Settings: Sym-NCO: $N \times$ Sampling; CORectifier: $N_A = 16$; both are with classic local search (LS) enabled.

Table 16 gives the detailed comparisons following PO (Pan et al., 2025a), which reports the zero-shot generalization results on the entire CVRPLIB (Set-X) set, in split of 22 small instances and 78 large instances.

## F.3 EXTENDED RESULTS ON THE EFFECT OF HYPER-PARAMETERS (MATNET)

As supplementary to Fig. 5 and Table 6 in the main context, we present the drop-parameter curves or numerical results about the effect of the core hyper-parameters for ATSP in Fig. 7 and Table 17, in support of our selection of relatively larger probabilities for rectification for ATSP. We speculate this quantitative difference from those of TSP and PCTSP is that learning from solely distance matrices with inherent asymmetry is more difficult for current sequential models as they basically benefit from sampling multiple symmetric (equivalent) solution trajectories, thus demanding a deeper involvement of expert guidance.

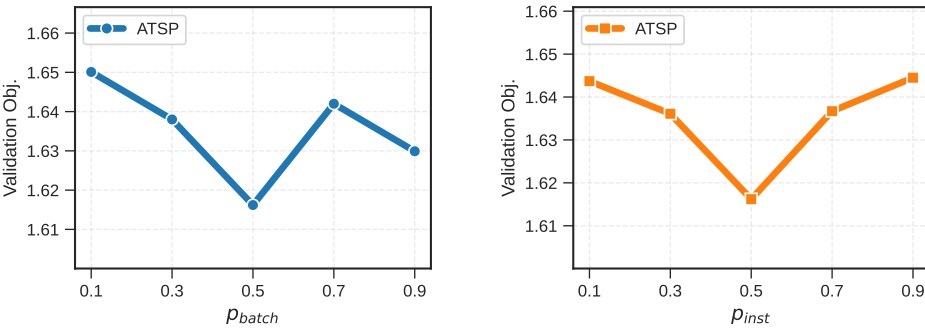

Figure 7: Effect of $p_{\text{batch}}$ and $p_{\text{inst}}$ on MatNet (for ATSP).

Table 17: Effect of the length range $(\alpha, \beta)$ of rectifiers on MatNet (for ATSP).

| $[\alpha, \beta]$ | OBJ. (ATSP-50) |
|---|---|
| $[0.0, 0.2]$ | 1.6703 |
| $[0.2, 0.4]$ | 1.6491 |
| $[0.4, 0.6]$ | 1.6448 |
| $[0.6, 0.8]$ | 1.6379 |
| $[0.8, 1.0]$ | **1.6162** |

## F.4 EXTENDED RESULTS OF STABILITY STUDY

**Variance of the Optimality Drops**. As supplementary to Table 18 in the main text, we offer the complete results of standard deviation of performance drops on the 10 evaluated benchmarks. We use "Greedy+2Opt" setting for TSP and ATSP, "Greedy+LS" for CVRP, and raw Greedy decoding for PCTSP and KP. The results validate the solving stability of CORectifier despite the inherent discrepancy among instance-wise difficulties.

Table 18: Full results of performance drops with respective standard deviations.

| TASK | DROP$\pm$STD. | TASK | DROP$\pm$STD. |
|---|---|---|---|
| TSP-50 | 0.132$\pm$0.281% | CVRP-50 | 0.984$\pm$0.900% |
| TSP-100 | 0.664$\pm$0.604% | CVRP-100 | 1.496$\pm$0.724% |
| TSP-500 | 3.886$\pm$0.601% | CVRP-200 | 2.052$\pm$0.581% |
| ATSP-50 | 0.585$\pm$0.674% | CVRP-500 | 3.185$\pm$0.527% |
| ATSP-100 | 2.394$\pm$1.024% | KP-50 | 0.013$\pm$0.046% |
| ATSP-200 | 4.638$\pm$0.921% | KP-100 | 0.012$\pm$0.029% |
| ATSP-500 | 7.751$\pm$0.784% | KP-200 | 0.009$\pm$0.017% |
| PCTSP-20 | 0.504$\pm$1.108% | KP-500 | 0.007$\pm$0.011% |
| PCTSP-50 | 1.445$\pm$1.328% | | |
| PCTSP-100 | 3.775$\pm$1.964% | | |

## F.5 EXTENDED RESULTS AND ANALYSIS ON KP

Table 19: Full results on KP. Compared results are partially quoted from POMO (Kwon et al., 2020) and GOAL (Drakulic et al., 2025). The objective values of the baseline methods are omitted because these existing literature do not open-source their original test datasets, while we have generated test instances with the exact distribution and settings they described, which shall be made public.

| METHOD | TYPE | KP-50 (1280 inst.) | | | KP-100 (1280 inst.) | | |
|---|---|---|---|---|---|---|---|
| | | OBJ.↑ | DROP↓ | TIME↓ | OBJ.↑ | DROP↓ | TIME↓ |
| OR-Tools (Cuvelier et al., 2023) | Exact | 20.021* | 0.000% | 0.001s | 40.302* | 0.000% | 0.002s |
| *Sequential Decision Methods* | | | | | | | |
| AM (Kool et al., 2018) | RL+G | – | 0.173% | – | – | 0.211% | – |
| POMO (Kwon et al., 2020) | RL+G | – | 0.130% | – | – | 0.19% | – |
| BQ-NCO (Drakulic et al., 2023) | SL+G | – | – | – | – | 0.10% | – |
| GOAL (Drakulic et al., 2025) | SL+G | – | – | – | – | 0.12% | – |
| CORectifier | RL+G | **20.018** | **0.013%** | 0.024s | **40.298** | **0.012%** | 0.043s |

| METHOD | TYPE | KP-200 (1280 inst.) | | | KP-500 (1280 inst.) | | |
|---|---|---|---|---|---|---|---|
| | | OBJ.↑ | DROP↓ | TIME↓ | OBJ.↑ | DROP↓ | TIME↓ |
| OR-Tools (Cuvelier et al., 2023) | Exact | 57.402* | 0.000% | 0.002s | 91.128* | 0.000% | 0.004s |
| *Sequential Decision Methods* | | | | | | | |
| AM (Kool et al., 2018) | RL+G | – | 0.325% | – | – | – | – |
| POMO (Kwon et al., 2020) | RL+G | – | 0.50% | – | – | 6.41% | – |
| BQ-NCO (Drakulic et al., 2023) | SL+G | – | 0.14% | – | – | 0.74% | – |
| GOAL (Drakulic et al., 2025) | SL+G | – | 1.63% | – | – | 2.40% | – |
| CORectifier | RL+G | **57.397** | **0.009%** | 0.058s | **91.121** | **0.007%** | 0.110s |

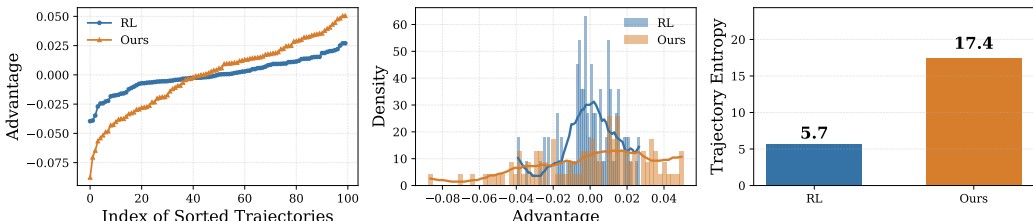

Figure 8: Exploration diversity on TSP-100. Left: Advantage separation. Mid: Advantage scale distribution. Right: Trajectory entropy.

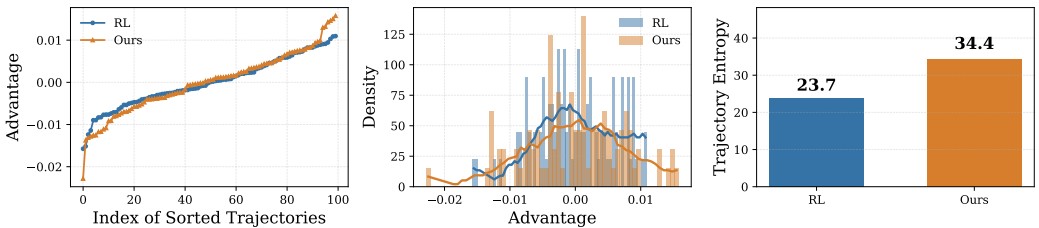

Figure 9: Exploration diversity on ATSP-100. Left: Advantage separation. Mid: Advantage scale distribution. Right: Trajectory entropy.

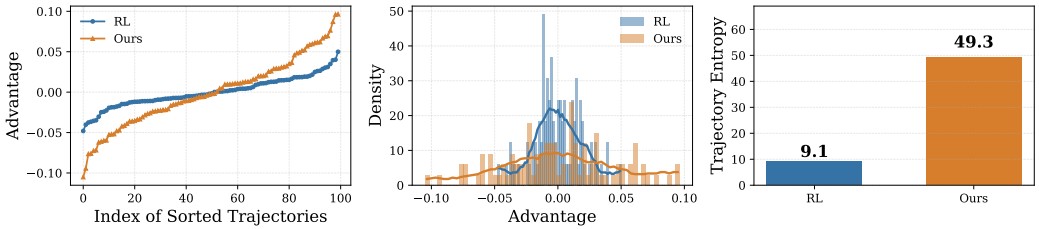

Figure 10: Exploration diversity on CVRP-100. Left: Advantage separation. Mid: Advantage scale distribution. Right: Trajectory entropy.

### F.6 EXTENDED VISUALIZATION OF THE EXPLORATION DIVERSITY

Fig. 8, Fig. 9, and Fig. 10 provide additional visualization of the exploration behaviors on TSP-100, ATSP-100, and CVRP-100. Across all problems, three consistent phenomena can be observed.

**Advantage separation.** The left panels show the sorted trajectory advantages. Our method consistently yields a wider spread between high- and low-advantage trajectories compared to the RL baseline. This indicates that CORectifier promotes a more diverse exploration profile, generating trajectories of both higher and lower advantage rather than collapsing toward a narrow band of behaviors.

**Advantage scale distribution.** The middle panels further confirm this effect. The RL baseline exhibits a sharper, more concentrated advantage distribution, whereas our method produces a noticeably broader density with heavier tails. This larger variance suggests that our agent explores more heterogeneous solution paths, covering regions that standard RL rarely visits.

**Trajectory entropy.** The right panels summarize trajectory-level diversity using a discrete entropy metric. Our method achieves substantially higher entropy across all tasks (e.g., 17.4 v.s. 5.7 on TSP-100, 34.4 v.s. 23.7 on ATSP-100, 49.3 v.s. 9.1 on CVRP-100), demonstrating that the generated solution trajectories differ more significantly from one another. This indicates a richer and less deterministic exploration process.

**Summary.** Taken together, these visualizations show that CORectifier consistently enhances exploration diversity from multiple perspectives: advantage ordering, advantage variance, and trajectory entropy. The effect is robust across symmetric, asymmetric, and capacitated routing problems. This provides further evidence that the rectified policy improves the quality of exploration rather than merely altering its magnitude.

## F.7 FEASIBILITY STUDY

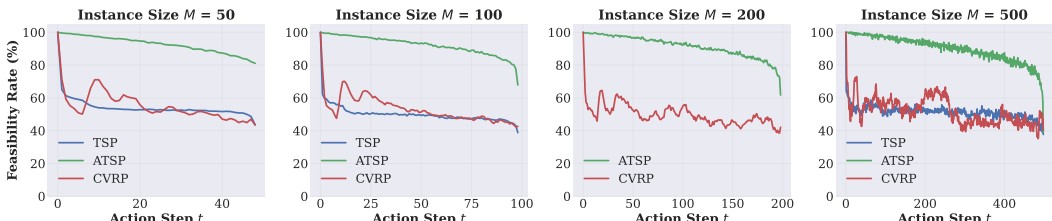

Figure 11: Per-step feasibility rate.

To understand how often expert rectifications are actually feasible during training, we measure the *per-step feasibility rate*: the proportion of samples where the expert-suggested next action remains valid under the current partial solution at step $t$. Table 20 summarizes results averaged over decision steps across tasks and scales. Fig. 11 shows the complete trend of teh feasibility rate with the decision steps grow. We observe that feasibility remains stable across instance sizes: around 50% for TSP/CVRP and above 90% for ATSP. This indicates that rectification opportunities do not diminish with longer horizons. Importantly, these feasibility levels are consistently higher than the rectification probabilities used during training, confirming that our rectifier operates in a well-controlled regime where infeasible expert actions rarely limit the effective rectification rate.

Table 20: Average per-step feasibility rate of expert rectification across tasks and instance sizes.

| Task | 50 | 100 | 200 | 500 |
|------|--------|--------|--------|--------|
| TSP  | 54.33% | 50.10% | –      | 51.36% |
| ATSP | 92.62% | 92.21% | 91.47% | 90.05% |
| CVRP | 51.31% | 50.33% | 48.32% | 50.62% |

## G A SKETCH OF PROOF FOR EQUATION 4

In this section, we provide a proof sketch for Eq. 4, i.e., $\exists \epsilon > 0, \mathbb{E}_{\tau \sim \pi_\theta(\cdot|G)} [c(\tau; G) - c(\tau_G^* \mid G)] > \epsilon, \forall G \in \mathcal{G}^*$, which is the central motivation and theoretical guarantee of the CORectifier framework. The equation asserts the existence of a positive quality margin $\epsilon$ between the expected cost of trajectories generated by the policy $\pi_\theta$ and the cost of reference trajectories $\tau_G^*$ for all labeled instances $G \in \mathcal{G}^*$. This margin ensures that the reference trajectories provide meaningful guidance during training. *Please note that this inequality, as well as the following proof, is more likely a conceptual illustration guaranteed by engineering practices rather than absolute mathematical rigor.*

**Quality of Reference Trajectories.** The reference trajectories $\tau_G^*$ are generated by established oracle solvers (e.g., Concorde for TSP, LKH-3 for ATSP, OR-Tools/ILS for PCTSP). These solvers are designed to produce exact or near-optimal solutions for their respective COPs. Formally, for any instance $G$, the cost $c(\tau_G^* \mid G)$ is guaranteed to be within a small factor of the (theoretical) optimal cost $c^*(G)$, often satisfying

$$c(\tau_G^* \mid G) \leq (1 + \delta)c^*(G), \tag{35}$$

for some small $\delta \geq 0$ (depending on the solver's approximation ratio). Therefore, since solutions generated by existing RL-based solvers possess strictly positive performance drops, $c(\tau_G^* \mid G)$ is reasonably qualified to constitute high-quality guiding signals.

**Policy-Generated Trajectories.** At the initial stages of training, the policy $\pi_\theta$ is typically suboptimal. Since COPs involve vast combinatorial action spaces, random exploration or naive policy updates often yield trajectories $\tau$ with costs $c(\tau; G)$ that are significantly higher than $c(\tau_G^* \mid G)$. Even as training progresses, unless the policy converges to the global optimum, the expectation $\mathbb{E}_{\tau \sim \pi_\theta(\cdot|G)}[c(\tau; G)]$ remains bounded away from $c(\tau_G^* \mid G)$ due to the complexity of the problem and the limited sample efficiency of vanilla RL.

**Existence of $\epsilon$-Margin.** Let

$$\Delta(G) := \mathbb{E}_{\tau \sim \pi_\theta(\cdot|G)}[c(\tau; G)] - c(\tau_G^* \mid G), \tag{36}$$

which, by the above arguments, implies $\Delta(G) > 0$ for each $G \in \mathcal{G}^*$. Since $\mathcal{G}^*$ is a finite set (in practice, a collection of labeled instances), and the policy $\pi_\theta$ is continuous in parameters $\theta$, the function $\Delta(G)$ attains a minimum over $\mathcal{G}^*$. Define $\epsilon := \min_{G \in \mathcal{G}^*} \Delta(G)$, and we have $\epsilon > 0$ because:

1) Each $\Delta(G) > 0$ (as reference trajectories are strictly better than the policy's expected performance initially or during training);

2) The minimum over a finite set of positive values is positive.

Thus, for all $G \in \mathcal{G}^*$, there exists some positive margin $\epsilon$ such that

$$\mathbb{E}_{\tau \sim \pi_\theta(\cdot|G)}[c(\tau; G) - c(\tau_G^* \mid G)] \geq \epsilon > 0 \tag{37}$$

holds, which satisfies Eq. 4.

**Practical Justification.** In practice, the value of $\epsilon$ may vary with problem size and instance distribution, but it remains positive as long as the oracle solvers outperform the policy. This margin justifies the hierarchical rectification mechanism in CORectifier. By replacing segments of policy-generated trajectories with reference segments, the rectified trajectories inherit the low cost of $\tau_G^*$, thereby providing strong learning signals (via advantage estimates) to guide the policy toward better regions.

This proof sketch relies on the realistic assumption that strong heuristic solvers provide high-quality solutions, which is standard in NCO practices. Empirical results in Sec. 4.2 should be viewed as grounded validation that the margin $\epsilon$ exists as CORectifier consistently improves over RL baselines.

## H    LIMITATIONS

While CORectifier significantly mitigates core challenges in RL-based CO solving (e.g., reward sparsity, sample inefficiency, scalability issue), not a single work could claim to fully resolve fundamental limitations inherent to broad RL frameworks, which may still manifest in specific scenarios. In some cases (e.g., novel CO variants with no established optimal solvers, or large-scale instances where expert trajectory generation is computationally intractable), acquiring references may be cost-prohibitive. That said, CORectifier demonstrates robust performance improvements over existing methods within the same RL4CO category when expert data is available. Critically, as stated in Sec. 4.3 (stability study), this paradigm exhibits a "degradation safety guarantee": as expert data quality declines or supervision is entirely absent, CORectifier naturally (the worst case) degenerates to vanilla RL, eliminating the risk of performance falling below the baseline established by standalone RL solvers. We hold it firmly that advancing the ML4CO field requires synergistic efforts across multiple directions (e.g., expert data distillation, unsupervised RL for CO, and backbone architecture innovation), and believe this work represents one step toward addressing RL4CO's core limitations.

## I    BROADER IMPACT

To the best of our knowledge, CORectifier represents one of the earliest efforts to integrate the complementary strengths of reinforcement learning (RL), supervised learning (SL), and imitation learning (IL) for CO solving. It further establishes a novel learning paradigm centered on hierarchically rectified trajectories to develop RL-based CO solvers with enhanced sample efficiency and awareness of local optimality. Beyond the specific domain of CO, this paradigm holds potential for extension to broader machine learning (ML) fields as a foundational learning framework. Its applicability hinges on two key task characteristics: 1) the task can be formalized as a sequential action-making process (consistent with the formulation of vast conventional RL algorithms); and 2) high-quality

expert action trajectories are relatively accessible. In such scenarios, the rectification mechanism introduced in CORectifier is expected to provide value by guiding an agent's random exploration and regularizing the intractably large action space from a new perspective.

## J   THE USE OF LARGE LANGUAGE MODELS

During the preparation of this work, Large Language Models (LLMs) were used solely for polishing the writing, with the goal of ensuring grammatical accuracy and readability of the textual contents.

