# OpenReview forum: "CORectifier: Hierarchical Trajectory Rectifications Boost Reinforcement Learning for Combinatorial Optimization"
_ICLR.cc/2026/Conference — Submitted to ICLR 2026_

### Official Review · Reviewer_sQuT · 2025-10-16

**Soundness:** 2
**Presentation:** 2
**Contribution:** 2
**Rating:** 2
**Confidence:** 4

**Summary:**

This paper proposes CORectifier, a reinforcement-learning-based framework for Neural Combinatorial Optimization (NCO) that introduces hierarchical trajectory rectifications.

The key idea is to probabilistically replace partial segments of policy-generated trajectories with locally optimal fragments from expert solutions during training.

This “Rectified Reinforcement Learning” (RRL) paradigm aims to alleviate reward sparsity and poor sample efficiency in standard RL-based solvers while retaining the flexibility of sequential decision making.
Empirical results on TSP, ATSP, and PCTSP show significant improvements over prior RL-based baselines and competitive performance compared to supervised and unsupervised methods.

**Strengths:**

**Originality**

-	The paper introduces an interesting and nontrivial attempt to integrate expert information into RL via partial trajectory replacement, bridging imitation and RL in a structured way. Extensive results with ablations, comparison with recent neural solvers
-	The three-level (batch, instance, sub-instance) rectification mechanism is well-motivated and could be a general recipe for hybrid learning in NCO.

**Quality**

-  common and consistent notation, good writing

**Significance**

-	Extensive results with ablations, comparison with recent neural solvers (not only RL-based ones) which show substantial gains in solution quality
-	The idea of reusing expert solutions as decomposable local fragments is promising, especially under limited supervision.

**Weaknesses:**

**Limited applicability beyond simple TSP-like problems**

The proposed rectification mechanism fundamentally relies on being able to replace local segments of a trajectory while maintaining feasibility.
This assumption breaks down for problems with richer constraints (e.g., scheduling with precedence relations, capacitated VRPs, PDPs), where feasibility depends on multi-dimensional states (capacity, time, precedence).
In such settings, ```Feas(a*, s)``` would likely fail for most rectifications, making the method ineffective.
Experiments are restricted to TSP variants, which all share the same single-tour structure.

**Lacking clarity**

The paper would benefit from a more detailed figure describing the rectification process or a small toy example. All in all, The definition of the rectifier $\mathcal{R}(\tau, \tau^\ast, \mathcal{M})$ seems underspecified (see questions).

**Dependence on expert-labeled data**

The approach requires high-quality reference solutions for supervision. For larger or more complex problems, such data are difficult and expensive to obtain.

**Scaling behavior**

The method underperforms on larger instances compared to recent heatmap-based approaches. The authors should explicitly discuss why this happens (e.g., policy entropy collapse, reduced rectification success) and suggest possible remedies.

**Lack of detailed analysis of the rectifier’s behavior**

The paper would benefit from more in-depth analysis on the rectifier operation. E.g., showing the impact of rectification on the reward; analysing how often a rectification fails (possibly grouped by step t, as I reckon rectification in later stages is more difficult as most nodes are already visited)

**Questions:**

- How could CORectifier be applied to problems with more complex constraints, such as job-shop scheduling, capacitated VRPs, or pickup-and-delivery problems?
Given that feasibility in these domains depends on non-local state variables, how would rectification be defined or enforced?

- In Algorithm 1 and Eq. (10), the rectification operator replaces certain actions in a sampled trajectory $\tau$ with successors of corresponding nodes in the expert trajectory $\tau^\ast$. However, after such a replacement, the subsequent actions in $\tau$ were originally generated under a different state (i.e., before rectification)- Could the authors clarify, how exactly the remainder of the trajectory, i.e., $\tau_{t+k+1:M}$ is constructed / restored to ensure feasibility?

---

> ### Author Response · Authors · 2025-11-17
> **Author Response to Reviewer sQuT (Page 1)**
>
> Dear Reviewer sQuT,
>
> Thanks for your valuable review and helpful suggestions, and we genuinely appreciate your recognition of our **well-motivated attempt, good writing, as well as extensive and promising experiments.**
>
> In the following, we sincerely present our detailed clarifications and/or additional experimental results in response to all your concerns. Moreover, we have accordingly updated our manuscript ([click here to download](https://openreview.net/pdf?id=xniQIl8oTw)) that reflects the comprehensive efforts we have made during rebuttal, with major changes marked in red text for your convenient reference. We stay more than willing to reach an extended consensus with you during the discussion phase! We are fully committed to reaching an extended consensus with you, and still stay positive towards our discussions!
>
> ---
> ### **Author responses to the weaknesses/questions:**
>
> > **W1 & Q1: Applicability beyond TSP-like problems**
> > The method assumes that local trajectory replacements remain feasible, which fails for problems with richer constraints (e.g., scheduling, VRP, PDP). How could this limitation be mitigated?
>
> **A:** Thank you very much for raising this important concern.
>
> First, we would like to emphasize that the routing problem family constitutes a well-established and widely recognized research direction within the Neural Combinatorial Optimization (NCO) community. **Numerous influential works published in top-tier venues have focused exclusively on routing problems [1–14], and their contributions are broadly regarded as foundational rather than limited in scope.**
>
> From a technical standpoint, our hierarchical rectification mechanism is particularly well-suited to routing tasks because of their *geometric nature*, which involves rich topological connectivity and spatially ordered dependencies. Although these problems contain well-structured constraints, as you rightly note, their intricate spatial correlations make them especially challenging for neural models to learn effectively, thus serving as ideal testbeds for demonstrating the benefits of our intra-instance-level rectification. In contrast, problems such as Maximum Independent Set (MIS) or the Knapsack Problem (KP) exhibit permutation-invariant objectives, where the order of element selection does not affect the final outcome. Therefore, our initial experiments concentrate on routing problems, where local order and spatial structure play pivotal roles.
>
> That said, we completely agree on the importance of evaluating generality beyond relatively simple routing constraints. To this end, we have extended our experiments to two additional combinatorial optimization problems with non-trivial structures: the **Capacitated Vehicle Routing Problem (CVRP)** and the **Knapsack Problem (KP)**.
>
> * **For CVRP**, we leverage the structural property that an optimal solution can be decomposed into multiple sub-tours. Accordingly, intra-instance-level rectification is applied to optimal sub-trajectories—specifically, route segments between a vehicle’s departure from and return to the depot. The decision to inject such an optimal sub-tour (with probability $p_{\text{sub}}$), instead of the $[\alpha,\beta]$ interval used in TSP) is made only when the agent is positioned at the depot. A simplified toy illustration of the sub-instance rectification mask (as introduced in Eq. (9) in the paper) is:
>
>  $$ \mathcal{M}\_{G_i}=\begin{pmatrix}
>   1\cdots1,0\cdots0,\cdots,1\cdots1 \\\\
>   1\cdots1,0\cdots0,\cdots,0\cdots0 \\\\
> \cdots \\\\
>   0\cdots0,1\cdots1,\cdots,1\cdots1
>   \end{pmatrix}_{N\times M'} $$
>   where contiguous blocks of ones mark the rectified sub-trajectories (with (N) samples per instance and (M') the decision-sequence length).
>
> * **For KP**, item selection is inherently order-independent. Since the intra-instance rectifier is designed to capture sequential relationships in locally ordered contexts, it is disabled during KP training. Instead, when a sampled trajectory is masked *True* for instance-level rectification, the entire trajectory is replaced with a randomly permuted sequence containing all items from the optimal reference solution.

---

> ### Author Response · Authors · 2025-11-17
> **Author Response to Reviewer sQuT (Page 2)**
>
> ### **a. Results for uniform CVRP-50/100:**
> (LS: classic local search for CVRP, proposed in [19])
>
> | **Method**| **Type**| **CVRP-50 (10k inst.)** ||| **CVRP-100 (10k inst.)** |||
> |:--- |:-- |:---:|:----------:|:---------:|:--:|:----------:|:---------:|
> |||**Obj.↓**| **Drop↓**  | **Time↓** |**Obj.↓**| **Drop↓**  | **Time↓** |
> | HGS| Heuristics |10.366*|0.000%   |  1.005s|15.563*|   0.000%|  20.027s  |
> | *Heatmap-Guided Methods*||||||||
> | Fast-T2T| SL+G|12.640|  21.835%|  0.009s|19.202|  23.333%   |  0.010s   |
> | COExpander| SL+G|11.979|  15.407%   |  0.033s   |17.497|  12.343%   |  0.047s|
> | Fast-T2T + LS| SL+G+LS|10.871|   4.836%   |  0.013s   |16.294|   4.698%   |  0.018s   |
> | COExpander + LS| SL+G+LS|10.773|   3.903%   |  0.037s   |16.224|   4.253%   |  0.055s|
> | *Sequential Decision Methods* ||||||||
> | AM| RL+G|10.98|   5.86%|–|16.80|   7.34%    |–|
> | POMO| RL+G|10.74|   3.52%    |–|16.15|3.00%    |–|
> | Sym-NCO| RL+G|10.769|   3.891%   |  0.087s   |16.220|   4.241%|  0.166s   |
> | GOAL| SL+G|10.906|   5.193%   |  0.504s   |16.342|   5.005%   |  0.962s   |
> | **CORectifier (ours)**| RRL+G|**10.540**| **1.668%** |  0.042s   |        **15.939**        | **2.425%** |  0.079s|
> | **CORectifier ($N_A=16$)**| RRL+G       |**10.447**| **0.768%** |  0.049s   |        **15.799**| **1.521%** |  0.108s   |
> | Sym-NCO + LS| RL+G+LS|10.505|1.910%   |  0.168s   |15.933|   2.379%   |  0.173s   |
> | GOAL + LS| SL+G+LS|10.628|2.519%|  0.507s   |15.959|   2.548%   |  0.969s|
> | **CORectifier + LS (ours)**| RRL+G+LS|**10.469**| **0.984%** |  0.045s   |**15.796**| **1.496%** |  0.085s   |
> | **CORectifier + LS ($N_A=16$)**  | RRL+G+LS|**10.412**| **0.437%** |  0.052s   |**15.706**| **0.919%** |  0.113s|
>
> ---
> ### **b. Results for uniform CVRP-200/500:**
>
> | **Method**| **Type**   | **CVRP-200 (100 inst.)** ||| **CVRP-500 (100 inst.)** |||
> |:--- |:------- |:--------:|:------:|:---------:|:-----:|:--------:|:---------:|
> |||**Obj.↓**| **Drop↓** | **Time↓** |**Obj.↓**| **Drop↓** | **Time↓** |
> | HGS| Heuristics |19.630*|  0.000%|  60.024s  |37.154*|  0.000%   | 360.376s  |
> | *Heatmap-Guided Methods*||||||||
> | Fast-T2T| SL+G|25.064|  27.616%  |  0.059s   |47.749|  28.509%  |  0.091s   |
> | COExpander| SL+G|22.402|  13.977%  |  0.145s   |43.901|  18.199%  |  0.554s   |
> | Fast-T2T + LS| SL+G+LS|20.662|  5.290%|  0.063s|39.195|  5.530%|  0.215s   |
> | COExpander + LS| SL+G+LS|20.587|  4.893%   |0.153s|39.121|  5.337%   |  0.605s|
> | *Sequential Decision Methods*||||||||
> | Sym-NCO| RL+G|20.662|  5.274%|  0.320s   |40.382|  8.723%|  0.769s   |
> | **CORectifier (ours)**|RRL+G|**20.270**|**3.260%**|0.159s|**39.129**|**5.343%**|0.368s|
> | **CORectifier ($N_A=16$)**| RRL+G|**20.129** |**2.541%**|0.219s|**38.874**|**4.650%**|1.485s|
> | Sym-NCO + LS| RL+G+LS|20.193|  2.880%   |  0.341s   |38.700|4.173%|  0.883s   |
> | **CORectifier + LS (ours)**| RRL+G+LS   |**20.032**|**2.052%**| 0.171s|**38.329**|**3.185%**|0.461s|
> | **CORectifier + LS ($N_A=16$)**  | RRL+G+LS   |**19.952**|   **1.638%**   |0.237s|**38.176**|**2.758%**|1.567s|
>
> ---
> ### **c. Zero-shot generalization results on CVRPLib:**
> (1000 instances total: 22 within (100, 200] clients and 78 within (203, 1000] clients. Results quoted from PO4CO[1].)
>
> | **Solver**| **Type**| **(0, 200] Drop↓** | **(200, 1000] Drop↓** | **Total Drop↓** |  **Time**  |
> | :----- | :--- | :---: | :----: | :--: | :--------: |
> | LKH3| Heuristic|0.36 %|1.18 %|1.00 %|    16 m    |
> | HGS| Heuristic|0.01 %|0.13 %|0.11 %|    16 m    |
> | NeuroLKH| Heuristic + SL |0.47 %|1.16 %|0.88 %|    16 m    |
> | LEHD| SL|11.11 %|12.73 %|12.25 %|1.67 s   |
> | BQ-NCO| SL|10.60 %|10.97 %|10.89 %|   3.36 s   |
> | POMO| RL|5.26 %|11.82 %|10.37 %|   0.80 s   |
> | Sym-NCO| RL|9.99 %|27.09 %|23.32 %|   0.87 s   |
> | Omni-POMO| RL|5.04 %|6.95 %|6.52 %|   0.75 s   |
> | ELG (RF)| RL|4.51 %|6.46 %|6.03 %|   1.90 s   |
> | ELG (PO)| RL|4.39 %|6.37 %|5.94 %|   1.90 s   |
> | **CORectifier (ours)** | RRL|**3.49 %**|**5.98 %**|**5.47 %**| 2.11 s |
>
> ---
> ### **d. Results for KP-50/100/200/500:**
>
> | **Method**| **Type** | **KP-50 (1280 inst.)** || **KP-100 (1280 inst.)** || **KP-200 (1280 inst.)** || **KP-500 (1280 inst.)** ||
> |:--|:---|:---:|:--:|:----:|:--:|:---:|:--:|:--:|-|
> |||**Obj.↑**| **Drop↓**|**Obj.↑**| **Drop↓**  |**Obj.↑**        | **Drop↓**  |**Obj.↑**| **Drop↓**  |
> | OR-Tools| Exact    |20.021*|   0.000%   |40.302*|   0.000%   |57.402*|   0.000%   |91.128*| 0.000%|
> | AM| RL+G|–|   0.173%|–|   0.211%   |–|   0.325%   |–| –|
> | POMO| RL+G|–|   0.130%   |–|   0.190%   |–|   0.500%   |–| 6.410%     |
> | BQ-NCO| SL+G|–|–|–|   0.100%   |–|   0.140%   |–| 0.740%     |
> | GOAL| SL+G|–|–|–|   0.120%   |–|   1.630%   |–| 2.400%     |
> | **CORectifier (ours)** | RRL+G|**20.018**| **0.013%** |**40.298**| **0.012%** |**57.397**| **0.009%** |**91.121**| **0.007%** |

---

> ### Author Response · Authors · 2025-11-17
> **Author Response to Reviewer sQuT (Page 3)**
>
> These extended experiments (above in Page 2) demonstrate that CORectifier is not confined to routing tasks; rather, it flexibly adapts its rectification strategy to diverse problem structures, including non-sequential and constraint-heavy combinatorial optimization settings.
>
> We have also included detailed problem formulations (**Appendix B**), data generation procedures (**Sec. 4.1**), comprehensive experimental results (**Tables 5, 8, 9, 15, 16, 19**), and model configuration details (**Table 13**) in the [revised manuscript](https://openreview.net/pdf?id=xniQIl8oTw), ensuring full narrative consistency with the main paper. We sincerely hope these extended studies help alleviate your concerns on this matter.
>
> ---
> > **W2 & Q2: Lack of clarity in rectifier definition**
> > i) The definition of the rectifier operator ($R$) is underspecified. A figure or toy example could help clarify. Can you provide such clarification? ii) In Algorithm 1 and Eq. (10), the rectification operator replaces certain actions in a sampled trajectory $\tau$ with successors of corresponding nodes in the expert trajectory $\tau^\*$. However, after such a replacement, the subsequent actions in $\tau$ were originally generated under a different state (i.e., before rectification)- Could the authors clarify, how exactly the remainder of the trajectory, i.e., $\tau_{t+k+1:M}$ is constructed / restored to ensure feasibility?
>
> **A:** Thank you very much for these careful and insightful questions. We are happy to clarify both the definition of the rectifier operator and how feasibility is preserved in the suffix of the trajectory.
>
> **(i) On the definition of the rectifier operator $\mathcal{R}$**
>
> For mathematical rigor, we define the rectification operator $\mathcal{R}(\tau,\tau_G^\*,\mathcal{M}) \mapsto \tau'$ in the paper from a **global** perspective, as if the full policy trajectory $\tau$ and the expert trajectory $\tau_G^\*$ were both precomputed and then “patched” according to the mask $\mathcal{M}$ as in Eq. (10). This global description was chosen mainly for *expository clarity*, so that the notion of “replacing segments” is easy to follow at a macroscopic level.
>
> However, in **practical training**, the rectification is implemented **online and locally at each step**, not by first generating a full $\tau$ and then post-hoc modifying it. Concretely (taking TSP as an example):
>
> 1. At step $t$, given the current partial tour $\tau_{1:t-1}$, the policy $\pi_\theta$ produces a distribution over feasible next nodes (with standard masking of already visited nodes).
> 2. If the rectification mask for this trajectory and time step satisfies $\mathcal{M}_{G_i}[j,t]=1$ and the expert successor $a_t^\*$ in $\tau_G^\*$ is feasible in the current state, we take **$a_t = a_t^*$** as the next action.
> 3. Otherwise, we sample $a_t$ from $\pi_\theta(\cdot \mid s_t)$ as in vanilla RL.
> 4. We then append $a_t$ to the partial trajectory and move to step $t+1$, repeating this procedure until the tour is complete.
>
> In other words, $\mathcal{R}$ is conceptually defined as operating on trajectories, but is *implemented* as a stepwise, autoregressive process that sometimes swaps the model's proposal with a feasible expert action at the current step. To make this clearer, we have added a pseudo-code style description (on TSP) below for your reference.
>
> ```python
> # B: batch_size, N: num_samples, M: num_nodes (consistent with the paper)
>
> class TSPTrainer:
>     # omit other functions ...
>     def _train_one_batch(self):
>         # data preparation ...
>
>         # sample from [alpha, beta] to decide the number of rectified actions
>         rectify_level = int(M * random.uniform(alpha, beta))
>         # construct the mask \mathcal{M} as in Eq. (9) in the paper: (B * N, M)
>         global_action_mask = torch.zeros(B * N, M).bool()
>         global_action_mask[:, :rectify_level] = True
>         global_action_mask.roll(random.randint(0, M - 1), dims=1)
>         # construct instance mask with prob. p_inst: (B * N,)
>         instance_mask = torch.rand(B * N) < p_inst if enable_rectify else None
>
>         while not done:
>             # extract intra-instance level mask from \mathcal{M}
>             action_mask = global_action_mask[:, self.env.selected_count].reshape(-1)
>             # forward pass, in which rectification is done
>             action, prob = self.model.forward(state, instance_mask, action_mask)
>             # update states and calculate rewards
>             state, reward, done = self.env.step(action)
>             # record probs for the rectified actions
>             prob_list = torch.cat((prob_list, prob[:, :, None]), dim=2)
>
>         # backpropagation ...
> ```

---

> ### Author Response · Authors · 2025-11-17
> **Author Response to Reviewer sQuT (Page 4)**
>
> ```
> class TSPModel(nn.Module):
>     # omit other functions ...
>     def forward(
>         self,
>         state: Step_State,
>         instance_mask: Tensor = None, # control instance-level rectifier
>         action_mask: Tensor = None    # control intra-instance rectifier
>     ):
>         """
>         Output the next action given current state,
>         with the action being optionally rectified.
>         d: hidden dimension
>         """
>         # load the reference expert trajectories
>         ref_tours = state.ref_tours
>         # set the first action as with POMO: [0, 1, ..., N]
>         if state.current_node is None:
>             selected = torch.arange(N)[None, :].expand(B, N)
>             prob = torch.ones(size=(B, N))
>         else:
>             # get node embeddings: (B, N, d)
>             node_embed = _get_encoding(state.current_node)
>             # decode probabilities for all candidate actions: (B, N, M)
>             # infeasible actions are masked with -infinity
>             probs = self.decoder(node_embed, ninf_mask=state.ninf_mask)
>             if self.training:
>                 # sample (initial) next action from current policy: (B, N)
>                 init_action = probs.reshape(B * N, -1).multinomial(1).squeeze(1).reshape(B, N)
>                 if instance_mask is not None:
>                     # prepare for rectification
>                     init_action = init_action.reshape(B * N)
>                     cur_node = state.current_node.reshape(B * N)
>                     unvisited = (state.ninf_mask == 0).reshape(B * N, -1)
>                     ref_tours = ref_tours.expand(B, N, M).reshape(B * N, M)
>                     # get rectified action
>                     rectified_action = _rectify_action(
>                         cur_node, init_action, action_mask,
>                         instance_mask, ref_tour, unvisited
>                     ).reshape(B, N)
>                     prob = _get_action_prob(probs, rectified_action)
>                 else: rectified_action = init_action
>
>         return rectified_action, prob
>
> def _rectify_action(
>     cur_node: Tensor,      # shape: (B * N,)
>     init_action: Tensor,   # shape: (B * N,)
>     instance_mask: Tensor, # shape: (B * N,)
>     action_mask: Tensor,   # shape: (B * N,)
>     ref_tour: Tensor,      # shape: (B * N, M)
>     unvisited: Tensor      # shape: (B * N, M)
> ):
>     # find the position of current node in the reference tours: (B * N,)
>     cur_node_idx = _get_idx_by_node(ref_tour, cur_node)
>     # get the position of the next node in the reference tours
>     ref_next_idx = (cur_node_idx + 1) % M
>     # get the next node in the reference tours: (B * N,)
>     ref_next_node = _get_node_by_idx(ref_tour, ref_next_idx)
>     # get the unvisited mask for the next reference action: (B * N,)
>     unvisited_mask = unvisited[torch.arange(B * N), ref_next_node]
>     # initialize new actions
>     rectified_action = torch.zeros_like(init_action).long()
>     # replace action if feasible
>     rectified_action = torch.where(unvisited_mask, ref_next_node, init_action)
>     # maintain the replaced action with instance and intra-instance rectification probabilities
>     rectified_action = torch.where(action_mask & instance_mask, rectified_action, init_action)
>
>     return rectified_action
> ```

---

> ### Author Response · Authors · 2025-11-17
> **Author Response to Reviewer sQuT (Page 5)**
>
> **(ii) On how $\tau_{t+k+1:M}$ is constructed and feasibility preserved**
>
> Your reading is correct that, if one literally *patched* an already-sampled trajectory $\tau$ by overwriting some actions with expert successors, then the remaining suffix $\tau_{t+k+1:M}$ would have been generated under a different state and could become inconsistent. This is *not* how we implement rectification.
>
> Instead, as shown in the code above, because rectification is performed **on the fly**, the suffix is **re-generated conditionally on the rectified prefix**, not reused from a trajectory sampled under the old state:
>
> * Suppose a block of $k$ consecutive steps $t,\dots,t+k-1$ is rectified using expert actions (subject to feasibility checks). After these replacements, the partial trajectory $\tau_{1:t+k}$ is already **consistent** with the environment dynamics and constraints (e.g., visited-node masks in TSP).
> * Then, at step $t+k+1$, the policy observes the updated state induced by $\tau_{1:t+k}$ (including all rectifications), and samples the next action according to $\pi_\theta(\cdot \mid s_{t+k+1})$ with the usual feasibility masks.
> * The subsequent actions $\tau_{t+k+2:M}$ are then generated **autoregressively** in exactly the same way as in standard RL, but conditioned on this rectified prefix.
>
> Therefore, the suffix $\tau_{t+k+1:M}$ is **not** taken from a trajectory that was generated before rectification; it is **newly sampled** after the rectified actions have been applied, so the underlying state and feasibility masks are always consistent. This design ensures that the full trajectory $\tau'$ remains feasible and coherent with the dynamics of the COP, and it avoids the state-mismatch issue you rightly pointed out.
>
> We have also clarified this point explicitly in the [revised manuscript](https://openreview.net/pdf?id=xniQIl8oTw) (**Appendix C**), and we thank you again for highlighting this subtle but important technical detail.
>
> ---
> > **W3: Dependence on expert-labeled data**
> > The approach relies on high-quality reference solutions, which are expensive or unavailable for large or complex problems.
>
> **A:** We fully agree that isolating the contribution of expert data quantity provides a clearer understanding of the rectified learning mechanism. In response, we have conducted additional experiments to study how the amount of expert supervision influences the training efficiency and final performance.
>
> Given the limited rebuttal timeline, we evaluated representative settings: TSP-100, ATSP-100, and CVRP-100, under three levels of expert data: **1.28k**, **12.8k**, and **128k** trajectories, training each model for **200 epochs** with all other hyperparameters fixed default.
>
> * **TSP-100**
>
> | Expert Data | Obj.↓  |
> |-|-|
> |128k| **7.9559** |
> |12.8k| 7.9562 |
> |1.28k| 7.9738 |
>
> * **ATSP-100**
>
> | Expert Data | Obj.↓  |
> |-|-|
> |128k| **1.6920** |
> |12.8k| 1.6925 |
> |1.28k| 1.6973 |
>
> * **CVRP-100**
>
> | Expert Data | Obj.↓   |
> |-|-|
> | 128k| **16.3118** |
> | 12.8k| 16.3127 |
> | 1.28k| 16.3132 |
>
> It can be observed that:
>
> 1. **More expert data naturally yields mild performance improvements**, as additional expert guidance further stabilizes early training.
> 2. **Even with only 1/100 of the expert data, the performance degradation is very small and no training collapse occurs**, highlighting that CORectifier remains stable and effective under sparse expert supervision.
>
> Overall, these results demonstrate that our framework is **robust to the quantity of expert data**, and remains practically applicable in scenarios where collecting abundant high-quality expert trajectories may be costly.
>
> **Beyond the technical results**, we respectfully emphasize that relying on oracle solvers to generate (near-)optimal reference data is a ***well-established and mainstream practice in the NCO community***.
> - **Numerous representative works [1–3,16-22,etc.]**, including many published at top-tier venues, adopt supervised learning (SL) paradigms that depend on oracle-generated labels (just as labeled data are indispensable for advancing modern AI research in vision, language, and planning domains, etc). In particular, recent **generative CO solvers [16–19]**, despite requiring substantial computational resources and large quantities of optimally labeled data, have been widely recognized and appreciated by the machine learning community.
> - In this context, our ***high-efficiency*** use of ***quantity-insensitive*** expert data is fully consistent with the prevailing methodology of the field rather than contradictory to its purpose. While obtaining sufficient optimal data is technically non-trivial in some cases, we firmly believe that enabling neural networks to **gradually capture the intrinsic structural patterns of NP-hard problems** and **narrow the performance gap with oracle solvers** undoubtedly represents a meaningful and forward-looking research direction, which contributes to deeper understanding and progress in learning-based CO.

---

> ### Author Response · Authors · 2025-11-17
> **Author Response to Reviewer sQuT (Page 6)**
>
> > **W4: Performance v.s. heatmap-based methods**
> > The method underperforms on larger instances compared to heatmap-based approaches. Please explain why this happens (e.g., policy entropy collapse, reduced exploration diversity).
>
> **A:** Thank you for this thoughtful question.
>
> 1. **From a high-level perspective**, we openly acknowledge the performance gap between RL-based sequential solvers and heatmap-guided supervised predictive methods as problem size increases (lines 458–460). This trend is well-documented in the NCO literature: **even the most recent RL4CO advances [4,12,15]** primarily evaluate their methods on instances no larger than 100-sized instances and conduct comparisons mainly within the RL4CO family, **in line with earlier mainstream works [2,4-12,15,etc.]**. In this context, where it is conventionally accepted that RL4CO solvers are evaluated on relatively small instances and solely compared against other RL-based approaches, our goal extends beyond competing for a “SOTA” title. Our hope is to provide a **holistic and transparent assessment of where RL-based solvers stand within the broader NCO ecosystem**, ultimately offering actionable insights for their future development. We believe that such transparency is academically constructive and should not be viewed as a limitation simply because RL does not surpass supervised alternatives on certain benchmarks.
>
> 2. **From a technical standpoint**, RL-based approaches inherently suffer from inefficient exploration in high-dimensional combinatorial spaces, leading to issues such as entropy collapse and reduced sampling diversity, precisely what you have pointed out. These limitations fundamentally restrict their scalability and ultimate performance. In fact, our CORectifier framework is specifically designed to mitigate this challenge by introducing **structured, limited expert guidance** at three granular levels. This hierarchical rectification increases exploratory entropy especially when the policy tends to converge prematurely, strengthens optimality awareness, thus substantially improving training effectiveness. As reflected in our results on TSP/ATSP/CVRP-200/500, CORectifier yields markedly smaller performance drops compared with existing RL-based solvers. While instances with 200–500 nodes may still be moderate in scale for heatmap-based methods, **they remain highly challenging for sequential RL**, and thus we view our progress as a meaningful step toward narrowing this long-standing gap.
>
> 3. **Dialectically**, just as Reviewer xVoR states, **TSP and ATSP are highly structured routing problems where heatmap-based solvers excel, but these methods often struggle to generalize to problems with more complex constraints**. Our extended empirical study (**Tables 4, 5, and 9 in the [revised version](https://openreview.net/pdf?id=xniQIl8oTw)**) reinforces this point: very few heatmap-guided baselines can be applied effectively to tasks such as PCTSP, CVRP, or KP, whereas RL-based approaches offer greater flexibility in modeling diverse constraints due to their sequential decision-making nature.
>
> **In summary**, we face up to the performance gap relative to heatmap-based approaches on certain large, well-structured routing benchmarks. At the same time, we believe that CORectifier constitutes a substantive advancement for RL-based combinatorial optimization: it demonstrably improves scalability and solution quality, and it strengthens the potential of RL methods to address more complex, constraint-rich CO tasks where purely supervised, heatmap-based paradigms typically struggle. We hope this perspective clarifies our motivation and positioning within the broader NCO landscape.

---

> ### Author Response · Authors · 2025-11-17
> **Author Response to Reviewer sQuT (Page 7)**
>
> > **W5: Analysis of the rectifier's behavior**
> >The paper would benefit from more in-depth analysis on the rectifier operation. E.g., showing the impact of rectification on the reward; analysing how often a rectification fails (possibly grouped by step t, as I reckon rectification in later stages is more difficult as most nodes are already visited)
>
> **A:** Thank you for the insightful suggestions! In the [revised manuscript](https://openreview.net/pdf?id=xniQIl8oTw) (**Appendix F.6 & F.7**), we have added two new analyses that directly address: 1) the **impact of rectification on the reward/advantage**, and 2) **how often rectification fails across different steps ($t$)**.
>
> **1. Impact of Rectification on the Reward (Advantage Distribution).** To show how rectification affects the reward signal, we include extended visualizations (**Figs. 8–10 in Appendix F.6**) comparing our method with the RL baseline across TSP-100, ATSP-100, and CVRP-100. The results consistently show:
>
> * **Wider advantage ranges** under our method, meaning the policy explores both higher- and lower-reward trajectories.
> * **Higher variance (smoother) in the advantage distribution**, indicating richer and more informative reward signals.
> * **Substantially higher trajectory entropy**, confirming that rectification expands exploration rather than collapsing it.
>
>     | Mean Trajectory Entropy | TSP-100  | ATSP-100 | CVRP-100 |
>     | ----------------------- | -------- | -------- | -------- |
>     | vanilla RL              | 5.7      | 23.7     | 9.1      |
>     | CORectifier (ours)      | **17.4** | **34.4** | **49.3** |
>
> These observations demonstrate that rectification **reshapes the reward distribution to be more diverse**, thereby improving the learning signal $A = reward - \mathbb{E}[reward]$.
>
> **2. How Often Rectification Fails, Grouped by Step ($t$).** To quantify this, we conduct a *per-step feasibility analysis* measuring the percentage of steps where a rectified action remains valid (i.e., unvisited). Please refer to the full illustration presented in **Appendix F.7 (Fig. 11) in the [revised manuscript](https://openreview.net/pdf?id=xniQIl8oTw)** for more details. **Key findings across tasks and sizes are:**
> * **TSP/CVRP maintain ~50% feasibility across all steps**, even at large scales (100–500).
> * **ATSP maintains >90% feasibility**, due to its asymmetric structure that limits repeated-node conflicts.
> * **Feasibility curves decline only moderately toward late steps**, confirming that later-stage rectification is *not* significantly harder.
>
> We also compute average feasibility rates to provide you with an overview on this point:
>
> | TSP-50 | TSP-100 | TSP-500 |
> | ------ | ------- | ------- |
> | 54.33% | 50.10%  | 51.36%  |
>
> | ATSP-50 | ATSP-100 | ATSP-200 | ATSP-500 |
> | ------- | -------- | -------- | -------- |
> | 92.62%  | 92.21%   | 91.47%   | 90.05%   |
>
> | CVRP-50 | CVRP-100 | CVRP-200 | CVRP-500 |
> | ------- | -------- | -------- | -------- |
> | 51.31%  | 50.33%   | 48.32%   | 50.62%   |
>
> Importantly, the rectification probability we set (0.1 for TSP/CVRP, 0.5 for ATSP) is *below* the measured feasibility ceiling, meaning that **actual rectification attempts rarely fail** and that **the rectifier’s effect on training is stable and controlled**.
>
> **3. Summary**
>
> The supplementary analyses provide a detailed view of the rectifier’s operation:
>
> * Rectification produces **richer and higher-entropy reward/advantage distributions**, improving policy learning.
> * The **step-wise feasibility** confirms that rectification remains effective throughout the whole trajectory, even for late-step decisions.
> * The overall feasibility is high across all tasks and scales, validating our rectification schedule and hyperparameter design.
>
> We believe these additions address the reviewer’s concerns and clarify the mechanism behind the rectifier’s contribution to CORectifier’s performance.

---

> ### Author Response · Authors · 2025-11-17
> **Author Response to Reviewer sQuT (Page 8, end)**
>
> ### **References**
>
> [1] An Efficient Graph Convolutional Network Technique for the Travelling Salesman Problem.
>
> [2] Graph Neural Network Guided Local Search for the Travelling Salesperson Problem.
>
> [3] Generalize a Small Pre-trained Model to Arbitrarily Large TSP Instances, AAAI 2021
>
> [4] Preference optimization for combinatorial optimization problems, ICML 2025
>
> [5] Attention, Learn to Solve Routing Problems! ICLR 2019
>
> [6] POMO: Policy Optimization with Multiple Optima for Reinforcement Learning, NeurIPS 2020
>
> [7] MatNet: Matrix Encoding Networks for Neural Combinatorial Optimization, NeurIPS 2021
>
> [8] Sym-NCO: Leveraging Symmetricity for Neural Combinatorial Optimization, NeurIPS 2022
>
> [9] MVMoE: Multi-Task Vehicle Routing Solver with Mixture-of-Experts, ICML 2024
>
> [10] Learning Collaborative Policies to Solve NP-hard Routing Problems, NeurIPS 2021
>
> [11] UniCO: On Unified Combinatorial Optimization via Problem Reduction to Matrix-Encoded General TSP, ICLR 2025
>
> [12] UCPO: A Universal Constrained Combinatorial Optimization Method via Preference Optimization, AAAI 2026
>
> [13] DualOpt: A Dual Divide-and-Optimize Algorithm for the Large-scale Traveling Salesman Problem, AAAI 2025
>
> [14] Unify ML4TSP: Drawing Methodological Principles for TSP and Beyond from Streamlined Design Space of Learning and Search, ICLR 2025
>
> [15] BOPO: Neural Combinatorial Optimization via Best-anchored and Objective-guided Preference Optimization, ICML 2025
>
> [16] DIFUSCO: Graph-based Diffusion Solvers for Combinatorial Optimization, NeurIPS 2023
>
> [17] T2T: From distribution learning in training to gradient search in testing for combinatorial optimization, NeurIPS 2023
>
> [18] Fast T2T: Optimization Consistency Speeds Up Diffusion-Based Training-to-Testing Solving for Combinatorial Optimization, NeurIPS 2024
>
> [19] COExpander: Adaptive Solution Expansion for Combinatorial Optimization, ICML 2025
>
> [20] GOAL: A Generalist Combinatorial Optimization Agent Learner, ICLR 2025
>
> [22] BQ-NCO: Bisimulation quotienting for efficient neural combinatorial optimization, NeurIPS 2023
>
> ---
> Again, we sincerely appreciate your valuable time and thoughtful review. We hope that our responses, together with the substantial supplementary experiments provided during the rebuttal, have adequately addressed your questions and resolved ambiguities. **We would be more than grateful if our clarifications help convey the contribution we aim to bring to the community, and we thank you in advance, if possible, for your kind reconsideration and any positively updated assessment**. We greatly appreciate your recognition and look forward to your feedback!
>
>
> Sincerely,
>
> Authors of Submission 11119

---

### Official Review · Reviewer_1exv · 2025-10-26

**Soundness:** 2
**Presentation:** 2
**Contribution:** 2
**Rating:** 2
**Confidence:** 4

**Summary:**

This paper proposes CORectifier as a method to address the challenge in Neural Combinatorial Optimization (NCO), where it becomes increasingly difficult to explore better solutions as training progresses. The authors present an approach that refines the exploration results of reinforcement learning (RL) using optimal solutions, and then utilizes the refined trajectories for further model training.

Specifically, the proposed training process operates as follows:
First, a batch is constructed by mixing problems with and without known optimal solutions. Then, among the multiple trajectories explored in parallel, a subset is selected. The selected trajectories are subsequently improved using the optimal solutions and masked rectifiers, and the improved trajectories are used to train the model.

Experiments are conducted on TSP, ATSP, and PCTSP, and the proposed method is compared with various heatmap-based and sequential decision methods.

**Strengths:**

- The paper introduces a novel training approach that refines parallelly generated trajectories using optimal solutions.

- The idea of leveraging optimal solution information to improve the quality of RL-based exploration is innovative and holds potential to enhance both training stability and convergence speed.

**Weaknesses:**

- The proposed method can only be applied when an oracle solver is available, since it relies on optimal solutions during training. However, if such a solver exists, the need for Neural Combinatorial Optimization (NCO) diminishes; conversely, when no oracle solver is available, the proposed approach cannot be applied.

- The method appears to be limited to TSP-type problems (i.e., problems involving Hamiltonian path finding) and may not generalize well to other types of combinatorial optimization tasks.

- In the experimental results, the method shows inferior performance compared to heatmap-guided methods, particularly as the problem size increases. Moreover, in the comparison with sequential decision methods, there are considerable discrepancies between the reported results in this paper (Table 1 and Table 2) and those presented in the original papers (e.g., TSP-POMO, ATSP-GOAL).

**Questions:**

- Is the RRL Loss presented in Equations (11) and (12) different from the Loss function used in POMO [28]? If so, what are the key differences?

- In Section 3.2.2 (2), is there a specific strategy for selecting the subset $\mathcal{\widehat{T_i}}$? Is it chosen randomly, or by some defined criterion?

---

> ### Author Response · Authors · 2025-11-17
> **Author Response to Reviewer 1exv (Page 1)**
>
> Dear Reviewer 1exv,
>
> Thanks for your valuable review, and we are especially thankful for your recognition on the **novelty of our training approach and its innovative potential to improve the RL4CO field**.
>
> In the following, we sincerely present our detailed clarifications and/or additional experimental results in response to your valuable concerns (W: Weakness, Q: Question, A: Answer). Moreover, we have accordingly updated our manuscript ([click here to download](https://openreview.net/pdf?id=xniQIl8oTw)) that reflects the comprehensive efforts we have done during rebuttal, with major changes marked in red text for your convenient reference. We stay more than willing to reaching an extended consensus with you during the discussion phase!
>
> ---
>
> ### **1. Clarifications on the weaknesses:**
>
> > **W1: Dependency on oracle solvers**
> > The proposed method requires oracle solvers for optimal solutions. If these solvers exist, the need for Neural Combinatorial Optimization diminishes; if they don’t, the approach cannot be applied. How do you justify this dependency?
>
> **A:** Thank you for raising this insightful and dialectical concern. We truly appreciate the opportunity to clarify our motivation and design choices.
>
> 1. **On the role of oracle solvers in NCO research.**
>    - We would like to respectfully emphasize that relying on oracle solvers to generate (near-)optimal reference data is a ***well-established and mainstream practice in the Neural Combinatorial Optimization (NCO) community***. Numerous representative works [1–9, etc.], including many published at top-tier venues, adopt supervised learning (SL) paradigms that depend on oracle-generated labels (just as labeled data are indispensable for advancing modern AI systems in vision, language, and planning domains, etc). In particular, recent *generative CO solvers* [1–4], **despite requiring substantial computational resources and large quantities of optimally labeled data**, have been **widely accepted, recognized, and appreciated** by the machine learning community for their significant contributions.
>
>     - In this context, our use of oracle solvers is fully consistent with the prevailing methodology of the field rather than contradictory to its purpose. While we acknowledge that classical oracle solvers may still dominate in practical engineering contexts, we firmly believe that enabling neural networks to **gradually capture the intrinsic structural patterns of NP-hard problems** and **narrow the performance gap with oracle solvers** represents a meaningful and forward-looking research direction, which contributes to deeper understanding and progress in learning-based combinatorial optimization.
>
> 2. **On mitigating data dependency through efficient design.**
>    That said, we genuinely acknowledge your concern about the potential burden of requiring optimal data. We have been mindful of this issue from the outset of our method design. Specifically, as stated in Line 289 of the paper, our *sub-instance-level rectification* allows each optimal solution $\tau^*$ to be reused multiple times across trajectories during training. This hierarchical design ***improves data utility and reusability***, enabling efficient supervision even with limited oracle data, which is unlike traditional SL approaches that require one-to-one global correspondence between labels and instances.
>
> 3. **On empirical validation of data efficiency.**
>    To further address this concern, we conducted additional quantitative experiments to analyze how our method behaves under varying quantity of expert data. Given the limited rebuttal timeline, we evaluated representative settings: TSP-100, ATSP-100, and CVRP-100, under three levels of expert data: **1.28k**, **12.8k**, and **128k** trajectories, training each model for **200 epochs** with all other hyperparameters fixed default.
>
> * **TSP-100**
>
> | Expert Data | Obj.↓  |
> | ----------- | ------ |
> | 128k        | **7.9559** |
> | 12.8k       | 7.9562 |
> | 1.28k       | 7.9738 |
>
> * **ATSP-100**
>
> | Expert Data | Obj.↓  |
> | ----------- | ------ |
> | 128k        | **1.6920** |
> | 12.8k       | 1.6925 |
> | 1.28k       | 1.6973 |
>
> * **CVRP-100**
>
> | Expert Data | Obj.↓   |
> | ----------- | ------- |
> | 128k        | **16.3118** |
> | 12.8k       | 16.3127 |
> | 1.28k       | 16.3132 |
>
>    The results demonstrate that CORectifier maintains strong performance even when the amount of oracle supervision is substantially reduced, confirming its robustness and data efficiency.
>
> In summary, while our approach leverages oracle solvers for initial supervision (as do many established NCO frameworks), it does so in a way that maximizes data utility, minimizes dependency, and preserves scalability. We sincerely hope this explanation clarifies our standpoint and that the reviewer recognizes our balanced view on combining classical optimization wisdom with modern learning-based advances.

---

> ### Author Response · Authors · 2025-11-17
> **Author Response to Reviewer 1exv (Page 2)**
>
> > **W2: Application to other problem types**
> > The method seems limited to TSP-type (Hamiltonian path) problems. Can it generalize to other types of combinatorial optimization tasks?
>
> **A:** Thank you much for this important concern!
> First, we would like to emphasize that the routing problem family is a well-established and widely recognized research direction within the Neural Combinatorial Optimization (NCO) community. **Numerous influential works published in top-tier venues have focused exclusively on routing problems [10-21, etc.], and their contributions are broadly regarded as foundational rather than limited in scope.**
>
> From a technical standpoint, our hierarchical rectification mechanism is particularly well-suited to routing tasks because of their *geometric nature*, which involves rich topological connectivity and spatially ordered dependencies. Although these problems possess well-structured constraints, as you rightly note, their intricate spatial correlations make them especially challenging for neural models to learn effectively. This complexity renders them ideal testbeds for demonstrating the benefits of our intra-instance-level rectification. In contrast, problems such as the Maximum Independent Set (MIS) or Knapsack Problem (KP) exhibit permutation-invariant objectives, where the order of element selection does not affect the final outcome. Therefore, our initial experiments concentrate on routing problems, where local order and spatial structure play pivotal roles.
>
> That said, we fully recognize the importance of assessing generality beyond tasks with relatively simple constraints. To this end, we have extended our experiments to two additional combinatorial optimization problems with non-trivial constraints: the **Capacitated Vehicle Routing Problem (CVRP)** and the **Knapsack Problem (KP)**.
>
> - **For CVRP**, we leverage a structural property of optimal solutions that the global route can be decomposed into multiple sub-tours. Accordingly, intra-instance-level rectification is applied to optimal sub-trajectories, specifically the route segment between a vehicle’s departure from and return to the depot. The decision to inject such an optimal sub-tour (with probability $p_{\text{sub}}$, instead of the $[\alpha, \beta]$ interval for TSP) is made only when the agent is positioned at the depot. A simplified toy illustration for the sub-instance level rectification mask (as introduced in Eq.(9) in the paper) is:
> $$ \mathcal{M}\_{G_i} = \begin{pmatrix} 1\cdots1, 0\cdots0,\cdots, 1\cdots1 \\\\ 1\cdots1, 0\cdots0,\cdots, 0\cdots0 \\\\ \cdots \\\\ 0\cdots0,1\cdots1,\cdots,1\cdots1 \end{pmatrix}_{N\times M'}, $$
>   where contiguous blocks of ones denote sub-trajectories selected for rectification. (N: number of samples per instance, $M'$: length of the decision trajectories)
>
> - **For KP**, item selection is inherently order-independent. Since the intra-instance rectifier is designed to capture sequential relationships in locally ordered contexts, it is disabled for KP training. Instead, when a sampled trajectory is masked as *True* for instance-level rectification, the entire trajectory is replaced by a randomly permuted sequence that includes all items from the optimal reference solution.
>
> ### **a. Results for uniform CVRP-50/100:**
> (LS: classic local search for CVRP, proposed in [4])
>
> | **Method**| **Type**| **CVRP-50 (10k inst.)** ||| **CVRP-100 (10k inst.)** |||
> |:--- |:-- |:---:|:----------:|:---------:|:--:|:----------:|:---------:|
> |||**Obj.↓**| **Drop↓**  | **Time↓** |**Obj.↓**| **Drop↓**  | **Time↓** |
> | HGS| Heuristics |10.366*|0.000%   |  1.005s|15.563*|   0.000%|  20.027s  |
> | *Heatmap-Guided Methods*||||||||
> | Fast-T2T| SL+G|12.640|  21.835%|  0.009s|19.202|  23.333%   |  0.010s   |
> | COExpander| SL+G|11.979|  15.407%   |  0.033s   |17.497|  12.343%   |  0.047s|
> | Fast-T2T + LS| SL+G+LS|10.871|   4.836%   |  0.013s   |16.294|   4.698%   |  0.018s   |
> | COExpander + LS| SL+G+LS|10.773|   3.903%   |  0.037s   |16.224|   4.253%   |  0.055s|
> | *Sequential Decision Methods* ||||||||
> | AM| RL+G|10.98|   5.86%|–|16.80|   7.34%    |–|
> | POMO| RL+G|10.74|   3.52%    |–|16.15|3.00%    |–|
> | Sym-NCO| RL+G|10.769|   3.891%   |  0.087s   |16.220|   4.241%|  0.166s   |
> | GOAL| SL+G|10.906|   5.193%   |  0.504s   |16.342|   5.005%   |  0.962s   |
> | **CORectifier (ours)**| RRL+G|**10.540**| **1.668%** |  0.042s   |        **15.939**        | **2.425%** |  0.079s|
> | **CORectifier ($N_A=16$)**| RRL+G       |**10.447**| **0.768%** |  0.049s   |        **15.799**| **1.521%** |  0.108s   |
> | Sym-NCO + LS| RL+G+LS|10.505|1.910%   |  0.168s   |15.933|   2.379%   |  0.173s   |
> | GOAL + LS| SL+G+LS|10.628|2.519%|  0.507s   |15.959|   2.548%   |  0.969s|
> | **CORectifier + LS (ours)**| RRL+G+LS|**10.469**| **0.984%** |  0.045s   |**15.796**| **1.496%** |  0.085s   |
> | **CORectifier + LS ($N_A=16$)**  | RRL+G+LS|**10.412**| **0.437%** |  0.052s   |**15.706**| **0.919%** |  0.113s|

---

> ### Author Response · Authors · 2025-11-17
> **Author Response to Reviewer 1exv (Page 3)**
>
> ### **b. Results for uniform CVRP-200/500:**
>
> | **Method**| **Type**   | **CVRP-200 (100 inst.)** ||| **CVRP-500 (100 inst.)** |||
> |:--- |:------- |:--------:|:------:|:---------:|:-----:|:--------:|:---------:|
> |||**Obj.↓**| **Drop↓** | **Time↓** |**Obj.↓**| **Drop↓** | **Time↓** |
> | HGS| Heuristics |19.630*|  0.000%|  60.024s  |37.154*|  0.000%   | 360.376s  |
> | *Heatmap-Guided Methods*||||||||
> | Fast-T2T| SL+G|25.064|  27.616%  |  0.059s   |47.749|  28.509%  |  0.091s   |
> | COExpander| SL+G|22.402|  13.977%  |  0.145s   |43.901|  18.199%  |  0.554s   |
> | Fast-T2T + LS| SL+G+LS|20.662|  5.290%|  0.063s|39.195|  5.530%|  0.215s   |
> | COExpander + LS| SL+G+LS|20.587|  4.893%   |0.153s|39.121|  5.337%   |  0.605s|
> | *Sequential Decision Methods*||||||||
> | Sym-NCO| RL+G|20.662|  5.274%|  0.320s   |40.382|  8.723%|  0.769s   |
> | **CORectifier (ours)**|RRL+G|**20.270**|**3.260%**|0.159s|**39.129**|**5.343%**|0.368s|
> | **CORectifier ($N_A=16$)**| RRL+G|**20.129** |**2.541%**|0.219s|**38.874**|**4.650%**|1.485s|
> | Sym-NCO + LS| RL+G+LS|20.193|  2.880%   |  0.341s   |38.700|4.173%|  0.883s   |
> | **CORectifier + LS (ours)**| RRL+G+LS   |**20.032**|**2.052%**| 0.171s|**38.329**|**3.185%**|0.461s|
> | **CORectifier + LS ($N_A=16$)**  | RRL+G+LS   |**19.952**|   **1.638%**   |0.237s|**38.176**|**2.758%**|1.567s|
>
> ### **c. Zero-shot generalization results on CVRPLib:**
>  (1000 instances total: 22 within (100, 200] clients and 78 within (203, 1000] clients. Results quoted from PO4CO[1].)
>
> | **Solver**| **Type**| **(0, 200] Drop↓** | **(200, 1000] Drop↓** | **Total Drop↓** |  **Time**  |
> | :----- | :--- | :---: | :----: | :--: | :--------: |
> | LKH3| Heuristic|0.36 %|1.18 %|1.00 %|    16 m    |
> | HGS| Heuristic|0.01 %|0.13 %|0.11 %|    16 m    |
> | NeuroLKH| Heuristic + SL |0.47 %|1.16 %|0.88 %|    16 m    |
> | LEHD| SL|11.11 %|12.73 %|12.25 %|1.67 s   |
> | BQ-NCO| SL|10.60 %|10.97 %|10.89 %|   3.36 s   |
> | POMO| RL|5.26 %|11.82 %|10.37 %|   0.80 s   |
> | Sym-NCO| RL|9.99 %|27.09 %|23.32 %|   0.87 s   |
> | Omni-POMO| RL|5.04 %|6.95 %|6.52 %|   0.75 s   |
> | ELG (RF)| RL|4.51 %|6.46 %|6.03 %|   1.90 s   |
> | ELG (PO)| RL|4.39 %|6.37 %|5.94 %|   1.90 s   |
> | **CORectifier (ours)** | RRL|**3.49 %**|**5.98 %**|**5.47 %**| 2.11 s |
>
> ### **d. Results for KP-50/100/200/500:**
>
> | **Method**| **Type** | **KP-50 (1280 inst.)** || **KP-100 (1280 inst.)** || **KP-200 (1280 inst.)** || **KP-500 (1280 inst.)** ||
> |:--|:---|:---:|:--:|:----:|:--:|:---:|:--:|:--:|-|
> |||**Obj.↑**| **Drop↓**|**Obj.↑**| **Drop↓**  |**Obj.↑**        | **Drop↓**  |**Obj.↑**| **Drop↓**  |
> | OR-Tools| Exact    |20.021*|   0.000%   |40.302*|   0.000%   |57.402*|   0.000%   |91.128*| 0.000%|
> | AM| RL+G|–|   0.173%|–|   0.211%   |–|   0.325%   |–| –|
> | POMO| RL+G|–|   0.130%   |–|   0.190%   |–|   0.500%   |–| 6.410%     |
> | BQ-NCO| SL+G|–|–|–|   0.100%   |–|   0.140%   |–| 0.740%     |
> | GOAL| SL+G|–|–|–|   0.120%   |–|   1.630%   |–| 2.400%     |
> | **CORectifier (ours)** | RRL+G|**20.018**| **0.013%** |**40.298**| **0.012%** |**57.397**| **0.009%** |**91.121**| **0.007%** |
>
> These additional experiments demonstrate that CORectifier is not confined to routing tasks. Rather, it can flexibly adapt its rectification strategy to different problem structures, including non-sequential and constraint-heavy combinatorial optimization settings.
>
> We have also included a detailed problem formulation (**Appendix B**), data generation procedure (**Sec. 4.1**), comprehensive experimental results (**Tables 5, 8, 9, 15, 16, 19**), and model configuration details (**Table 13**) for these supplementary task evaluations in the [revised manuscript](https://openreview.net/pdf?id=xniQIl8oTw), ensuring full narrative consistency with the main paper structure. We sincerely hope that these extended studies help alleviate your concern on this matter.

---

> ### Author Response · Authors · 2025-11-17
> **Author Response to Reviewer 1exv (Page 4)**
>
> > **W3: Comparative performance**
> > i) The method performs worse than heatmap-guided methods as problem size increases and ii) shows discrepancies compared to reported results of baselines like TSP-POMO or ATSP-GOAL. Could you explain these inconsistencies?
>
> **A:** Thank you for this valuable question. We would like to provide a detailed clarification.
>
> For **i) the comparison with heatmap-guided methods**:
>
> * **First**, at a broader level, we openly acknowledge the existing performance gap between RL-based sequential solvers and heatmap-guided supervised predictive methods as problem size increases (lines 458–460). This observation is consistent with the general trend observed in the NCO literature. Even the most recent RL-based innovations **published this year [10,18,19]** have primarily evaluated their methods on instances no larger than TSP/CVRP-100 and conducted comparisons mainly within the constructive family, which is consistent with **earlier mainstream works [6,10-19, etc]**. So, **in the background where it is conventionally accepeted that RL4CO solvers need only to be tested on small sizes and compared within the RL family**, our goal has largely extended beyond competing for a “SOTA” title among the sequential solvers; rather, we hope to offer a **holistic understanding of where RL-based solvers stand in the broader NCO ecosystem** and to provide actionable insights for their future evolution. **We therefore respectfully believe that such transparency is academically constructive and should not be viewed as a shortcoming merely because it does not surpass supervised counterparts in absolute metrics on certain benchmarks.**
>
> * **Technically**, this performance disparity has long been a known challenge in the RL4CO community. RL methods inherently suffer from inefficient exploration in high-dimensional combinatorial spaces, which limits their scalability and optimality. Our proposed CORectifier framework directly targets this issue by incorporating ***limited, hierarchically structured expert guidance*** across three levels. This design helps RL solvers develop stronger optimality awareness and improves training efficiency, leading to substantial scalability gains, especially evident in results on test sets such as TSP/ATSP/CVRP-200/500, where CORectifier achieves significantly smaller performance drops compared to existing RL-based solvers. **While we acknowledge that instances with 200–500 nodes may still be considered moderate in scale for heatmap-based solvers, they represent a far more challenging domain for RL4CO solvers** due to the inherent difficulty of sequential decision-making in large combinatorial spaces. **We therefore regard our progress as an important step toward narrowing this gap and advancing the frontier of scalable, sequential RL-based optimization.**
>
> * **Moreover**, as Reviewer xVoR insightfully mentioned, **TSP and ATSP are highly structured routing problems, on which heatmap-based methods comfortably excel, yet these approaches often struggle to generalize to tasks with more complex or heterogeneous constraints**. Our empirical results (**Tables 4, 5, and 9 in the [revised version](https://openreview.net/pdf?id=xniQIl8oTw)**) further support this observation: few (if any) heatmap-guided baselines can be applied to such settings or produce promising results as they do on TSPs. In contrast, RL-based approaches, owing to their sequential decision-making paradigm, are inherently more flexible and can better adapt to diverse formulations, including PCTSP, CVRP, and KP.
>
> * **In summary**, while we acknowledge the performance gap with heatmap-guided methods on certain large, structured problems, we believe that CORectifier represents a meaningful step forward in both concretely improving RL's scalability and performance for CO solving, and strengthening the potential to handle more complex, constraint-rich combinatorial problems where purely and globally supervised approaches tend to fail.

---

> ### Author Response · Authors · 2025-11-17
> **Author Response to Reviewer 1exv (Page 5)**
>
> For **ii) the discrepancy in the reported results of TSP-POMO and ATSP-GOAL**, the primary reason lies in the fact that the original papers did **not fully open-source their test datasets**, which has led to inconsistent reporting across subsequent RL4CO studies.
>
> * **For TSP-POMO**, we observe notable variations in reported performance across different works. Taking the comparative performance between POMO and Sym-NCO for instance:
>
>   – In the original paper, POMO reports a **1.07%** optimality gap on TSP-100 under *greedy decoding*.
>
>   – In **Sym-NCO**[14], the same setting yields a **1.04%** gap for POMO versus **0.94%** for Sym-NCO.
>
>   – In **BQ-NCO**[9], the authors report POMO (with *8× augmentation*) achieving **0.13%**, compared to Sym-NCO (with *100× sampling*) at **0.64%**, and *no absolute objectives are provided*.
>
>   – In **PO4CO**[10], POMO is reported at **0.15%** versus Sym-NCO at **0.39%**, again without revealing absolute objective values or whether any post-inference search procedures were applied.
>
>     Across much of the earlier literature, there is a lack of **standardized evaluation protocols**, including differences in greedy vs. augmented decoding, sampling levels, and undisclosed post-processing, and many works did **not release the exact datasets** used in their experiments. This has resulted in difficulty for aligning the comparative landscapes. To ensure *fairness*, *reproducibility*, and *comparability*, we follow **ML4TSP** [21], a unified benchmark that evaluates mainstream NCO solvers on publicly available, standardized datasets. In our experiments, we faithfully reproduce the results reported in ML4TSP and quote them accordingly. Specifically, for TSP-POMO, we use **1280 uniform TSP instances** for TSP-50/100 and **128 instances** for TSP-500, all under **greedy decoding with no test-time sampling or augmentation**.
>
> - **For ATSP-GOAL,** we observe a similar issue: the original paper reports only *relative gap metrics* for both the baselines and their proposed method, and the test instances used in evaluation are **not publicly released**. To ensure fair and reproducible comparison, we therefore rely on an **established open-source graph-CO benchmark** [22], which provides unified test datasets and standardized implementations. All results we report are obtained strictly to ensure that our metrics are **fully reproducible, transparently comparable**, and aligned with best practices in machine learning research. We believe such unified evaluation settings are essential and should be encouraged across the NCO field.
>
> Thank you again for raising this important point. We hope that our clarifications help alleviate your concerns regarding the empirical comparison protocol used in our study.

---

> ### Author Response · Authors · 2025-11-17
> **Author Response to Reviewer 1exv (Page 6, end)**
>
> ### **2. Responses to the questions:**
>
> > **Q1: Difference in loss formulation**
> > Is the RRL Loss (Equations 11–12) different from the loss function used in POMO [28]? If so, what are the key differences?
>
> Here is a polished and clearer version of your answer:
>
> ---
>
> **A:** Thank you for the question. Our loss formulation is largely consistent with that of POMO in terms of how the policy gradient and baseline are computed. Specifically, as in POMO (line 8 of Algorithm 1, page 5 in its original paper), we use a shared baseline
>
> $$b(G_i) = \frac{1}{N} \sum_{j'=1}^N R_{i,j'}$$
> to compute the advantage ($N$ trajectories sampled for instance $G_i$)
> $$ A_{i,j} = R_{i,j} - b(G_i).$$
>
> The key difference lies in the **log-probability term** used in the policy-gradient loss. In POMO, the loss is computed using
> $$ \log \pi_\theta(\tau \mid G), $$
> i.e., the log-probability of the *original, freely explored trajectory*.
>
> In contrast, our RRL loss uses
>
> $$\log \pi_\theta(\tau' \mid G),$$
>
> where $\tau' = \mathcal{R}(\tau, \tau^*_G, \mathcal{M})$ is the **rectified trajectory** produced by our hierarchical rectification module. That is, the policy is trained toward trajectories that integrate expert-guided corrections while still preserving exploration diversity.
>
> > **Q2: Selection strategy for rectified subsets**
> > In Section 3.2.2 (2), how is the subset $\widehat{T}_i$ selected? Randomly or by a defined criterion?
>
> **A:** The subset $\widehat{T}_i$ is randomly selected without any tricky heuristics, ensuring that all sampled trajectories for a given instance have an equal chance of being rectified and thereby preserving diversity across samples. Thank you for raising this point, and we have accordingly clarified this selection rule (though minor) in the [revised paper](https://openreview.net/pdf?id=xniQIl8oTw) (line 274).
>
> ---
> ### References
> ***1. Well-acknowledged supervised approaches involving oracle solvers:***
>
> [1] DIFUSCO: Graph-based Diffusion Solvers for Combinatorial Optimization, NeurIPS 2023
>
> [2] T2T: From distribution learning in training to gradient search in testing for combinatorial optimization, NeurIPS 2023
>
> [3] Fast T2T: Optimization Consistency Speeds Up Diffusion-Based Training-to-Testing Solving for Combinatorial Optimization, NeurIPS 2024
>
> [4] COExpander: Adaptive Solution Expansion for Combinatorial Optimization, ICML 2025
>
> [5] An Efficient Graph Convolutional Network Technique for the Travelling Salesman Problem.
>
> [6] Graph Neural Network Guided Local Search for the Travelling Salesperson Problem.
>
> [7] GOAL: A Generalist Combinatorial Optimization Agent Learner, ICLR 2025
>
> [8] Generalize a Small Pre-trained Model to Arbitrarily Large TSP Instances, AAAI 2021
>
> [9] BQ-NCO: Bisimulation quotienting for efficient neural combinatorial optimization, NeurIPS 2023
>
> ***2. Well-acknowledged learning methods that 1) mainly focus on routing problems or 2) evaluate instances with no more than size=100:***
>
> [10] Preference optimization for combinatorial optimization problems, ICML 2025
>
> [11] Attention, Learn to Solve Routing Problems! ICLR 2019
>
> [12] POMO: Policy Optimization with Multiple Optima for Reinforcement Learning, NeurIPS 2020
>
> [13] MatNet: Matrix Encoding Networks for Neural Combinatorial Optimization, NeurIPS 2021
>
> [14] Sym-NCO: Leveraging Symmetricity for Neural Combinatorial Optimization, NeurIPS 2022
>
> [15] MVMoE: Multi-Task Vehicle Routing Solver with Mixture-of-Experts, ICML 2024
>
> [16] Learning Collaborative Policies to Solve NP-hard Routing Problems, NeurIPS 2021
>
> [17] UniCO: On Unified Combinatorial Optimization via Problem Reduction to Matrix-Encoded General TSP, ICLR 2025
>
> [18] UCPO: A Universal Constrained Combinatorial Optimization Method via Preference Optimization, AAAI 2026
>
> [19] BOPO: Neural Combinatorial Optimization via Best-anchored and Objective-guided Preference Optimization, ICML 2025
>
> [20] DualOpt: A Dual Divide-and-Optimize Algorithm for the Large-scale Traveling Salesman Problem, AAAI 2025
>
> ***3. Open-source benchmarks that we adopt in part to support our evaluations:***
>
> [21] Unify ML4TSP: Drawing Methodological Principles for TSP and Beyond from Streamlined Design Space of Learning and Search, ICLR 2025
>
> [22] ML4CO-Bench-101: Benchmark Machine Learning for Classic Combinatorial Problems on Graphs, NeurIPS 2025
>
> ---
>
> Again, we sincerely appreciate your valuable time and thoughtful review. We hope that our responses and the additional experiments provided during the rebuttal have helped clarify the technical contributions and address your concerns. **If our clarifications have alleviated earlier doubts, we would be more than grateful for your kind reconsideration of our work toward a more positive assessment**. We deeply respect your judgment and sincerely anticipate your reply!
>
> Sincerely,
>
> Authors of Submission 11119

---

### Official Review · Reviewer_xVoR · 2025-10-30

**Soundness:** 2
**Presentation:** 4
**Contribution:** 2
**Rating:** 2
**Confidence:** 4

**Summary:**

This paper presents CORectifier, an extension of reinforcement learning (RL) methods for solving combinatorial optimization problems (COPs). The key idea is to incorporate expert solutions (reference trajectories from oracle solvers) into the RL training process in a systematic way. Specifically, the proposed Rectified Reinforcement Learning (RRL) framework probabilistically replaces parts of the model’s predicted trajectories with expert segments, enabling the training procedure to benefit from both reinforcement learning (exploration-based optimization) and supervised/imitation learning (expert guidance). The method introduces a hierarchical rectification mechanism operating at batch, instance, and sub-instance levels, effectively balancing exploration and guidance. Empirical results on several COP benchmarks (TSP, ATSP, and PCTSP) show that CORectifier consistently outperforms prior RL- and SL-based baselines, achieving improved sample efficiency, stability, and scalability.

**Strengths:**

The paper is mathematically well-structured and takes care to clearly define all variables and formulations, which improves readability and rigor.

The proposed idea of combining supervised (imitation) and reinforcement learning is conceptually sound and aligns with recent efforts to improve sample efficiency and stability in neural combinatorial optimization.

The method has practical appeal, as many real-world optimization settings can provide partial or full expert solutions that could be leveraged in a similar rectification manner.

**Weaknesses:**

1)
The applicability of the proposed approach appears fundamentally limited to relatively simple combinatorial optimization problems such as TSP and ATSP, which involve minimal or well-structured constraints. The method’s core mechanism—replacing segments of a policy-generated trajectory with expert subsequences—implicitly assumes that any inserted segment remains feasible within the overall solution. However, in more realistic or constraint-heavy problems (e.g., CVRP, VRPTW, scheduling or assignment tasks with capacity or time-window constraints), such partial replacements can easily violate global feasibility, leading to invalid solutions or an extremely low feasible-sampling rate during training. This structural fragility makes the method difficult to extend beyond toy-like routing benchmarks. The paper does not discuss strategies for preserving feasibility under complex constraints, nor does it include experiments on problems with non-trivial feasibility conditions. As a result, the proposed approach seems best suited for simplified academic benchmarks rather than real-world CO applications.

2)
The paper would benefit from a stronger empirical validation of its claimed ability to leverage expert data. For instance, an ablation experiment varying the number of expert trajectories (e.g., 1K, 10K, 100K) could help demonstrate whether the model genuinely improves as more expert supervision becomes available. Such an analysis would clarify whether the reported gains stem from the proposed rectified learning principle or from engineering-heavy interventions such as dynamic scheduling, hyperparameter tuning, and imitation-based warm-up. In its current form, it remains unclear how much of the observed performance improvement can be attributed to the central idea of combining RL and SL, rather than to these auxiliary mechanisms.

**Questions:**

Line 080: in analogy to the language models when the next token grow increasingly distant from the initial state, leading to unstable and inaccurate predictions.
The sentence could be rephrased for clarity. I cannot understand what it means.

---

> ### Author Response · Authors · 2025-11-17
> **Author Response to Reviewer xVoR (Page 1)**
>
> Dear Reviewer xVoR,
>
> Thanks for your precious time reviewing our work, and we are especially grateful for your recognition of our **rigorous presentation, conceptual soundness, and practical appeal**.
>
> In the following, we sincerely present our detailed clarifications and additional experimental results in response to your valuable concerns (W: Weakness, Q: Question, A: Answer). Additionally, we have accordingly updated our manuscript ([click here to download](https://openreview.net/pdf?id=xniQIl8oTw)) that reflects the comprehensive efforts we have done during rebuttal, with major changes marked in red text for your convenient reference. We stay fully positive and committed to reaching an extended consensus with you during the discussion phase!
>
> ---
> ### **1. Clarifications on the weaknesses:**
>
> > **W1: Applicability concern**
> > The method appears limited to relatively simple CO problems (TSP, ATSP) with well-structured constraints. Partial replacement of trajectories may break feasibility in complex tasks like CVRP or scheduling. How can feasibility be maintained under such constraints? Since the approach assumes feasible replacement segments, it may not scale to real-world constraint-heavy problems. Are there strategies to adapt the method to these scenarios?
>
> **A:** Thank you much for this important concern!
> First, we would like to emphasize that the routing problem family is a well-established and widely recognized research direction within the Neural Combinatorial Optimization (NCO) community. Numerous influential works published in top-tier venues have focused *exclusively* on routing problems [1–9, etc.], and their contributions are broadly regarded as foundational rather than limited in scope.
>
> From a technical standpoint, our hierarchical rectification mechanism is particularly well-suited to routing tasks because of their *geometric nature*, which involves rich topological connectivity and spatially ordered dependencies. Although these problems possess well-structured constraints, as you rightly note, their intricate spatial correlations make them especially challenging for neural models to learn effectively. This complexity renders them ideal testbeds for demonstrating the benefits of our intra-instance-level rectification. In contrast, problems such as the Maximum Independent Set (MIS) or Knapsack Problem (KP) exhibit permutation-invariant objectives, where the order of element selection does not affect the final outcome. Therefore, our initial experiments concentrate on routing problems, where local order and spatial structure play pivotal roles.
>
> That said, we fully recognize the importance of assessing generality beyond tasks with relatively simple constraints. To this end, we have extended our experiments to two additional combinatorial optimization problems with non-trivial constraints: the **Capacitated Vehicle Routing Problem (CVRP)** and the **Knapsack Problem (KP)**.
>
> - **For CVRP**, we leverage a structural property of optimal solutions that the global route can be decomposed into multiple sub-tours. Accordingly, intra-instance-level rectification is applied to optimal sub-trajectories, specifically the route segment between a vehicle’s departure from and return to the depot. The decision to inject such an optimal sub-tour (with probability $p_{\text{sub}}$, instead of the $[\alpha, \beta]$ interval for TSP) is made only when the agent is positioned at the depot. A simplified illustration for the sub-instance level rectification mask (as introduced in Eq. (9) in the paper) is:
> $$ \mathcal{M}\_{G_i} = \begin{pmatrix} 1\cdots1, 0\cdots0,\cdots, 1\cdots1 \\\\ 1\cdots1, 0\cdots0,\cdots, 0\cdots0\\\\ \cdots \\\\  0\cdots0,1\cdots1,\cdots,1\cdots1 \end{pmatrix}_{N\times M'}, $$
>   where contiguous blocks of ones denote sub-trajectories selected for rectification. (N: number of samples per instance, $M'$: length of the decision trajectories)
>
> - **For KP**, item selection is inherently order-independent. Since the intra-instance rectifier is designed to capture sequential relationships in locally ordered contexts, it is disabled for KP training. Instead, when a sampled trajectory is masked as *True* for instance-level rectification, the entire trajectory is replaced by a randomly permuted sequence that includes all items from the optimal reference solution.

---

> ### Author Response · Authors · 2025-11-17
> **Author Response to Reviewer xVoR (Page 2)**
>
> ### **a. Results for uniform CVRP-50/100:**
> (LS: classic local search for CVRP, proposed in [10])
>
> | **Method**| **Type**   | **CVRP-50 (10k inst.)** ||| **CVRP-100 (10k inst.)** |||
> |:--- |:-- |:---:|:----------:|:---------:|:--:|:----------:|:---------:|
> |||**Obj.↓**| **Drop↓**  | **Time↓** |**Obj.↓**| **Drop↓**  | **Time↓** |
> | HGS| Heuristics |10.366*|   0.000%   |  1.005s   |15.563*|   0.000%   |  20.027s  |
> | *Heatmap-Guided Methods*||||||||
> | Fast-T2T| SL+G|12.640|  21.835%   |  0.009s   |19.202|  23.333%   |  0.010s   |
> | COExpander| SL+G|11.979|  15.407%   |  0.033s   |17.497|  12.343%   |  0.047s|
> | Fast-T2T + LS| SL+G+LS    |10.871|   4.836%   |  0.013s   |16.294|   4.698%   |  0.018s   |
> | COExpander + LS| SL+G+LS    |10.773|   3.903%   |  0.037s   |16.224|   4.253%   |  0.055s   |
> | *Sequential Decision Methods* ||||||||
> | AM| RL+G|10.98|   5.86%    |–     |16.80|   7.34%    |     –     |
> | POMO| RL+G|10.74|   3.52%    |     –     |16.15|   3.00%    |     –     |
> | Sym-NCO| RL+G       |10.769|   3.891%   |  0.087s   |16.220|   4.241%   |  0.166s   |
> | GOAL| SL+G       |10.906|   5.193%   |  0.504s   |16.342|   5.005%   |  0.962s   |
> | **CORectifier (ours)**| RRL+G|**10.540**| **1.668%** |  0.042s   |        **15.939**        | **2.425%** |  0.079s   |
> | **CORectifier ($N_A=16$)**| RRL+G       |**10.447**| **0.768%** |  0.049s   |        **15.799**        | **1.521%** |  0.108s   |
> | Sym-NCO + LS| RL+G+LS    |10.505|   1.910%   |  0.168s   |          15.933|   2.379%   |  0.173s   |
> | GOAL + LS| SL+G+LS    |10.628|   2.519%   |  0.507s   |          15.959          |   2.548%   |  0.969s   |
> | **CORectifier + LS (ours)**   | RRL+G+LS    |       **10.469**        | **0.984%** |  0.045s   |        **15.796**        | **1.496%** |  0.085s   |
> | **CORectifier + LS ($N_A=16$)**  | RRL+G+LS    |       **10.412**        | **0.437%** |  0.052s   |        **15.706**        | **0.919%** |  0.113s   |
>
> ### **b. Results for uniform CVRP-200/500:**
>
> | **Method**| **Type**   | **CVRP-200 (100 inst.)** ||| **CVRP-500 (100 inst.)** |||
> |:--- |:------- |:--------:|:------:|:---------:|:-----:|:--------:|:---------:|
> |||**Obj.↓**| **Drop↓** | **Time↓** |**Obj.↓**| **Drop↓** | **Time↓** |
> | HGS| Heuristics |19.630*|  0.000%|  60.024s  |37.154*|  0.000%   | 360.376s  |
> | *Heatmap-Guided Methods*||||||||
> | Fast-T2T| SL+G|25.064|  27.616%  |  0.059s   |47.749|  28.509%  |  0.091s   |
> | COExpander| SL+G|22.402|  13.977%  |  0.145s   |43.901|  18.199%  |  0.554s   |
> | Fast-T2T + LS| SL+G+LS|20.662|  5.290%|  0.063s|39.195|  5.530%|  0.215s   |
> | COExpander + LS| SL+G+LS|20.587|  4.893%   |0.153s|39.121|  5.337%   |  0.605s|
> | *Sequential Decision Methods*||||||||
> | Sym-NCO| RL+G|20.662|  5.274%|  0.320s   |40.382|  8.723%|  0.769s   |
> | **CORectifier (ours)**|RRL+G|**20.270**|**3.260%**|0.159s|**39.129**|**5.343%**|0.368s|
> | **CORectifier ($N_A=16$)**| RRL+G|**20.129** |**2.541%**|0.219s|**38.874**|**4.650%**|1.485s|
> | Sym-NCO + LS| RL+G+LS|20.193|  2.880%   |  0.341s   |38.700|4.173%|  0.883s   |
> | **CORectifier + LS (ours)**| RRL+G+LS   |**20.032**|**2.052%**| 0.171s|**38.329**|**3.185%**|0.461s|
> | **CORectifier + LS ($N_A=16$)**  | RRL+G+LS   |**19.952**|   **1.638%**   |0.237s|**38.176**|**2.758%**|1.567s|
>
> ### **c. Zero-shot generalization results on CVRPLib:**
> (1000 instances total: 22 within (100, 200] clients and 78 within (203, 1000] clients. Results quoted from PO4CO[1].)
>
> | **Solver**| **Type**| **(0, 200] Drop↓** | **(200, 1000] Drop↓** | **Total Drop↓** |  **Time**  |
> | :----- | :--- | :---: | :----: | :--: | :------: |
> | LKH3| Heuristic|0.36 %|1.18 %|1.00 %|    16 m    |
> | HGS| Heuristic|0.01 %|0.13 %|0.11 %|    16 m    |
> | NeuroLKH| Heuristic + SL |0.47 %|1.16 %|0.88 %|    16 m    |
> | LEHD| SL|11.11 %|12.73 %|12.25 %|1.67 s   |
> | BQ-NCO| SL|10.60 %|10.97 %|10.89 %|   3.36 s   |
> | POMO| RL|5.26 %|11.82 %|10.37 %|   0.80 s   |
> | Sym-NCO| RL|9.99 %|27.09 %|23.32 %|   0.87 s   |
> | Omni-POMO| RL|5.04 %|6.95 %|6.52 %|   0.75 s   |
> | ELG (RF)| RL|4.51 %|6.46 %|6.03 %|   1.90 s   |
> | ELG (PO)| RL|4.39 %|6.37 %|5.94 %|   1.90 s   |
> | **CORectifier (ours)** | RRL|**3.49 %**|**5.98 %**|**5.47 %**| 2.11 s |
>
> ### **d. Results for KP-50/100/200/500:**
>
> | **Method**| **Type** | **KP-50 (1280 inst.)** || **KP-100 (1280 inst.)** || **KP-200 (1280 inst.)** || **KP-500 (1280 inst.)** ||
> |:--|:---|:---:|:--:|:----:|:--:|:---:|:--:|:--:|-|
> |||**Obj.↑**| **Drop↓**|**Obj.↑**| **Drop↓**  |**Obj.↑**        | **Drop↓**  |**Obj.↑**| **Drop↓**  |
> | OR-Tools| Exact    |20.021*|   0.000%   |40.302*|   0.000%   |57.402*|   0.000%   |91.128*| 0.000%|
> | AM| RL+G|–|   0.173%|–|   0.211%   |–|   0.325%   |–| –|
> | POMO| RL+G|–|   0.130%   |–|   0.190%   |–|   0.500%   |–| 6.410%     |
> | BQ-NCO| SL+G|–|–|–|   0.100%   |–|   0.140%   |–| 0.740%     |
> | GOAL| SL+G|–|–|–|   0.120%   |–|   1.630%   |–| 2.400%     |
> | **CORectifier (ours)** | RRL+G|**20.018**| **0.013%** |**40.298**| **0.012%** |**57.397**| **0.009%** |**91.121**| **0.007%** |

---

> ### Author Response · Authors · 2025-11-17
> **Author Response to Reviewer xVoR (Page 3)**
>
> These additional experiments demonstrate that CORectifier is not confined to routing tasks. Rather, it can flexibly adapt its rectification strategy to different problem structures, including non-sequential and constraint-heavy combinatorial optimization settings.
>
> We have also included a detailed problem formulation (**Appendix B**), data generation procedure (**Sec. 4.1**), comprehensive experimental results (**Tables 5, 8, 9, 15, 16, 19**), and model configuration details (**Table 13**) for these supplementary task evaluations in the [revised manuscript](https://openreview.net/pdf?id=xniQIl8oTw), ensuring full narrative consistency with the main paper structure. We sincerely hope that these extended studies help alleviate your concern on this matter.
>
> > **W2: Effectiveness of expert data**
> > The empirical validation does not isolate how much improvement comes from rectified learning itself versus other engineering factors (dynamic scheduling, tuning, warm-up). Could you include ablation results varying the number of expert trajectories (e.g., 1k, 10k, 100k)?
>
> **A:** Thank you for this constructive suggestion. We fully agree that isolating the contribution of expert data quantity provides a clearer understanding of the rectified learning mechanism. In response, we have conducted additional experiments to study how the amount of expert supervision influences the training efficiency and final performance.
>
> Given the limited rebuttal timeline, we evaluated representative settings: TSP-100, ATSP-100, and CVRP-100, under three levels of expert data: **1.28k**, **12.8k**, and **128k** trajectories, training each model for **200 epochs** with all other hyperparameters fixed default.
>
> * **TSP-100**
>
> | Expert Data | Obj.↓  |
> | ----------- | ------ |
> | 128k        | **7.9559** |
> | 12.8k       | 7.9562 |
> | 1.28k       | 7.9738 |
>
> * **ATSP-100**
>
> | Expert Data | Obj.↓  |
> | ----------- | ------ |
> | 128k        | **1.6920** |
> | 12.8k       | 1.6925 |
> | 1.28k       | 1.6973 |
>
> * **CVRP-100**
>
> | Expert Data | Obj.↓   |
> | ----------- | ------- |
> | 128k        | **16.3118** |
> | 12.8k       | 16.3127 |
> | 1.28k       | 16.3132 |
>
> Across all benchmarks, we observe two consistent patterns:
>
> 1. **More expert data naturally yields mild performance improvements**, as additional expert guidance further stabilizes early training.
> 2. **Even with only 1/100 of the expert data, the performance degradation is very small and no training collapse occurs**, highlighting that CORectifier remains highly stable and effective even under extremely sparse expert supervision.
>
> Overall, these results demonstrate that our proposed rectified learning framework is **robust to the quantity of expert data**, and thus remains practically applicable in scenarios where collecting abundant high-quality expert trajectories may be costly. We hope these ablations clearly demonstrate that the performance gains arise primarily from the rectified learning design rather than from the volume of expert data or auxiliary engineering components.

---

> ### Author Response · Authors · 2025-11-17
> **Author Response to Reviewer xVoR (Page 4, end)**
>
> ### **2. Responses to the questions:**
>
> > **Q1: Clarity of a specific sentence (Line 080)**
> > The sentence “in analogy to the language models when the next token grow increasingly distant from the initial state” is unclear.
>
> **A:** Thank you very much for pointing this out and for giving us the opportunity to clarify. The original sentence was intended to draw an analogy between **reinforcement learning (RL) in long decision horizons** and (early) **language modeling** in autoregressive generation. Specifically, as the decision sequence in RL becomes longer, the agent’s state distribution can drift farther away from the initial (well-learned) regions of the state space, similar to how, in language models, predictions tend to become less reliable as the generated tokens grow increasingly distant from the initial context. This "drift" often leads to reduced stability during long rollouts.
>
> To improve clarity, we have rephrased the sentence in the [revised manuscript](https://openreview.net/pdf?id=xniQIl8oTw) (line 082) as follows:
>
> > "This phenomenon is analogous to autoregressive language models, where prediction quality deteriorates as the generated sequence extends farther from the initial context, causing cumulative errors and instability."
>
> We hope this revision makes our intended meaning clear and improves the readability of the paragraph.
>
> ---
> ### **References**
>
> [1] Attention, Learn to Solve Routing Problems! ICLR 2019
>
> [2] Sym-NCO: Leveraging Symmetricity for Neural Combinatorial Optimization, NeurIPS 2022
>
> [3] MVMoE: Multi-Task Vehicle Routing Solver with Mixture-of-Experts, ICML 2024
>
> [4] Learning Collaborative Policies to Solve NP-hard Routing Problems, NeurIPS 2021
>
> [5] UniCO: On Unified Combinatorial Optimization via Problem Reduction to Matrix-Encoded General TSP, ICLR 2025
>
> [6] Unify ML4TSP: Drawing Methodological Principles for TSP and Beyond from Streamlined Design Space of Learning and Search, ICLR 2025
>
> [7] GLOP: Learning Global Partition and Local Construction for Solving Large-scale Routing Problems in Real-time, AAAI 2024
>
> [8] Unsupervised Learning for Solving the Travelling Salesman Problem, NeurIPS 2023
>
> [9] DualOpt: A Dual Divide-and-Optimize Algorithm for the Large-scale Traveling Salesman Problem, AAAI 2025
>
> [10] COExpander: Adaptive Solution Expansion for Combinatorial Optimization, ICML 2025
>
> ---
> Again, we express our sincere gratitude for your valuable time and thoughtful review. We genuinely hope that our responses have adequately addressed your questions and clarified possible ambiguities or misunderstandings. **We would be deeply appreciative if, upon your reconsideration of our supplementary efforts, a stronger consensus could be reached regarding the contributions our work aims to make to the community**. We remain fully open to further discussion and look forward to your reply.
>
> Sincerely,
>
> Authors of Submission 11119

---

### Official Review · Reviewer_UfUe · 2025-10-30

**Soundness:** 2
**Presentation:** 3
**Contribution:** 2
**Rating:** 4
**Confidence:** 5

**Summary:**

This paper proposes CORectifier, a rectified reinforcement learning (RRL) framework that integrates hierarchical supervision into reinforcement learning to improve sample efficiency and reward sparsity in neural combinatorial optimization (NCO). The proposed method regularizes the exploration process by probabilistically replacing partial policy-generated trajectories with high-quality segments from expert solutions.

**Strengths:**

1. The proposed method has good generality and can be adapted to multiple NCO approaches.
2. The core idea is simple and easy to understand, integrating expert-guided rectification into RL in a straightforward way.
3. The experimental section is rich, covering multiple tasks and baselines.

**Weaknesses:**

1. The motivation is not entirely clear. Although the paper points out three limitations of existing RL-based NCO methods, it does not clearly explain how the proposed method effectively addresses each of them.
2. The advantage of the proposed way to combine RL and imitation learning (IL) is not deeply analyzed. It is intuitive that integrating IL can improve RL, but it is unclear whether the proposed RRL provides substantial benefits over simple integration schemes such as first-IL-then-RL two-stage training or approaches like [1] that use local search to refine sampled rollouts as expert demonstrations during training.
3. During trajectory rectification, feasibility checks are applied, so the actual number of replaced segments is unknown. An analysis of this aspect would help understand the mechanism behind the improvement.
4. The backbone models used for each problem type are not clearly stated in the main text.
5. The proposed framework introduces many hyperparameters, raising concerns about potential tuning difficulty and task-specific parameter dependency.
6. The hyperparameter study lacks convincing justification. For instance, the choice of 0.1 for $p_{batch}$, $p_{inst}$, $\alpha$ and $\beta$ seems arbitrary, and it is unclear how performance changes with smaller values.
7. The technical presentation of the trajectory rectification process (involving masks and mathematical notation) is somewhat difficult to follow, which may hinder understanding of the implementation details.
8. The reference formatting does not fully conform to the ICLR citation style.

[1] Preference optimization for combinatorial optimization problems, ICML 2025.

**Questions:**

1. Are the three hierarchical levels (batch, instance, intra-instance) truly complementary? Could the authors provide ablation results for enabling/disabling each level independently?
2. Is the proposed approach applicable to non-routing combinatorial optimization problems?
3. The definition of $M$ as problem size seems inconsistent. In the Intra-instance Level section, $M$ sometimes appears to represent the length of the decision sequence. Could the authors clarify this definition?

---

> ### Author Response · Authors · 2025-11-17
> **Author Response to Reviewer UfUe (Page 1)**
>
> Dear Reviewer UfUe,
>
> Thanks for your meticulous review and helpful suggestions! In the following, we are more than pleased to present our item-by-item clarifications in response to your concerns. (W: Weakness, Q: Question, A: Answer. Similar issues are summarized and combined.) Moreover, we have accordingly updated our manuscript ([click here to download](https://openreview.net/pdf?id=xniQIl8oTw)) that reflects the comprehensive efforts we have done during rebuttal, with major changes marked in red text for your convenient reference.
>
> ---
> ### **1. Clarifications on the weaknesses:**
>
> > **W1: Motivation clarity**
> > Although the paper lists three limitations of existing RL-based NCO methods, it does not clearly explain how the proposed method effectively addresses each of them.
>
> **A:** Thank you for your thoughtful comment. While the three limitations we highlighted for prior RL4CO methods are inherently interconnected rather than separable, our extensive empirical results (including additional studies conducted during the rebuttal) provide concrete evidence that CORectifier effectively alleviates these long-standing challenges, such as ineffective exploration and poor scalability. For clarity, we summarize below how each limitation is directly addressed by our hierarchical rectification mechanism.
>
> **1. Ineffective training due to sub-optimal exploration**
>
> - **Limitation.** RL4CO quickly collapses to low-entropy, low-quality trajectories, producing weak policy evolving.
>
> - **How CORectifier helps.** Our **instance-level** and **intra-instance** rectifiers inject short expert subsequences into sampled rollouts, at random positions and lengths. This:
>
>     * steers exploration toward high-value regions,
>     * prevents entropy collapse,
>     * improves the quality and separability of policy gradients.
>
>     Following [1], we hereby additionally report the **trajectory entropy comparison** (calculated as the sum of entropy at each step in $\tau$, sampled from the trained models) for your direct reference:
>
>     || TSP-100  | ATSP-100 | CVRP-100 |
>     | - | -------- | -------- | -------- |
>     | vanilla RL         | 5.7      | 23.7     | 9.1      |
>     | CORectifier (ours) | **17.4** | **34.4** | **49.3** |
>
> - **Evidence.** Higher trajectory entropy, clearer advantage separation, and smoother advantage density distribution (see **Fig. 6, 8-10 in our [updated manuscript](https://openreview.net/pdf?id=xniQIl8oTw), Appendix F.6**), indicating significantly improved diversity of the explored strategies.
>
>
> **2. Inefficient sampling caused by sparse end-of-trajectory rewards**
>
> - **Limitation**. RL receives only a single terminal reward after constructing a long action sequence (e.g., a full tour), making the sampling process inefficient and leaving the long-horizon action chain weakly supervised and difficult to interpret.
>
> - **How CORectifier helps.** Rectified partial solutions act as **implicit intermediate supervision**, providing informative guidance at many time steps, which is toward offering local optimality signals on constructing the entire trajectory. The three rectifier levels:
>
>     * **batch-level** ensures each batch contains reliable signals,
>     * **instance-level** controls how many rollouts receive expert guidance,
>     * **intra-instance** provides fine-grained step-level corrections.
>
>     This substantially reduces training instability by **improving the sampling process with stronger optimality awareness**, while providing **finer-grained control over trajectory construction** in a more interpretable manner.
>
> - **Evidence.** Faster convergence and much more stable training curves (e.g., **Fig. 3-4**); consistently strong gains on **all five tasks with different structures and constraints**.
>
> **3. Poor scalability of RL4CO to larger instances**
>
> - **Limitation.** Existing RL approaches degrade severely beyond 100 nodes due to inefficient exploration and weak signals.
>
> - **How CORectifier helps.** By improving exploration quality and reducing variance, CORectifier enables the policy to learn **reusable local structures** and maintain **effective sampling diversity**, which improves scalability.
>
> - **Evidence.** On TSP/ATSP/CVRP with 200–500 nodes, **a scale at which existing RL4CO solvers [2-7] typically avoid evaluation even as of the year of 2025**, CORectifier reduces the large-instance performance gap between RL and strong heatmap-based supervised solvers by **up to 89%**, while maintaining robust cross-size generalization. Moreover, on real-world CVRPLIB instances with 100–1000 customers, CORectifier achieves a state-of-the-art **5.47%** optimality gap, outperforming **seven** recent mainstream RL-based CO solvers.
>
> We hope these clarifications makes clear that CORectifier directly targets and substantially alleviates the fundamental challenges of RL-based NCO methods, thereby strengthening the motivation and necessity of our design.

---

> ### Author Response · Authors · 2025-11-17
> **Author Response to Reviewer UfUe (Page 2)**
>
> > **W2: Integration of RL and IL**
> > The advantage of combining RL and imitation learning (IL) is not deeply analyzed. It remains unclear whether the proposed Rectified RL provides substantial benefits over simpler integration schemes such as two-stage IL-then-RL training or local-search-based methods[1].
>
> **A:** Thanks for your concern. In the initial submission, we have provided illustrative results ablating the training components (shown in **Fig.3-4 and Table 3**), which validates our proposed rectified RL produces independent benefits over vanilla RL and IL practices. Here, we fully appreciate your suggestion to conduct additional ablation experiments on the comparison with 1) IL-then-RL training, and 2) the comparison with local-search-based methods[1] as you mentioned:
> - **Comparative results on IL-then-RL training** (100 epochs):
>
> | TSP-100| Obj.↓| Drop↓|
> |-|-|-|
> | IL-then-RL| 8.091     | 4.321%     |
> | **RRL (ours)** | **8.039** | **3.655%** |
>
> | ATSP-100       | Obj.↓     | Drop↓      |
> |-|-|-|
> | IL-then-RL     | 1.708     | 9.065%     |
> | **RRL (ours)** | **1.668** | **6.504%** |
>
> - **Comparative results with PO[1]** (quoted exact data reported in the original paper on the well-known CVRPLIB benchmark, full results on the CVRP task are detailed in our answer to **Q2** below):
>
> | **Solver**| **Type** | **(0, 200] Drop↓** | **(200, 1000] Drop↓** | **Total Drop↓** |
> |:--|:-|:--:|:-:|:--:|
> | LEHD| SL|11.11 %|12.73 %|12.25 %|
> | BQ-NCO| SL|10.60 %|10.97 %|     10.89 %     |
> | POMO| RL|5.26 %|11.82 %|10.37 %|
> | Sym-NCO| RL|9.99 %|27.09 %|     23.32 %     |
> | Omni-POMO| RL|5.04 %|6.95 %|     6.52 %      |
> | ELG| RL|4.51 %|6.46 %         |6.03 %      |
> | PO[1]| RL|4.39 %       |6.37 %|     5.94 %      |
> | **CORectifier (ours)** | RRL|     **3.49 %**|**5.98 %**|   **5.47 %**    |
>
> These supplementary results together offer a more comprehensive picture of the benefits of our proposed training scheme, demonstrating **clear advantages over vanilla RL, pure IL, and sequential hybrid approaches such as IL-then-RL pipelines or local-search–based refinement methods**.
>
>
> > **W3: Rectification feasibility**
> > During trajectory rectification, feasibility checks are applied, but the actual number of replaced segments is unknown. An analysis of this would help clarify the mechanism behind the improvement.
>
> **A:** Thank you for this insightful question. We agree that understanding *how many rectifications are actually feasible* during training is important for interpreting the effectiveness of our method.
>
> To address this, we conducted an additional study measuring the **per-step feasibility rate** of the rectification operator across three representative tasks (TSP, ATSP, CVRP) and multiple problem sizes. The feasibility rate is defined as the percentage of rectifiable steps where the expert-suggested next action is still valid (i.e., unvisited) given the current partial solution.
>
> **Average Feasibility Rate:**
>
> | TSP-50 | TSP-100 | TSP-500 |
> | ------ | ------- | ------- |
> | 54.33% | 50.10%  | 51.36%  |
>
> | ATSP-50 | ATSP-100 | ATSP-200 | ATSP-500 |
> | ------- | -------- | -------- | -------- |
> | 92.62%  | 92.21%   | 91.47%   | 90.05%   |
>
> | CVRP-50 | CVRP-100 | CVRP-200 | CVRP-500 |
> | ------- | -------- | -------- | -------- |
> | 51.31%  | 50.33%   | 48.32%   | 50.62%   |
>
> **Key Findings：**
>
> 1. **Feasibility remains consistently high across sizes.**
>
>    * TSP and CVRP maintain a stable feasibility rate around **50%**, even at large scales (100–500 nodes).
>    * ATSP shows extremely high feasibility (**over 90%**) due to its asymmetric structure, which naturally restricts repeated visits and thus preserves expert-successor validity.
>
> 2. **Feasibility does *not* degrade with larger instance sizes.**
>    This confirms that the rectifier’s effectiveness is not impaired as the problem grows, and rectifications remain meaningful even in long-horizon trajectories.
>
> 3. **This validates our chosen rectification probabilities.**
>    For example:
>
>    * A low probability (0.1) is chosen for TSP/CVRP because roughly half the steps are feasible, which balances expert guidance without overwhelming exploration.
>    * A higher probability (0.5) is suitable for ATSP, where feasibility exceeds 90%, making rectification highly reliable.
>
>     These results confirm that **the actual number of replaced actions is well-controlled by our set probabilities and rarely blocked by infeasibility** as the gated rectification rate defined by hyperparameters are generally lower than the inherent feasibility rates, supporting the design of our hierarchical rectification levels.
>
> We have also [updated the manuscript](https://openreview.net/pdf?id=xniQIl8oTw) to include a **per-step feasibility curve** (**Fig. 11, Appendix F.7**), providing a more complete picture of the rectification behavior. We hope this analysis clarifies why rectification is consistently feasible and how this contributes to the mechanism behind CORectifier’s performance improvements.

---

> ### Author Response · Authors · 2025-11-17
> **Author Response to Reviewer UfUe (Page 3)**
>
> > **W4: Backbone specification**
> > The backbone models used for each problem type are not clearly stated in the main text.
>
> **A:** Thanks for your comment. The backbone models for each problem type are indeed specified in the *Adaptability Study* paragraph of Section 4.3, with detailed architectural descriptions provided in **Appendix D**. Specifically, we employ **POMO** for TSP/CVRP/KP, **MatNet** for ATSP, and **AM** for PCTSP. These backbones share similar Transformer-based structures and are widely adopted for their respective problem domains, illustrating the flexibility of our framework across different neural architectures. Following your suggestion, we have clarified this information more explicitly in the [revised version](https://openreview.net/pdf?id=xniQIl8oTw) (Lines 527–529). We thank you for pointing out this opportunity to improve the clarity of our presentation.
>
>
> > **W5 & W6: Hyperparameter complexity and justification**
> > i) The framework introduces many hyperparameters, which raises concerns about tuning difficulty and task-specific dependency. ii) The choices for ($p_{batch}, p_{inst}, \alpha, \beta$) appear arbitrary. Please justify these and show how performance varies with smaller values.
>
> **A:** Thank you for raising these thoughtful concerns. We clarify the hyperparameter design and its practical implications as follows.
> 1. **Number and role of hyperparameters.** Although CORectifier introduces four primary hyperparameters governing the three rectification levels (for TSP-related tasks), **this quantity of hyperparameters is modest compared with standard practice in modern ML models**. More importantly, each hyperparameter has a **clear, interpretable role** (batch-level coverage, instance-level rectification frequency, and intra-instance rectification span), and together they form a **minimal yet sufficient** set to control rectification granularity.
> 2. **Justification and sensitivity.** Our initial submission **already included extensive studies on hyperparameter behavior (Fig. 5, Fig. 7, Table 6, Table 17 in our [manuscript](https://openreview.net/pdf?id=xniQIl8oTw))**. As shown there, these hyperparameters are not highly sensitive, and the chosen values were selected through lightweight pre-experiments over a small grid. Thus, the settings are **empirically motivated rather than arbitrary**, and tuning is not demanding in practice.
> 3. **Effect of smaller rectification probabilities.** Please kindly note that we have explicitly remarked in the paper that "***we avoid setting $p_{\text{batch}}$, $p_{\text{inst}}$, $\alpha$, or $\beta$ to values smaller than 0.1 in case the model would degenerate into a vanilla RL paradigm***" (lines 502–504) and that "***guaranteed by its design principle, as the quality of expert data decreases, CORectifier is (in the worst case) degraded to and bounded by vanilla RL's performance***" (lines 524–526). To validate this claim further, we conduct additional experiments on TSP-50 (as continuous for those reported in the paper) for 100 epochs across batch/instance/intra-instance probabilities <0.1.
>
> | $p_\text{batch}$ | Obj.↓     | $p_\text{inst}$ | Obj.↓ | $[\alpha,\beta]$ | Obj.↓ |
> | ---- | -- | -- | ----- | -- | ----- |
> | 0.5| 5.782     | 0.5| 5.796 |[0.5,1.0]| 5.770      |
> | 0.4| 5.776     | 0.4| 5.789 |[0.4,0.5]| 5.778      |
> | 0.3| 5.779     | 0.3| 5.753 |[0.3,0.4]| 5.799      |
> | 0.2| 5.770     | 0.2| 5.773 |[0.2,0.3]| 5.822      |
> | 0.1| **5.765** | 0.1| **5.747** |[0.1,0.2]| **5.752**      |
> | 0.08| 5.769          | 0.08| 5.750      |[0.08,0.1]| 5.754      |
> | 0.06| 5.766          | 0.06| 5.749      |[0.06,0.08]| 5.753       |
> | 0.04| 5.769          | 0.04| 5.750      |[0.04,0.06]| 5.760      |
> | 0.02| 5.770          | 0.02| 5.751      |[0.02,0.04]| 5.755      |
>
> The results (shown above) confirm that:
> - performance peaks around probability $\approx$ 0.1,
> - further lowering rectification probability results in only minor and smooth degradation,
> - extremely small values (<0.05) provide too little supervision to be impactful, which is consistent with our theoretical expectation.
>
> 4. **Flexibility across tasks, not over-parameterization.** The hyperparameters are not rigid and adapt naturally across tasks. For example, In KP, intra-instance rectification can be disabled due to permutation invariance. In CVRP, rectification operates at the *sub-tour* level, replacing $[\alpha,\beta]$ with a single $p_\text{sub}$ (we detail the extension to the KP and CVRP tasks in our response to **Q2** below). This illustrates that our design **prioritizes problem-appropriate control with minimal tuning complexity, instead of introducing unnecessary hyperparameter proliferation.**
>
> We hope these clarifications alleviate your concerns and demonstrate that CORectifier’s hyperparameter design is principled, interpretable, and empirically stable.

---

> ### Author Response · Authors · 2025-11-17
> **Author Response to Reviewer UfUe (Page 4)**
>
> > **W7: Technical presentation**
> > The mathematical notation and description of the rectification process (masks, etc.) are difficult to follow and may hinder reproducibility.
>
> Thank you for your comment. We are happy to clarify the technical details and implementation of the rectifier operator.
>
> For mathematical rigor, we define the rectification operator $\mathcal{R}(\tau,\tau_G^\*, \mathcal{M}) \mapsto \tau'$ in the paper from a **global** perspective, as if the full policy trajectory $\tau$ and the expert trajectory $\tau_G^*$ were both precomputed and then “patched” according to the mask $\mathcal{M}$ (Eq. (10)). This global description was chosen mainly for *expository clarity*, so that the notion of “replacing segments” is easy to follow at a macroscopic level.
>
> However, in **practical training**, the rectification is implemented **online and locally at each step**, not by first generating a full $\tau$ and then post-hoc modifying it. Concretely (taking TSP as an example):
>
> 1. At step $t$, given the current partial tour $\tau_{1:t-1}$, the policy $\pi_\theta$ produces a distribution over feasible next nodes (with standard masking of already visited nodes).
> 2. If the rectification mask for this trajectory and time step satisfies $\mathcal{M}_{G_i}[j,t]=1$ and the expert successor $a_t^\*$ in $\tau_G^\*$ is feasible in the current state, we take **$a_t = a_t^*$** as the next action.
> 3. Otherwise, we sample $a_t$ from $\pi_\theta(\cdot \mid s_t)$ as in vanilla RL.
> 4. We then append $a_t$ to the partial trajectory and move to step $t+1$, repeating this procedure until the tour is complete.
>
> In other words, $\mathcal{R}$ is conceptually defined as operating on trajectories, but is *implemented* as a stepwise, autoregressive process that sometimes swaps the model's proposal with a feasible expert action at the current step. To make this clearer, we have added a pseudo-code style description (on TSP) below for your reference.
>
> ```python
> # B: batch_size, N: num_samples, M: num_nodes (consistent with the paper)
>
> class TSPTrainer:
>     # omit other functions ...
>     def _train_one_batch(self):
>         # data preparation ...
>
>         # sample from [alpha, beta] to decide the number of rectified actions
>         rectify_level = int(M * random.uniform(alpha, beta))
>         # construct the mask \mathcal{M} as in Eq. (9) in the paper: (B * N, M)
>         global_action_mask = torch.zeros(B * N, M).bool()
>         global_action_mask[:, :rectify_level] = True
>         global_action_mask.roll(random.randint(0, M - 1), dims=1)
>         # construct instance mask with prob. p_inst: (B * N,)
>         instance_mask = torch.rand(B * N) < p_inst if enable_rectify else None
>
>         while not done:
>             # extract intra-instance level mask from \mathcal{M}
>             action_mask = global_action_mask[:, self.env.selected_count].reshape(-1)
>             # forward pass, in which rectification is done
>             action, prob = self.model.forward(state, instance_mask, action_mask)
>             # update states and calculate rewards
>             state, reward, done = self.env.step(action)
>             # record probs for the rectified actions
>             prob_list = torch.cat((prob_list, prob[:, :, None]), dim=2)
>
>         # backpropagation ...
> ```

---

> ### Author Response · Authors · 2025-11-17
> **Author Response to Reviewer UfUe (Page 5)**
>
> ```
> class TSPModel(nn.Module):
>     # omit other functions ...
>     def forward(
>         self,
>         state: Step_State,
>         instance_mask: Tensor = None, # control instance-level rectifier
>         action_mask: Tensor = None    # control intra-instance rectifier
>     ):
>         """
>         Output the next action given current state,
>         with the action being optionally rectified.
>         d: hidden dimension
>         """
>         # load the reference expert trajectories
>         ref_tours = state.ref_tours
>         # set the first action as with POMO: [0, 1, ..., N]
>         if state.current_node is None:
>             selected = torch.arange(N)[None, :].expand(B, N)
>             prob = torch.ones(size=(B, N))
>         else:
>             # get node embeddings: (B, N, d)
>             node_embed = _get_encoding(state.current_node)
>             # decode probabilities for all candidate actions: (B, N, M)
>             # infeasible actions are masked with -infinity
>             probs = self.decoder(node_embed, ninf_mask=state.ninf_mask)
>             if self.training:
>                 # sample (initial) next action from current policy: (B, N)
>                 init_action = probs.reshape(B * N, -1).multinomial(1).squeeze(1).reshape(B, N)
>                 if instance_mask is not None:
>                     # prepare for rectification
>                     init_action = init_action.reshape(B * N)
>                     cur_node = state.current_node.reshape(B * N)
>                     unvisited = (state.ninf_mask == 0).reshape(B * N, -1)
>                     ref_tours = ref_tours.expand(B, N, M).reshape(B * N, M)
>                     # get rectified action
>                     rectified_action = _rectify_action(
>                         cur_node, init_action, action_mask,
>                         instance_mask, ref_tour, unvisited
>                     ).reshape(B, N)
>                     prob = _get_action_prob(probs, rectified_action)
>                 else: rectified_action = init_action
>
>         return rectified_action, prob
>
> def _rectify_action(
>     cur_node: Tensor,      # shape: (B * N,)
>     init_action: Tensor,   # shape: (B * N,)
>     instance_mask: Tensor, # shape: (B * N,)
>     action_mask: Tensor,   # shape: (B * N,)
>     ref_tour: Tensor,      # shape: (B * N, M)
>     unvisited: Tensor      # shape: (B * N, M)
> ):
>     # find the position of current node in the reference tours: (B * N,)
>     cur_node_idx = _get_idx_by_node(ref_tour, cur_node)
>     # get the position of the next node in the reference tours
>     ref_next_idx = (cur_node_idx + 1) % M
>     # get the next node in the reference tours: (B * N,)
>     ref_next_node = _get_node_by_idx(ref_tour, ref_next_idx)
>     # get the unvisited mask for the next reference action: (B * N,)
>     unvisited_mask = unvisited[torch.arange(B * N), ref_next_node]
>     # initialize new actions
>     rectified_action = torch.zeros_like(init_action).long()
>     # replace action if feasible
>     rectified_action = torch.where(unvisited_mask, ref_next_node, init_action)
>     # maintain the replaced action with instance and intra-instance rectification probabilities
>     rectified_action = torch.where(action_mask & instance_mask, rectified_action, init_action)
>
>     return rectified_action
> ```
>
> We hope that the detailed illustration above ameliorates your confusion caused by the mathematical notations. We have also clarified this point explicitly in the [revised manuscript](https://openreview.net/pdf?id=xniQIl8oTw) (**Appendix C**), and we thank you again for helping us improve the readability of our paper! We promise to open-source the full code upon publication.
>
> > **W8: Reference formatting**
> > The reference formatting does not fully conform to the ICLR citation style.
>
> **A:** Thank you for pointing this out. During preparation of the initial submission, we have carefully checked that the ICLR 2026 Author Guide does not prescribe a strict reference format, but we agree that adhering to the community's conventional style improves consistency and readability. We have therefore [revised our paper](https://openreview.net/pdf?id=xniQIl8oTw) with the bibliography and in-text citations to conform to the default ICLR reference formatting.

---

> ### Author Response · Authors · 2025-11-17
> **Author Response to Reviewer UfUe (Page 6)**
>
> ### **2. Responses to the questions:**
>
> > **Q1: Complementarity of hierarchical levels**
> > Are the three hierarchical levels (batch, instance, intra-instance) truly complementary? Could you provide ablations disabling each level independently?
>
> **A:** Thank you for raising this important question. We would like to clarify the complementary roles of the three hierarchical rectification levels.
>
> First, by design, the three levels operate in a **cascaded, progressively finer-grained manner**, where each level *filters* and *controls* how much rectification is applied (as illustrated in Fig. 1). Concretely:
>
> * The **batch-level** determines whether any rectification is applied to a batch at all.
> * The **instance-level** determines which sampled trajectories within a batch are rectified.
> * The **intra-instance level** controls *where* along a trajectory rectification is applied.
>
> Because these levels form a **chain of gating mechanisms, setting any level’s probability to zero immediately disables the entire rectification process**. This is intentional: it prevents over-dominating the learning process with expert supervision while preserving a healthy balance between expert guidance and free RL exploration, which is essential for preventing mode collapse and maintaining sampling diversity.
>
> That said, we fully agree that understanding the contribution of each level’s control is valuable. To this end, we conducted ablations by **loosening each level’s gating to the maximum**, i.e., setting $p_{\text{batch}} = 1$, $p_{\text{inst}} = 1$, or the intra–instance rectification probability to 1 (while keeping all other hyperparameters at their default values). This setup effectively removes the hierarchical filtering effect at that particular level, allowing us to assess how the absence of such control impacts the training dynamics.
>
> Ablation experiments were performed on representative **TSP-100** and **ATSP-100** benchmarks, each trained for **100 epochs** under identical training configurations. Our intention is to reveal meaningful trends within the limited time budget of the rebuttal phase.
>
> ### **Results on TSP-100:**
>
> | Setting                  | Obj.↓     | Drop↓      |
> | ------------------------ | --------- | ---------- |
> | w/o batch level          | 8.080     | 4.183%     |
> | w/o instance level       | 8.061     | 3.937%     |
> | w/o intra-instance level | 8.077     | 4.134%     |
> | **Full model**           | **8.039** | **3.655%** |
>
> ### **Results on ATSP-100:**
>
> | Setting                  | Obj.↓     | Drop↓      |
> | ------------------------ | --------- | ---------- |
> | w/o batch level          | 1.714     | 9.422%     |
> | w/o instance level       | 1.751     | 11.804%    |
> | w/o intra-instance level | 1.729     | 10.374%    |
> | **Full model**           | **1.668** | **6.504%** |
>
>
> These experiments clearly demonstrate the **dependent and complementary** roles of the three rectification levels. Removing the hierarchical gating at any level consistently degrades performance, confirming that the multi-level design is crucial for maintaining training stability, ensuring an effective balance between expert guidance and exploration, and achieving the best empirical performance.

---

> ### Author Response · Authors · 2025-11-17
> **Author Response to Reviewer UfUe (Page 7)**
>
> > **Q2: Applicability beyond routing problems**
> > Is the proposed approach applicable to non-routing combinatorial optimization problems?
>
> **A:** Thank you for this insightful question.
>
> - First, we would like to emphasize that the routing problem family itself constitutes a well-established and widely recognized research direction within the Neural Combinatorial Optimization (NCO) community. Numerous influential works published in top-tier venues have focused exclusively on routing problems [2–10], and their contributions have been broadly acknowledged as foundational rather than limited in scope.
>
> - From a technical perspective, our hierarchical rectification mechanism is particularly well-suited to routing tasks because their *geometric nature* involves rich topological connectivity and spatially ordered dependencies. These characteristics make routing problems especially challenging for neural models to learn effectively, and thus ideal for demonstrating the benefits of our intra-instance-level rectification. In contrast, tasks such as Maximum Independent Set (MIS) or Knapsack Problem (KP) exhibit permutation-invariant objectives, where the order of selected elements does not affect the outcome. Hence, our initial experiments primarily focus on routing problems, where local order and spatial structure play crucial roles.
>
> That said, we acknowledge the importance of evaluating generality beyond routing tasks. To this end, we have extended our experiments to two additional combinatorial optimization problems featuring non-trivial constraints: the **Capacitated Vehicle Routing Problem (CVRP)** and the **Knapsack Problem (KP)**.
>
> - **For CVRP**, we leverage a structural property of optimal solutions that the global route can be decomposed into multiple sub-tours. Accordingly, intra-instance-level rectification is applied to optimal sub-trajectories, specifically the route segment between a vehicle’s departure from and return to the depot. The decision to inject such an optimal sub-tour (with probability $p_{\text{sub}}$, instead of the $[\alpha, \beta]$ interval for TSP) is made only when the agent is positioned at the depot.
>
> - **For KP**, item selection is inherently order-independent. Since the intra-instance rectifier is designed to capture sequential relationships in locally ordered contexts, it is disabled for KP training. Instead, when a sampled trajectory is masked as *True* for instance-level rectification, the entire trajectory is replaced by a randomly permuted sequence that includes all items from the optimal reference solution.
>
> - **a. Results for uniform CVRP-50/100:** (LS: local search)
>
> | **Method**| **Type**   | **CVRP-50 (10k inst.)** ||| **CVRP-100 (10k inst.)** |||
> |:------- |:------ |:-----:|:-------:|:-------:|:-----:|:-------:|:-------:|
> |||        **Obj.↓**        | **Drop↓**  | **Time↓** |        **Obj.↓**         | **Drop↓**  | **Time↓** |
> | HGS                           | Heuristics |         10.366*         |   0.000%   |  1.005s   |         15.563*          |   0.000%   |  20.027s  |
> | *Heatmap-Guided Methods*      ||||||||
> | Fast-T2T| SL+G       |         12.640          |  21.835%   |  0.009s   |          19.202          |  23.333%   |  0.010s   |
> | COExpander| SL+G       |         11.979          |  15.407%   |  0.033s   |          17.497          |  12.343%   |  0.047s   |
> | Fast-T2T + LS| SL+G+LS    |         10.871          |   4.836%   |  0.013s   |          16.294          |   4.698%   |  0.018s   |
> | COExpander + LS| SL+G+LS    |         10.773          |   3.903%   |  0.037s   |          16.224          |   4.253%   |  0.055s   |
> | *Sequential Decision Methods* ||||||||
> | AM| RL+G       |10.98          |   5.86%    |     –     |          16.80           |   7.34%    |     –     |
> | POMO| RL+G|10.74          |   3.52%    |     –     |          16.15|   3.00%    |     –     |
> | Sym-NCO| RL+G       |         10.769          |   3.891%   |  0.087s   |          16.220          |   4.241%   |  0.166s   |
> | GOAL| SL+G       |10.906          |   5.193%   |  0.504s   |          16.342          |   5.005%   |  0.962s   |
> | **CORectifier (ours)**        | RRL+G       |       **10.540**        | **1.668%** |  0.042s   |        **15.939**        | **2.425%** |  0.079s   |
> | **CORectifier ($N_A=16$)**       | RRL+G       |       **10.447**        | **0.768%** |  0.049s   |        **15.799**        | **1.521%** |  0.108s   |
> | Sym-NCO + LS| RL+G+LS    |         10.505          |   1.910%   |  0.168s   |          15.933          |   2.379%   |  0.173s   |
> | GOAL + LS| SL+G+LS    |         10.628          |   2.519%   |  0.507s   |          15.959          |   2.548%   |  0.969s   |
> | **CORectifier + LS (ours)**   | RRL+G+LS    |       **10.469**        | **0.984%** |  0.045s   |        **15.796**        | **1.496%** |  0.085s   |
> | **CORectifier + LS ($N_A=16$)**  | RRL+G+LS    |       **10.412**        | **0.437%** |  0.052s   |        **15.706**        | **0.919%** |  0.113s   |

---

> ### Author Response · Authors · 2025-11-17
> **Author Response to Reviewer UfUe (Page 8)**
>
> - **b. Results for uniform CVRP-200/500:**
>
> | **Method**| **Type**   | **CVRP-200 (100 inst.)** ||| **CVRP-500 (100 inst.)** |||
> |:--- |:------- |:--------:|:------:|:---------:|:-----:|:--------:|:---------:|
> |||**Obj.↓**| **Drop↓** | **Time↓** |**Obj.↓**| **Drop↓** | **Time↓** |
> | HGS| Heuristics |19.630*|  0.000%|  60.024s  |37.154*|  0.000%   | 360.376s  |
> | *Heatmap-Guided Methods*||||||||
> | Fast-T2T| SL+G|25.064|  27.616%  |  0.059s   |47.749|  28.509%  |  0.091s   |
> | COExpander| SL+G|22.402|  13.977%  |  0.145s   |43.901|  18.199%  |  0.554s   |
> | Fast-T2T + LS| SL+G+LS|20.662|  5.290%|  0.063s|39.195|  5.530%|  0.215s   |
> | COExpander + LS| SL+G+LS|20.587|  4.893%   |0.153s|39.121|  5.337%   |  0.605s|
> | *Sequential Decision Methods*    ||||||||
> | Sym-NCO| RL+G|20.662|  5.274%|  0.320s   |40.382|  8.723%|  0.769s   |
> | **CORectifier (ours)**|RRL+G|**20.270**|**3.260%**|0.159s|**39.129**|**5.343%**|0.368s|
> | **CORectifier ($N_A=16$)**| RRL+G|**20.129** |**2.541%**|0.219s|**38.874**|**4.650%**|1.485s|
> | Sym-NCO + LS| RL+G+LS|20.193|  2.880%   |  0.341s   |38.700|4.173%|  0.883s   |
> | **CORectifier + LS (ours)**| RRL+G+LS   |**20.032**|**2.052%**| 0.171s|**38.329**|**3.185%**|0.461s|
> | **CORectifier + LS ($N_A=16$)**  | RRL+G+LS   |**19.952**|   **1.638%**   |0.237s|**38.176**|**2.758%**|1.567s|
>
> - **c. Zero-shot generalization results on CVRPLib:** 1000 instances total: 22 within (100, 200] clients and 78 within (203, 1000] clients. Results quoted from PO4CO[1].
>
> | **Solver**| **Type**   | **(0, 200] Drop↓** | **(200, 1000] Drop↓** | **Total Drop↓** |  **Time**  |
> | :--------- | :-------- | :--------: | :---------: | :----: | :--------: |
> | LKH3| Heuristic|0.36 %       |         1.18 %        |      1.00 %     |    16 m    |
> | HGS| Heuristic      |       0.01 %       |         0.13 %        |      0.11 %     |    16 m    |
> | NeuroLKH| Heuristic + SL |       0.47 %       |         1.16 %        |      0.88 %     |    16 m    |
> | LEHD| SL             |       11.11 %      |        12.73 %        |     12.25 %     |   1.67 s   |
> | BQ-NCO| SL             |       10.60 %      |        10.97 %        |     10.89 %     |   3.36 s   |
> | POMO| RL             |       5.26 %       |        11.82 %        |     10.37 %     |   0.80 s   |
> | Sym-NCO| RL             |       9.99 %       |        27.09 %        |     23.32 %     |   0.87 s   |
> | Omni-POMO| RL             |       5.04 %       |         6.95 %        |      6.52 %     |   0.75 s   |
> | ELG (RF)| RL             |       4.51 %       |         6.46 %        |      6.03 %     |   1.90 s   |
> | ELG (PO)| RL|       4.39 %       |         6.37 %        |      5.94 %     |   1.90 s   |
> | **CORectifier (ours)** | RRL|     **3.49 %**     |       **5.98 %**      |    **5.47 %**   | **2.11 s** |
>
>
> - **d. Results for KP-50/100/200/500:**
>
>
> | **Method**             | **Type** | **KP-50 (1280 inst.)** |            | **KP-100 (1280 inst.)** |            | **KP-200 (1280 inst.)** |            | **KP-500 (1280 inst.)** |            |
> |:---------------------- |:-------- |:----------------------:|:----------:|:-----------------------:|:----------:|:-----------------------:|:----------:|:-----------------------:| ---------- |
> |                        |          |       **Obj.↑**        | **Drop↓**  |        **Obj.↑**        | **Drop↓**  |        **Obj.↑**        | **Drop↓**  |        **Obj.↑**        | **Drop↓**  |
> | OR-Tools               | Exact    |        20.021*         |   0.000%   |         40.302*         |   0.000%   |         57.402*         |   0.000%   |         91.128*         | 0.000%     |
> | AM                     | RL+G     |           –            |   0.173%   |            –            |   0.211%   |            –            |   0.325%   |            –            | –          |
> | POMO                   | RL+G     |           –            |   0.130%   |            –            |   0.190%   |            –            |   0.500%   |            –            | 6.410%     |
> | BQ-NCO                 | SL+G     |           –            |     –      |            –            |   0.100%   |            –            |   0.140%   |            –            | 0.740%     |
> | GOAL                   | SL+G     |           –            |     –      |            –            |   0.120%   |            –            |   1.630%   |            –            | 2.400%     |
> | **CORectifier (ours)** | RRL+G     |       **20.018**       | **0.013%** |       **40.298**        | **0.012%** |       **57.397**        | **0.009%** |       **91.121**        | **0.007%** |

---

> ### Author Response · Authors · 2025-11-17
> **Author Response to Reviewer UfUe (Page 9, end)**
>
> These additional experiments demonstrate that CORectifier is not confined to routing tasks. Rather, it can flexibly adapt its rectification strategy to different problem structures, including non-sequential and constraint-heavy combinatorial optimization settings.
>
> We have also included a detailed problem formulation (**Appendix B**), data generation procedure (**Sec. 4.1**), comprehensive experimental results (**Tables 5, 8, 9, 15, 16, 19**), and model configuration details (**Table 13**) for these supplementary task evaluations in the [revised manuscript](https://openreview.net/pdf?id=xniQIl8oTw), ensuring full narrative consistency with the main paper structure. We sincerely hope that these extended studies help alleviate your concern on this matter.
>
> > **Q3: Definition of the notation $M$**
> > The definition of $M$ as problem size seems inconsistent. In the Intra-instance Level section, it sometimes appears as the decision sequence length. Please clarify.
>
> **A:** Thank you for catching this minor inconsistency and for giving us the chance to clarify. Throughout the paper, the notation $M$ is intended to denote the *problem size* (e.g., the number of nodes in TSP-like problems, the number of clients in CVRP, or the number of items in KP). In our initial submission, since the studied problem types (mainly TSP-family tasks) have a one-to-one correspondence between the problem size and the decision trajectory length, we used $M$ uniformly for simplicity.
>
> However, when extending our study to non-routing tasks, we realized that this notation could lead to ambiguity (e.g., in KP, the trajectory length can vary depending on capacity and item weights). To address this, we have **introduced a new notation $M'$** in the [revised manuscript](https://openreview.net/pdf?id=xniQIl8oTw) (line 203) to explicitly represent the *length of the decision trajectory*, along with a clear explanation of its relationship with $M$ and our consideration for narative simplicity. Typically, $M'$ is a function of $M$, as larger problem instances naturally involve longer decision chains, so this clarification does not affect our core methodology or the analysis of how CORectifier mitigates RL limitations as CO problem scales increase. Again, we appreciate your careful review and this valuable question!
>
> ---
>
> ### **References:**
> [1] Preference optimization for combinatorial optimization problems, ICML 2025.
>
> [2] Attention, Learn to Solve Routing Problems! ICLR 2019
>
> [3] Sym-NCO: Leveraging Symmetricity for Neural Combinatorial Optimization, NeurIPS 2022
>
> [4] MVMoE: Multi-Task Vehicle Routing Solver with Mixture-of-Experts, ICML 2024
>
> [5] Learning Collaborative Policies to Solve NP-hard Routing Problems, NeurIPS 2021
>
> [6] UCPO: A Universal Constrained Combinatorial Optimization Method via Preference Optimization, AAAI 2026
>
> [7] UniCO: On Unified Combinatorial Optimization via Problem Reduction to Matrix-Encoded General TSP, ICLR 2025
>
> [8] Unify ML4TSP: Drawing Methodological Principles for TSP and Beyond from Streamlined Design Space of Learning and Search, ICLR 2025
>
> [9] GLOP: Learning Global Partition and Local Construction for Solving Large-scale Routing Problems in Real-time, AAAI 2024
>
> [10] Unsupervised Learning for Solving the Travelling Salesman Problem, NeurIPS 2023
>
> ---
> Once again, we sincerely thank you for your time, thoughtful evaluation, and constructive feedback. We hope that our clarifications have adequately addressed your concerns and resolved potential misunderstandings. **We would be truly grateful if, upon your kind reconsideration of our supplementary efforts, you find that our work offers meaningful contributions to the NCO/RL4CO community and merits a positive assessment**. We remain fully open to further discussion and would greatly appreciate any additional comments you may have.
>
>
> Sincerely,
>
> Authors of Submission 11119

---

### Author Response · Authors · 2025-11-26
**Global Response by Authors to AC and All Reviewers (1/2)**

## **Global Response by Authors**

---
## **Dear Area Chair and Reviewers,**

We sincerely thank you for your time and for the reviewers’ careful evaluations and constructive suggestions. During the rebuttal period, we have **made *every* effort to address all raised questions and concerns in full**, providing detailed clarifications, substantial supplementary experiments, and a carefully updated version of the manuscript reflecting all improvements made in response to the feedback.

- ### **As the author–reviewer discussion phase has now passed its midpoint, we note that we have *NOT* yet received *ANY* feedback from the reviewers. We therefore remain eager to learn whether our initial responses and additional analyses have sufficiently addressed the raised concerns.**

- To facilitate further evaluation, we respectfully provide below a **concise global summary** of the paper’s key motivations, technical contributions, and the additional experimental evidence contributed during the rebuttal, for your convenient reference.

---
### **Summary of Our Work and Key Technical Contributions:**

This work is motivated by three long-standing challenges of RL-based neural combinatorial optimization (RL4CO): **premature exploration collapse**, **extremely sparse terminal rewards**, and **severe performance degradation as problem scale increases**. These issues fundamentally limit the stability, efficiency, and scalability of sequential RL solvers, especially beyond small benchmark sizes.

To address these challenges, we propose **CORectifier**, a **hierarchical trajectory rectification framework** that injects ***limited, structured expert guidance*** directly into the RL training process. Rather than replacing exploration with imitation, CORectifier introduces rectification at the **batch**, **instance**, and **intra-instance** levels, enabling fine-grained and probabilistic correction of sampled trajectories while preserving exploration diversity and interpretability. This design effectively improves gradient quality, mitigates entropy collapse, and alleviates reward sparsity by providing denser, structure-aware learning signals along long decision sequences.

**Technical contributions:**

1) We propose a new learning paradigm for solving COPs with a novel rectification mechanism to resolve RL-methods' sample efficiency and reward sparsity.
2) To our best knowledge, our method marks the earliest NCO attempts to explore the synergy between RL and SL/IL, with hierarchical guiding signals operated at batch-wise, instance-wise, and intra-instance levels.
3) We adapt our paradigm to 3 mainstream RL-based CO models and achieves enhanced training and solution quality across cases, validating the **general applicability of our framework**.
4) We expand the comparison spectrum for RL-based methods to include neural solvers beyond just RL variants, showing up to **59.7\%** and **26.5\%** gains (w/o augmentation) over RL/SL methods, and improving RL solvers on TSP-500 by up to **89.8%**.

These results highlight CORectifier as a meaningful step toward more scalable and reliable RL-based combinatorial optimization.

---

### Author Response · Authors · 2025-11-26
**Global Response by Authors to AC and All Reviewers (2/2)**

## **Global Response by Authors**

---
### **A Roadmap for Comprehensive Experiments:**

---
| Experiment Type| Location in paper section / Rebuttal post| Summary| Note|
| - | - | - | - |
| **Main Results: TSP**| Tables 1 (main)| Results on TSP-50/100/500, the first RL-based method to compare with both sequential-decision methods and heatmap-guided methods on TSP > 100. **Relative drop reduction within RL4CO: 47.69%**           ||
| **Main Results: ATSP**| Tables 2 (main)                                         | Results on ATSP-50/100/200/500, the first RL-based method to compare with both sequential-decision methods and heatmap-guided methods on ATSP $\ge$ 200. **Relative drop reduction within RL4CO: 32.05%** |     |
| **Main Results: PCTSP**             | Tables 4 (main)                                         | Results on PCTSP-20/50/100, validating the capability of capturing **more complex constraints** and **adaptability to more backbones** (AM). **Relative drop reduction within RL4CO: 59.67%**             |     |
| **Main Results: CVRP**| Tables 5 (main)                                         | Results on CVRP-50/100/200/500, validating the capability of capturing **more complex constraints**. **Relative drop reduction within RL4CO: 37.22%**|     Suppelented to address **Q2 (UfUe)**, **W1 (xVoR)**, **W2 (1exv)**, **W1 (sQuT)** |
| **Main Results: KP**| Tables 9 (main), Table 19 (Appendix F.5, full)          | Results on KP-50/100/200/500, validating the capability of learning **non-graph COPs**, with over **10x** performance improvement|Suppelented to address **Q2 (UfUe)**, **W1 (xVoR)**, **W2 (1exv)**, **W1 (sQuT)**|
| **Ablation Study: RRL paradigm**    | Table 3; Fig. 3, 4 (main)                               | **Both numerical and visual** comparison of **ours, only RL, only IL, and w/ & w/o IL pretraining**; validates the synergy of our proposed hierarchical integration of IL and RL.                         |     |
| **Ablation Study: 3 gating levels** | Response to Reviewer UfUe (Page 6)                      | Demonstrate the **dependent and complementary roles of the three rectification levels**| Suppelented to address **Q1 (UfUe)**|
| **Generalization Study**            | Table 7, 8 (main); Table 14-16 (Appendix F.1 & F.2)     | **Robust zero-shot generalization to real-world benchmarks**, TSPLIB and CVRPLIB, with per-instance performance given.|CVRP part is suppelented to address **Q2 (UfUe)**, **W1 (xVoR)**, **W2 (1exv)**, **W1 (sQuT)**|
| **Hyper-parameter Studies**| Fig. 5, Table 6 (main); Fig. 7, Table 17 (Appendix F.3) | **Studying sensitivity of $p_{batch}$, $p_{inst}$, $[\alpha,\beta]$**, confirming the stability and design choices.|Supplemented finer-grained results (p<0.1) to address **W5/W6 (UfUe)**|
| **Scalability Study**| Table 10 (main) |Discussing the **lingering bottleneck** on graph-based problems with more than 100 nodes in RL4CO practices. **Relative drop reduction within RL4CO: 89.8%** ||
| **Stability Study: Metric Variance** | Table 11; Table 18 (Appendix F.4) | Confirming **solving stability and statistical significance** with reported standard deviations of the optimality drop over instances |
| **Stability Study: Less Expert Guidance** | Response to Reviewer xVoR (Page 3) | Confirming **learning stability** with 1.28k/12.8k/128k expert data, validating the **robustness under sparse supervision** | Supplemented to address **W2 (xVoR), W1 (1exv), W3 (sQuT)**|
| **Diversity Studies** | Fig.6 (main), Fig. 8-10 (Appendix F.6) | visualizing the average **advantage values and distributions**, and comparing **trajectory entropies**; validating the increase in sample diversity | Supplemented to address **W1 (UfUe)**, **W4/W5 (sQuT)**|
| **Feasibility Studies** | Fig. 11, Fig. 20 (Appendix F.7) | Visualizing the per-step feasibility rate; reporting avaerage feasibility, confirming **sufficient constraint satisfaction** over the rectification actions | Supplemented to address **W3 (UfUe)**, **W5 (sQuT)**|

---
In addition to the extensive empirical studies, we also provided detailed **clarifications on the rectification operator $\\mathcal{R}$**, including a **PyTorch-style pseudo-code example** for illustration (**Appendix C.1**) to address **W7 (UfUe)** and **W2/Q2 (sQuT)**, a **comprehensive explanation of the masking mechanisms required for handling complex constraints** such as those in KP and CVRP (**Appendix C.2**), and a **thorough refinement of the main text** to improve organization, clarity, and overall readability.


---
### **We sincerely hope to learn whether our continued efforts have adequately addressed the reviewers’ concerns and, if so, whether our work might merit reconsideration at this stage. We uphold the review process and open discussion encouraged by ICLR, and we greatly appreciate your time and look forward to your feedback.**

Best,

Authors of Submission 11119

---

### Meta-Review · Area_Chair_tryb · 2026-01-07

**Summary:**

After carefully checking the paper, the reviews, the rebuttal, and the author-reviewer discussions, I think the weak points outweight the strong points. All reviewers agreed that the paper has a fundamental limitation in that its problem formulation and solution rely heavily on expert-provided data. After carefully reviewing the manuscript and the rebuttal, I believe the authors have not adequately addressed this concern. The proposed approach should be validated in more complex and realistic settings to demonstrate its effectiveness. The weaknesses are not likely to be fixed in the camera-ready version. Thus, I recommend rejecting this paper.

**Reviewer Concerns:**

The score remains unchanged. I have read the rebuttal, and believe it does not affect the review. I believe the rebuttal resolved some issues regarding the experimental setup and ablation studies. However, the key concerns about the motivation, generality, and efficiency of the proposed method remain unaddressed.

**Reviewer Scores:**

The score will remain unchanged. I have read the rebuttal, and believe it will not affect the scores.

---

### Decision · Program_Chairs · 2026-01-26

Reject